# The interaction of Solar Radiation Modification with Earth System Tipping Elements

Gideon Futerman[1], Mira Adhikari[2], Alistair Duffey[3], Yuanchao Fan[4], Jessica Gurevitch[5], Peter Irvine[3], Claudia Wieners[6]

[1]Department of Earth Sciences, University of Oxford, Oxford, OX1 3AN, United Kingdom
[2]Department of Geography, Kings College London, London, WC2B 4BG, United Kingdom
[3]Department of Earth Sciences, University College London, WC1E 6BT, United Kingdom
[4]Institute of Environment and Ecology, Tsinghua Shenzhen International Graduate School, Tsinghua University, Shenzhen, 518000, China
[5]Department of Forestry and Natural Resources, Purdue University, West Lafayette, IN 47907, United States of America
[6]Institute for Marine and Atmospheric Research, Utrecht University, 3584 CC, Netherlands

*Correspondence to*: Gideon Futerman (GFuterman@hotmail.co.uk)

**Abstract.** The avoidance of hitting tipping points has been invoked as a significant benefit of Solar Radiation Modification (SRM) techniques, however, the physical science underpinning this has thus far not been comprehensively assessed. This review assesses the available evidence for the interaction of SRM with a number of earth system tipping elements in the cryosphere, the oceans, the atmosphere and the biosphere, with a particular focus on the impact of Stratospheric Aerosol Injection. We review the scant available literature directly addressing the interaction of SRM with the tipping elements or for closely related proxies to these elements. However, given how limited this evidence is, we also give a first-order indication of the impact of SRM on the tipping elements by assessing the impact of SRM on their drivers. We then briefly assess whether SRM could halt or reverse tipping once feedbacks have been initiated. Finally, we suggest pathways for further research. We find that, when temperature is a key driver of tipping, well-implemented, homogenous, peak-shaving SRM could be at least partially effective at reducing the risk of hitting most tipping points examined relative to the same emission pathway scenarios without SRM. Nonetheless, very large uncertainties remain, particularly when drivers less strongly coupled to temperature are important, and considerably more research is needed before many of these large uncertainties can be resolved.

# 1 Introduction

Climate Change caused by anthropogenic greenhouse gas (GHG) emissions is increasingly recognised as a major threat to human and ecological systems (IPCC, 2023). One aspect of climate change that is

gaining increased attention are earth system tipping points (Lenton et al., 2023), which are seen as potentially triggering dangerous changes increasing the risk of negative impacts of anthropogenic climate change and thus demand action to reduce the likelihood of hitting them (Lenton et al., 2019). These impacts of climate change also have to be considered alongside the growing crisis of biodiversity loss, which is less widely recognised but is nonetheless dangerously pushing ecological systems towards lower biodiversity states (Legagneux et al., 2018). Climate change and biodiversity loss may influence and reinforce each other (climate-induced habitat loss; reduced CO2 uptake).

Solar Radiation Modification (SRM, a.k.a. Solar geoengineering) has been proposed as a set of methods that could ameliorate some of these climate risks by reflecting a fraction of incoming sunlight and to cool the Earth directly, and is gaining salience at national (National Academies of Sciences and Medicine, 2021) and international (United Nations Environment Programme, 2023) levels. SRM has been discussed in the context of these growing dangers to humans and the biosphere from tipping points (Bellamy, 2023; Heutel et al., 2016; National Academies of Sciences and Medicine, 2021), but thus far, no comprehensive review of the impact of SRM on a variety of earth system tipping elements have been performed. We discuss the potential for SRM to help avoid, postpone or precipitate hitting tipping points in the cryosphere, atmosphere, oceans, and biosphere, with particular attention to the impact on the drivers of tipping in these systems, as well as assess the possibility of SRM reversing tipping once tipping points have been hit.

## 1.1 Tipping Elements

Several definitions for tipping elements in the earth system have been suggested (Armstrong McKay et al., 2022; Lenton et al., 2008; Van Nes et al., 2016). While details differ, their common denominator is that at a critical threshold (the tipping point) a small additional change in some driver leads to qualitative changes in the system (e.g., Fig. 1a,b). As explicitly stated in Armstrong McKay et al., (2022) and  Van Nes et al. (2016), and described in nearly all examples in Lenton et al. (2008), these qualitative changes are brought about by self-perpetuating processes caused by  positive feedbacks which drive the system to a new state. While the "state" of climate tipping elements can often be characterised by a single indicator, for example the mass of the Greenland ice sheet, this may not hold for ecological systems, which may have a variety of stable assemblages (Fig. 1f).

We use the word "driver" for the key variables external to the system that initiate the relevant changes, and "dynamics" for the self-accelerating processes that accomplish the tipping. Typically, once these processes have kicked in, they will continue even if the drivers stop increasing, or even decrease. An edge case is threshold-free feedbacks, such as Marine Methane Hydrates (Armstrong McKay et al.,

2022; Lenton et al., 2008; Van Nes et al., 2016), systems in which positive feedbacks play a role but are
not strong enough to lead to run-away processes (Fig. 1e). These are commonly discussed alongside
tipping elements, so some examples will be discussed here. When referring collectively to the systems
discussed in this article, we will use the term 'tipping element' and only classify further where
necessary.

Not just the magnitude, but also the trajectory of drivers may determine whether tipping occurs. For
example, ice sheets have long response times and may only tip if the temperature overshoot is of
sufficient duration (Ritchie et al., 2021; Wunderling et al., 2022a). On the other hand, some tipping
elements may be more susceptible to fast changes than to slow changes (rate-induced tipping, Fig. 1d),
even if the eventual magnitude of the change is the same (Ashwin et al., 2012). Some systems may have
more than one driver (e.g., precipitation change and deforestation in the Amazon).

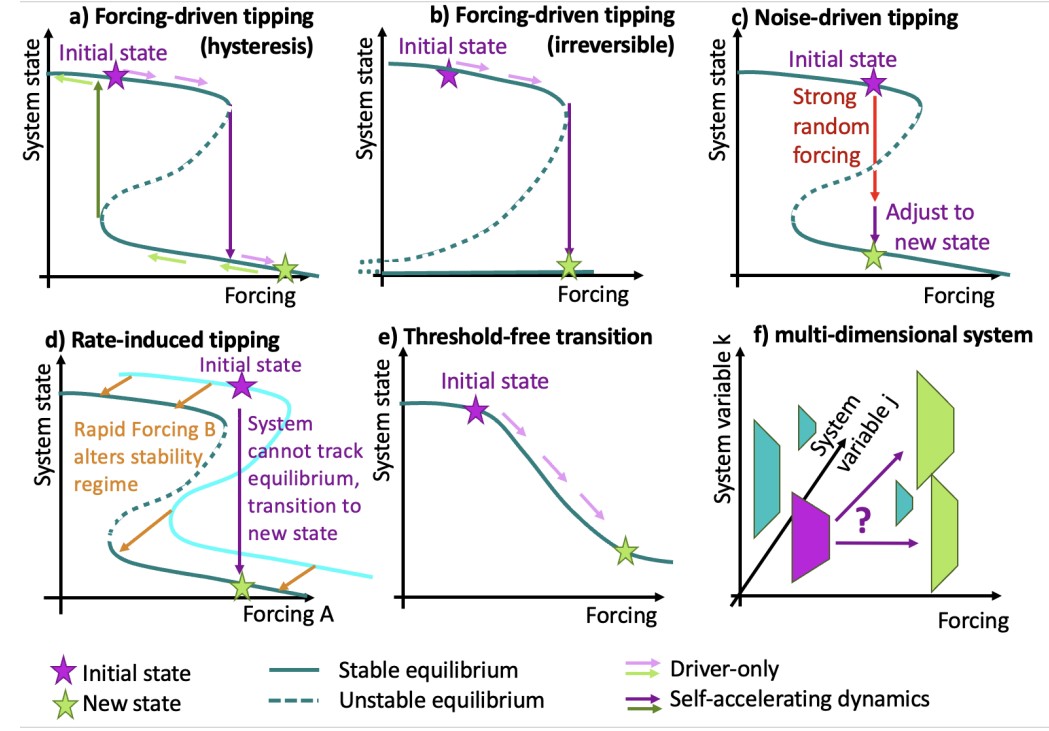

*Figure 1 Different tipping processes. Solid (dashed) lines denote stable (unstable) equilibria. a,b)*
*Drivers (change in forcing) push the system closer to the tipping point; when it is reached, the system*
*undergoes self-perpetuating changes ("feedbacks") and reaches a new state. The process can be*
*reversible (possibly with hysteresis) if the forcing is reverted (a) or completely irreversible (b; e.g. loss*
*of a specific ecosystem assemblage due to species extinction). c) Random fluctuations push the system*

*into an alternative state even before the actual tipping point is reached; easier if already close to*
*tipping point. d) Rapid forcing changes prevent the slowly evolving system from tracking its original*
*equilibrium state, causing a transition (rate-dependent tipping). e) Threshold-free feedbacks lead to*
*strong system changes under forcing, but no self-reinforcing dynamics (tipping) occurs. f) Complex*
*systems (e.g. ecological systems) cannot necessarily be captured by a single system variable and may*
*have many equilibrium states; final outcome may e.g. depend on precise forcing trajectory.*

Armstrong McKay et al. (2022) tie their tipping points to global warming thresholds. However, a tipping element may have other climate drivers, e.g. precipitation in the Amazon region, thus making the tipping point not merely global-temperature-related. When only greenhouse-gas-induced climate change is considered, one might assume that non-temperature drivers scale with GMST, which acts as proxy for the overall strength of climate change. However, if SRM is considered, other climate drivers do not necessarily scale with GMST; for example, SRM may restore GMST but fail to restore precipitation in the Amazon (Jones et al., 2018). Especially in ecological systems, drivers not related to climate, such as human-induced deforestation, also play a key role (Sect. 5.2).

**1.2 Solar Radiation Modification**

While phasing out (net) greenhouse gas emissions remains the only way to address the root cause of climate change, various climate intervention approaches have been suggested to complement mitigation and reduce global warming and its impacts. This includes Solar Radiation Modification (SRM), a set of proposed technologies aimed at increasing the earth's albedo, reducing incoming solar radiation and thus reducing global surface temperatures (National Academies of Sciences and Medicine, 2021). Stratospheric Aerosol Injection (SAI) is currently the best researched and the most plausible candidate to generate significant, fairly homogeneous cooling, and thus is the deployment method primarily discussed in this article. SAI would mimic the effect of large volcanic eruptions by injecting particles or precursor gas (most commonly suggested is SO2) into the stratosphere to create a thin reflective aerosol cloud.

Even if SRM can be used to reverse Global Mean Surface Temperature (GMST) rise from increasing Greenhouse Gas concentrations (Tilmes et al., 2020), it does not reverse the anthropogenic greenhouse effect, but acts through a different mechanism, i.e. reflecting sunlight. This means that SRM does not cancel the effect of increased greenhouse gas concentrations perfectly. Although modelling studies suggest that SRM might bring many relevant climate variables closer to their pre-industrial values (Irvine et al., 2019), residual changes to atmospheric, oceanic and ecological systems would remain. SRM might introduce additional effects, such as changes in regional hydrological cycles relative to both

same emission scenarios and same temperature scenarios (Ricke et al., 2023), or changes in the balance between direct and indirect solar radiation. Alongside its physical impacts, the possible political and societal effects of SRM may be equally important, including the risk of conflict (Bas and Mahajan, 2020), mitigation deterrence (McLaren, 2016), and issues of imperialism (Surprise, 2020), democracy (Stephens et al., 2021) and justice (Horton and Keith, 2016; Táíwò and Talati, 2022). We stress that the risks and potential benefits of SRM does not solely depend on its effects on climate, including tipping points, but would have to be assessed in a holistic risk assessment framework.

SRM implementation could follow many scenarios, with various background greenhouse gas trajectories, SRM approaches (SAI or alternatives), deployment sites, starting and end times, and intensities (MacMartin et al., 2022), potentially including a mix of more or less coordinated regional approaches (Ricke, 2023). Unless otherwise specified, we assume a "peak-shaving" scenario, i.e. background greenhouse gas trajectory that would lead to a potentially large, multi-decade temperature overshoot, which is eventually brought under control by negative emission technologies. Against this background, SAI is used to produce a largely homogeneous cooling that limits global mean surface temperature (GMST) overshoot to a constant target, such as 1.5ºC above pre-industrial, resembling (MacMartin et al., 2018; Tilmes et al., 2020). Unless specified, we assume the impacts of SRM are relative to the same emissions pathway without SRM deployment.

## 1.3 Solar Radiation Modification and Tipping Elements

SRM might prevent earth sub-systems (tipping elements) from crossing tipping points, or it might push systems over tipping points. In ecological systems, which have many drivers and many possible states, it is also possible that both SRM and climate change without SRM would lead to hitting different tipping points within the same tipping element. The question may then not be *whether* tipping can be caused or prevented, but *which* tipping will occur under certain conditions.

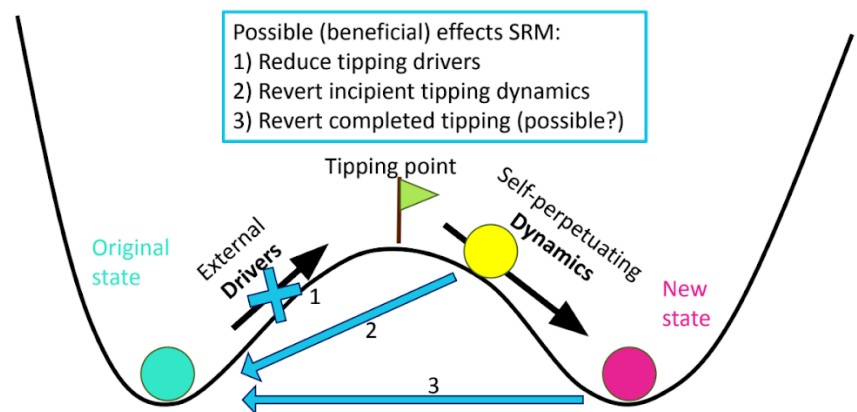

*Figure 2. Possible ways by which SRM could counteract tipping.*

*1) Reducing drivers of tipping before the critical threshold (tipping point) is reached. 2) Reverting tipping dynamics (shortly) after it is initialised, but before tipping is completed, such that the tipping feedbacks have begun but the process is not yet complete. 3) Revert tipping after it is completed. This may not be possible or practicable in many cases. While not depicted here, SRM may also adversely affect some tipping points.*

SRM may prevent tipping in several ways (Fig. 2).  First, SRM may *prevent* a tipping point from being reached by reducing or counteracting drivers of tipping. This would require a timely implementation of SRM, i.e. before the tipping point is reached.  If SRM were terminated before other measures (e.g. negative emissions) are in place to reduce drivers, SRM may only postpone tipping. Moreover, if insufficient amounts of SRM were used - maintaining, for example, a constant SRM forcing rather than the constant Global Mean Surface Temperature (GMST) assumed in the peak shaving scenario - SRM may also only postpone tipping.

In the absence of direct (modelling) evidence on SRM's impact on a tipping element, a first indication can be obtained by studying how SRM might affect known drivers. If the relevant drivers roughly scale with GMST, we expect that SRM would reduce the likelihood of tipping compared to the same GHG concentration without SRM. If the key drivers are precipitation, regional climate or other factors that are not directly related to global temperature, then the effect of SRM might be harder to determine, particularly due to our much higher uncertainty in modelling studies of the impact of SRM on these climatic variables. Some of these drivers may also strongly depend on the design of the SRM scheme.

SRM might conceivably revert tipping if tipping dynamics has already started (process 2 in Fig. 2), but not completed, or even after completion (process 3 in Fig. 2). As the complexity of the feedbacks and nature of hysteresis are generally less well understood than the initial drivers, the potential for reversal is often much harder to assess, especially in the absence of dedicated studies. It would be difficult in practice to design SRM for reverting incipient tipping (similar to "emergency deployment" discussed in Lenton (2018)), because precise prediction of the onset of tipping is impossible (Lenton, 2018). Reversal of completed tipping, even if theoretically possible, might require unfeasibly high SRM intensities in case of hysteresis, and would likely play out over timescales much larger than policy timescales. Therefore we will not explicitly discuss it. Our main focus is prevention of tipping drivers, because more evidence is available and because it may be more practically relevant for near-term decision-making. Reversal (process 2 in Fig. 2) will be discussed where appropriate.

This study reviews a number of key tipping elements and threshold-free feedbacks, largely following those laid out in Armstrong McKay et al. (2022). We aim to provide a preliminary analysis of the interaction of SRM with a wide - but not exhaustive - range of tipping elements. Each section is then structured as follows. Firstly, we assess the drivers and mechanisms of the tipping process. This was done to allow us to then review the impact of SRM on these drivers to give a first order indication of whether SRM could prevent - and to a lesser extent, if it could reverse - tipping. Where available, we also review direct modelling evidence of the effect of SRM on the tipping elements, although many of the models used don't have sufficient complexity to actually show tipping dynamics in the elements, which is a limitation. Finally, we provide recommendations for future research.

**1.4 Results overview**

| Tipping Element | Effect on Drivers | Reversibility | Strength of evidence base |
|---|---|---|---|
| Greenland Ice Sheet collapse (GIS) (Sect. 2.1) | DC: **Atmospheric warming (+, Eff)** Precipitation (-, Part-Over) <br><br> **Overall: Partial-Effective compensation (??)** | **Likely ineffective.** While destabilisation of GrIS could be prevented, reversing previous losses is not possible on multidecadal/centennial timescales due to ice sheet inertia | **Intermediate** - basic theory and several model studies suggest SAI could offset drivers, limited evidence on reversibility |

| | | | |
|---|---|---|---|
| Antarctic Ice Sheet collapse (AIS) (Sect. 2.2) | DC: Atmospheric warming (+, Part-Eff) Ocean warming (+, No-Part) Precipitation (-, Part-Eff) CA: Circumpolar deep water driven melt (+, Worse-No)<br><br>**Overall: Unknown(???)** | **Likely ineffective.** As ocean thermal forcing is the primary driver of current mass loss, reversal would be difficult on decadal to centennial timescales due to ocean and ice sheet inertia. | **Weak** - the Marine Ice Cliff Instability tipping point is largely theoretical and few studies exist on SAI's impacts on Antarctica. |
| Mountain Glacier loss (MG) (Sect. 2.3) | DC: Atmospheric warming (+, Part-Eff) Precipitation (-, Part-Over)<br><br>**Overall: Partial-Effective compensation (?)** | **Likely partially effective.** Atmospheric cooling could reverse the surface elevation feedback, depending on how much surface elevation has decreased. Cooling may also increase precipitation falling as snow. | **Intermediate** - basic theory and several model studies suggest SAI could offset most drivers, but limited evidence on reversibility and glaciers outside mid latitude Asia. |
| Winter Arctic sea-ice abrupt loss (WASI) (Sect. 2.5) | DC: **near-surface atmospheric warming** (+, Part)<br><br>**Overall: Partial compensation (??)** | **Likely effective** with sufficient local cooling. | **Intermediate** – supported by several studies, including inter-modal comparisons, and theory, although no study explicitly assesses the impact of SAI on threshold behaviour. |
| Summer sea-ice decline, both Arctic and Antarctic (SSI) (Sect. 2.5) | DC: **near-surface atmospheric warming** (+, Part-Eff) CA: Ocean and atm. circulation (+/-,Unk)<br><br>**Overall: Partial-Effective compensation (?)** | **Likely effective** with sufficient local cooling. | **Intermediate** – supported by several studies, including inter-modal comparisons, and theory |

| | | | |
|---|---|---|---|
| Boreal permafrost thaw (BPF) (Sect. 2.6) | DC: **soil warming** (+, Eff) Increased precipitation (+, Eff), CA: increased wildfire (+, Unk), vegetation change (+/-, Unk) <br><br> **Overall: Effective compensation (??)** | **Likely ineffective** for abrupt thaw. Gradual thaw is likely a threshold-free feedback process without tipping dynamics. | **Intermediate** – supported by several studies, and basic theory for the main driver. However, various processes impacting GHG release from permafrost thaw are not captured in current ESMs. |
| Marine methane hydrates loss at continental shelf (MMC) <br><br> (Sect. 2.7) | DC: **ocean warming (at shelf depth)** (+, Unk) <br><br> **Overall: Unknown(???)** | **N/A** – methane release from hydrates is likely a threshold-free feedback process without large-scale tipping dynamics. The carbon that had been previously released would remain in the atmosphere after SRM deployment. | **Weak** – no studies directly assess the impact of SRM. |
| Atlantic Meridional Overturning Circulation collapse (AMOC) (Sect. 3.1) | DC: **Surface ocean warming** (+,Part-Eff), Precip - Evap increase (+, Eff-Over), CA: **Greenland ice loss** (+,Part-Eff), Sea ice loss (+?, Eff) <br><br> **Overall: Partial-Over compensation (??)** | **Uncertain, but possibly partially effective.** Surface cooling might help restart deep convection and deepwater formation. Sea ice expansion may however impede surface heat loss | **Intermediate**. Several modelling studies suggest SRM reduces weakening; models may underestimate AMOC stability. |
| Sub-Polar Gyre collapse (SPG) (Sect. 3.2) | DC: **Surface ocean warming** (+,Part-Eff), Precip - Evap increase (+, Eff-Over), CA: **Greenland ice loss** (+,Part-Eff), Sea ice loss (+?, Eff) <br><br> **Overall: No-Effective compensation  (???)** | **Uncertain, but possibly partially effective.** Surface cooling might help restart deep convection.  Sea ice expansion may however impede surface heat loss. | **Weak.** Model disagreement about whether and when SPG could tip. Only one model study dedicated to SRM effect on SPG. |

| | | | |
|---|---|---|---|
| Antarctic Bottom Water collapse (AABW) (Sect. 3.3) | CA: **Antarctic ice melt** (+, No-Part). Wind changes, heat flux (?) <br><br> **Overall: Unknown (???)** | **Unknown**. Dependent on the effect of SRM on Antarctic ice melt. | **Very weak.** Poor process understanding; no dedicated studies on effect of SRM. |
| Marine Stratocumulus Collapse (MSC) (Sect. 4.1) | DC: **GHG forcing** (+, No), **Atmospheric warming** (+, Eff). <br><br> **Overall: Partial compensation (???)** | **Partially effective.** SRM could reverse warming and might reverse tipping point, but not for extremely high GHG forcing. | **Very weak** - This tipping point and SAI's effects on it are largely hypothetical. |
| Amazon Rainforest Dieback (AR) (Sect. 5.2) | DC: **Drought (+, Worse-Eff)**, Atmospheric warming (+, Eff), Precipitation loss (+, Worse-Eff), vapour pressure deficit (+, Part-Eff), CA/NC: **Fire (+, Worse-Part; No for human-caused wildfires)** NC: **deforestation/degradation (+,No)** <br><br> **Overall: No-Partial compensation (???)** with regional heterogeneity. In West Amazon, **overall Worsening-Partial compensation (???)**, however this is less significant for regional tipping than the East Amazon. | **Unknown, but likely ineffective.** Likely heterogenous impacts, and dependent on the very uncertain impacts of SRM on the tipping microclimate. | **Weak**. Weak process understanding, and many relevant processes sub-grid scale so poorly captured in ESMs. It may be highly dependent on deployment scheme. |

| | | | |
|---|---|---|---|
| Shallow Sea Tropical Coral Reefs loss (TCR)<br><br>(Sect. 5.3) | DC: **Surface ocean warming** (+, Eff), storm intensity (+, Part),<br>CA: ocean water acidity (+, Worse-No), disease spread (+, No-Unk)<br>NC: Fishing (+, No), Pollution (+, No)<br><br>**Overall:<br>Partial-Effective<br>compensation (?)** | **Likely ineffective to partially effective** with significant regional heterogeneity. After some mass mortality events, corals can reestablish themselves, whereas in other regions macroalgae establish themselves which SRM is unlikely to reverse. | **Intermediate.** Strong process understanding, although the relative importance of drivers still unclear. Very few modelling studies explicitly on the impact of SRM on corals. Some very limited experimental work on MCB. |
| Himalaya-to-Sunderbans system biodiversity loss (HTS) (Sect. 5.4) | DC: **Atmospheric warming** (+, Part-Eff), Monsoon precipitation (+/-, Unk)<br>CA: **glacier melt** (+, Part), sea level rise (+, Part)<br>NC: land-use change (+,No)<br><br>**Overall:Unknown (???)** | Uncertain, likely with significant regional heterogeneity. For example, glaciers could be restored and the ecosystems reliant on them, but in other cases (e.g. where keystone species have gone extinct) reversal may be impossible. | **Weak**. Despite some process understanding, very limited modelling of tipping dynamics or the relative importance of different factors, no explicit studies of the impact of SRM on the system as a whole. |
| Northern Boreal Forests dieback (NBF) (Sect. 5.5) | DC: **Atmospheric warming (+, Eff),** permafrost thawing (+, Eff); **Precipitation changes (+/-, Part-Over)**;<br>CA: snow cover loss (+, Part-Over), wildfires (+, Part)<br>CA: Insect outbreak (+, Part-Eff)<br><br>**Overall: Partial compensation (??)** | **Likely effective** over century timescales. Trees that shifted northward could recolonise the tipped areas, although microclimatic effects, and precipitation effects, make this uncertain. | **Weak**. Despite some process understanding and some confidence of SRM's impact on the temperature controlled mechanisms, there is a lack of any modelling of the impacts of SRM on the forests, which means understanding the impacts of the other factors are very uncertain. |

*Table 1: The Effect of SRM on Earth System Tipping Elements*

*Effect on Drivers means the effect of SRM on the drivers of tipping before the tipping point is reached*
*(Stage 1 of Fig. 2). The drivers named here are mostly the "primary drivers" listed in Lenton et al.*
*(2023), although "secondary drivers" have been added when appropriate. We follow Lenton et al.*
*(2023) in referring to Direct Climate (DC) drivers (e.g. warming), Climate-Associated (CA) drivers (eg*
*sea ice loss affecting AMOC), and Non-climate (CA) drivers (e.g. deforestation). Bolded drivers are*
*primary drivers. We indicate whether the driver impacts tipping by using + (exacerbates tipping) and -*
*(reduces tipping). We then use a letter code to assess the impact of SRM in a scenario with roughly*
*neutralised GMST, as laid out in Sect. 1.3 on these drivers. **Over**compensation (>125%), nearly*
***Eff**ective compensation (75 to 125%), **Part**ial compensation (25 to 75%), **No** compensation (-25 to*
*25%), **Worse**ning (<-25%) and **Unk**nown (no judgement can be made). These numbers are necessarily*
*imprecise 'best guesses' based on the evidence. We then use 0-3 question marks to say how large our*
*uncertainty is.*
*Reversibility means the effect of SRM on tipping once the tipping point is reached and self-perpetuating*
*feedbacks have set in, but before tipping is complete (Stage 2 of Fig. 2).*

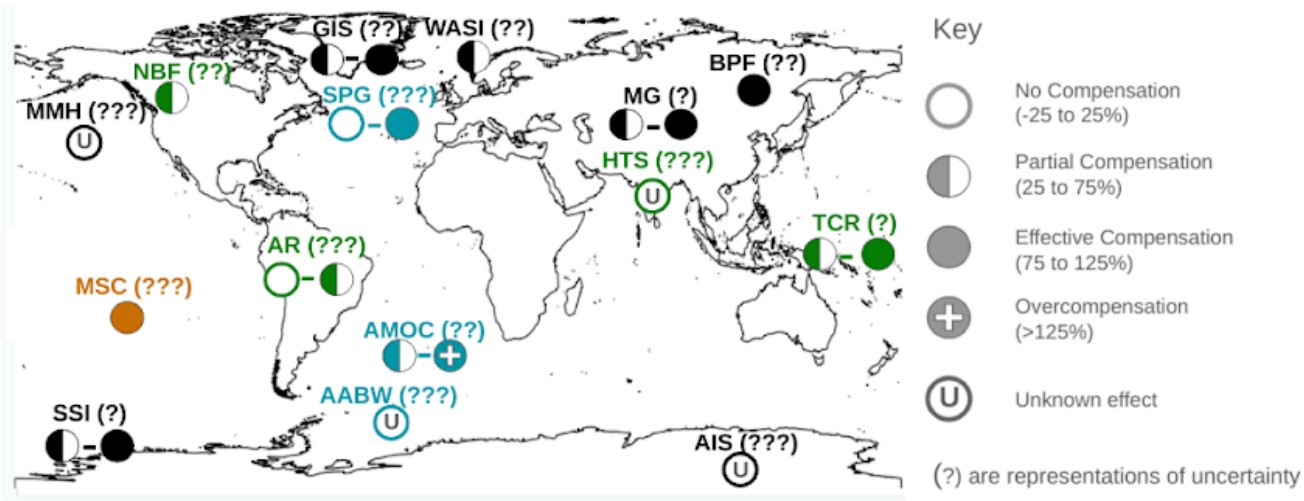

*Figure 3: The Effect of SRM on Earth System Tipping Elements*
*Abbreviations found in Table 1. We colour cryosphere elements black (AIS= Antarctic Ice Sheet, BPF=*
*Boreal Permafrost Thaw, GIS= Greenland Ice Sheet Collapse, MG= Mountain Glaciers, MMH=*
*Marine Methane Hydrates Loss at the continental shelf, SSI= Summer Sea Ice decline, WASI=Winter*
*Arctic Sea Ice abrupt loss, Sect. 2), ocean elements blue (AABW=Antarctic Bottom Water Collapse,*
*AMOC= Atlantic Meridional Overturning Circulation Collapse, SPG=Sub-Polar Gyre Collapse, Sect.*
*3), atmosphere elements brown (Marine Stratocumulus Collapse, Sect. 4) and biosphere elements green*
*(AR=Amazon Rainforest Dieback, HTS=Himalaya-to-Sunderbans system biodiversity loss,*
*NBF=Northern Boreal Forests dieback, TCR= Tropical Coral Reefs Loss, Sect. 5). The compensation*
*and uncertainty judgements is our assessment for the overall effect on drivers from Table 1.*

Out of the 15 tipping elements assessed (Table 1, Fig. 3), the available evidence suggests that SRM would probably reduce tipping drivers at least partially for 9 tipping elements. No tipping element was found to have the overall effect of SRM on its drivers exclusively worsened, although some tipping drivers were made worse and in some tipping elements (e.g. the Amazon), there may be regions where tipping risk worsens, even if it doesn't overall. For four tipping elements no judgement on the sign of SRM influence could be made due to lack of evidence. Our uncertainty was judged to be considerable to very large for 13 tipping elements. The evidence base was judged as weak or very weak for 8 of the tipping elements, and intermediate for the remaining 7; no tipping element had a strong evidence base for the impact of SRM on it. Compared to SRM's effect on drivers, its potential to reverse ongoing tipping is much harder to assess. If our (highly uncertain) findings are correct, then a well-implemented peak-shaving SAI programme would reduce the probability of tipping for most tipping elements, while using SRM to reverse tipping once it started may be much more difficult and uncertain.

# 2 Cryosphere

## 2.1 Greenland Ice Sheet Collapse

Over the past few decades, mass loss from the Greenland ice sheet has accelerated (Shepherd et al., 2012), its mass balance has become more negative (Otosaka et al., 2023) and surface elevation has also declined (Chen et al., 2021; Yang et al., 2022). This mass loss has been increasingly dominated by surface melt, which is expected to continue to be the major influence of Greenland sea level contribution over the next century (Enderlin et al., 2014; Goelzer et al., 2020). The release of freshwater from melting is also expected to slow the AMOC (Sect. 3.1), affecting global heat transfer (Golledge et al., 2019).

In the future, Greenland appears committed to significant mass loss, with the IPCC projecting the *likely range* (17-83 percentile range) of sea level contributions of between 0.01-0.1m and 0.09-0.18m by 2100 for the SSP1-2.6 and SSP5-8.5 emissions scenarios, respectively (Fox-Kemper et al., 2021). For 2300, *likely* sea level contributions are more uncertain, but range from 0.11–0.25m for SSP1-2.6 and 0.31–1.74m for SSP5-8.5. Aschwanden et al. (2019) find that the surface-elevation feedback (Sect. 2.1.1) plays a role in the persistent mass loss from Greenland, even when temperatures are stabilised at 2500. This study may overestimate surface melt rates, however, due to the assumption of spatially uniform warming. There is *limited evidence* for complete mass loss from Greenland between 1.5-3°C of sustained warming, but for 3-5°C, there is *medium confidence* in near-complete loss over several thousand years (Fox-Kemper et al., 2021). It ought to be noted that, whilst the IPCC AR6 assessment (Fox-Kemper et al. 2021) finds that the evidence for collapse under 3°C is limited, paleoclimatic data

does find evidence for past collapses in this range (Christ et al., 2021), leading to Lenton et al. (2023)
placing the critical threshold between 0.8-3$^O$C of warming.

### 2.1.1 Drivers and Feedbacks

Controls on the Greenland ice sheet are strongly driven by atmospheric temperature changes, consisting
of the interlinked surface-elevation and melt-albedo feedbacks (Levermann and Winkelmann, 2016;
Robinson et al., 2012; Tedesco et al., 2016). These feedbacks are closely linked to surface mass balance.

Surface mass balance describes the balance of accumulation and ablation on a glacier or ice sheet's
surface. Accumulation comes from snowfall, while loss is a result of melting and runoff, evaporation,
and wind driven redistribution of snow (Lenaerts et al., 2019). If ablation across a glacier or ice sheet
outweighs accumulation, surface mass balance is negative, meaning it is losing mass overall. Total mass
balance also considers mass gains and losses from ice in contact with the ocean, such as basal melt and
calving.

When a glacier or ice sheet undergoes surface melting, its elevation decreases. At lower altitudes,
surface air temperature rises (Notz, 2009), allowing more surface melting and a further decrease in
elevation (Lenton et al., 2008). At a critical threshold, this surface-elevation feedback mechanism could
continue unabated. Melting also exposes bare ice, old ice and ground, and creates melt ponds, all of
which have a lower albedo than snow. These surfaces absorb more incoming solar radiation, leading to
increased heating and more melt (Notz, 2009). This melt-albedo feedback can be exacerbated by the
presence of debris such as black carbon and dust on the ice surface, reducing albedo before melt has
even occurred (Goelles et al., 2015; Kang et al., 2020). Both of these feedbacks could, however, be
partially mitigated by post-glacial rebound. Post-glacial rebound describes the gradual rise in the Earth's
crust following glacier retreat, when the burden of the overlying ice pushing it down has been removed.
This would counteract some surface lowering, though would likely not occur on useful timescales to
alleviate the rapid mass loss if these feedbacks were triggered (Aschwanden et al., 2019).

### 2.1.2 The impacts of SRM

SRM would lower atmospheric temperatures rapidly, decreasing the amount of surface melting on the
Greenland ice sheet (Irvine et al., 2018). Irvine et al. (2009) found that even partially offsetting warming
(by decreasing the solar constant) in a 4 x $CO_2$ world would be enough to slow the sea level
contribution from the ice sheet and prevent collapse. Both (Irvine, 2012; Moore et al., 2010) found that
Greenland collapse could even be reversed if SRM strategies managed to offset the radiative forcing at a

fast enough rate. In contrast, Applegate and Keller (2015) find that while SRM can reduce the rate of mass loss from Greenland, it cannot completely stop it, and strong hysteresis prevents rapid regrowth when temperatures are reverted. Fettweis et al. (2021) also see reduced surface melt when reducing the solar constant from a high forcing to a medium forcing scenario compared with a high emissions scenario, in part due to a weakening of the melt-albedo feedback. However, this reduction is not enough to prevent negative mass balance being reached by the end of the century, and therefore a possible tipping point being crossed.

Using an energy balance model for the whole ice sheet and an ice dynamics model for the Jakobshavn Isbrae drainage basin Moore et al. (2019) estimate that Greenland mass loss is decreased by 15-20% under the G4 Geoengineering Model Intercomparison Project (GeoMIP) scenario, which involves a 5 Tg injection of $SO_2$ per year from 2020 to 2070 under an RCP4.5 scenario, compared with RCP4.5 alone. This is due to the reduction in surface melting and dynamic losses, despite a slight strengthening of the Atlantic Meridional Overturning Circulation increasing heat transfer to high latitudes under G4. Moore et al. (2023) then build on this by using two ice sheet models to also include the impact of ocean temperature and dynamic losses for the whole ice sheet. They find that the reduction in ice dynamic losses and surface melt under G4 is strongly model dependent but G4 does reduce both by an average of 35% compared with RCP4.5. Reduction is not uniform due to the topographic differences in drainage basins across the ice sheet.

Lee et al. (2023) find that SAI at 60°N is effective at reducing surface melt and runoff from the ice sheet, but impacts are not localised with cooling throughout the northern hemisphere and a southward shift of the Intertropical Convergence Zone. However, mirroring SAI in the southern hemisphere has been shown to minimise this shift (Nalam et al., 2018; Smith et al., 2022).

SAI may also result in some sulphate deposition in southern and western Greenland (Visioni et al., 2020). This would lower the albedo and could enhance the melt-albedo feedback, though the extent to which this would be negated by the decrease in temperatures and incoming solar radiation is unknown.

## 2.2 Antarctic Ice Sheet Collapse

*Likely* sea level contributions from Antarctica by 2100 range from 0.03-0.27m under SSP1-2.6, to 0.03-0.34m under SSP5-8.5 (Fox-Kemper et al., 2021). As for Greenland, there is deep uncertainty in projections to 2300, but these range from –0.14 to 0.78m and –0.27 to 3.14m without the inclusion of marine ice cliff instability (Sect. 2.2.1), for SSP1-2.6 and SSP5-8.5, respectively. Substantial melting would inject large amounts of cold freshwater into the oceans, potentially changing oceanic circulation

by inhibiting Antarctic Bottom Water formation (Li et al., 2023a; Rahmstorf, 2006), a key component in global heat transfer (Bronselaer et al., 2018). As for Greenland, between 1.5-3°C sustained warming, there is limited evidence on the complete loss of the West Antarctic Ice Sheet, but for 3-5°C, substantial or complete loss is projected for both the West Antarctic Ice Sheet (*medium confidence*) and the Wilkes Subglacial Basin in East Antarctica *(low confidence)* over several thousand years (Fox-Kemper *et al.* 2021). Similar to the Greenland Ice Sheet, Lenton *et al.* (2023) places the critical thresholds lower than the IPCC, with 1-3$^O$C for the West Antarctic Ice Sheet and 2-6$^O$C for the Wilkes Sub-Glacial Basin in East Antarctica, again partially based on paleoclimatic data.

Mass loss from Antarctica is currently driven primarily by the ocean, which melts and thins the base of ice shelves (IMBIE Team, 2020). This reduces their buttressing capabilities, which can increase ice velocities and discharge into the ocean (Gudmundsson et al., 2019). Current Antarctic air temperatures mean surface melting is limited and not a major component of direct mass loss, but it is expected to increase the likelihood of ice shelf disintegration in future (van Wessem et al., 2023).

## 2.2.1 Drivers and Feedbacks

Both the East and West Antarctic Ice Sheet are tipping elements which could be triggered due to ice sheet instabilities. The West Antarctic Ice Sheet is grounded almost completely below sea level (Morlighem et al., 2019). Many areas are situated on reverse (retrograde) bed slopes, meaning that here, the bedrock in the interior is more depressed than the coasts due to the weight of the overlying ice, and so it slopes downwards inland (Weertman, 1974).

This topography makes the West Antarctic Ice Sheet vulnerable to marine ice sheet instability (MISI), where rapid retreat and collapse could be initialised due to a destabilising of grounding lines (the area where grounded ice begins floating to become an ice shelf or calves into the ocean (Pattyn, 2018)). If grounding line retreat reaches the reverse slope of the bed, a tipping point can be initiated as continued retreat puts the grounding line in deeper waters where the ice is thicker. As the flux of ice across the grounding line is related to ice thickness, this increases ice discharge and pushes the grounding line further downslope in a positive feedback that can only be reversed if buttressing increases or the bed slope reverses (Gudmundsson, 2013; Weertman, 1974).

Parts of the East Antarctic Ice Sheet are similarly grounded below sea level with reverse bed slopes and so are also potentially vulnerable to MISI, such Wilkes and Aurora Basins, and Wilkes Land, with the latter being the main region of mass loss in the East Antarctic Ice Sheet (Rignot et al., 2019).

The major driver of MISI is ocean thermal forcing, e.g. from the upwelling of Circumpolar Deep Water. This water mass can be more than 4°C warmer than the freezing point and is driving basal melting in the Amundsen Sea Embayment (Jacobs et al., 2011). CDW upwelling is wind driven, and may have been influenced by anthropogenic climate change, though this process is poorly understood (Dotto et al., 2019; Holland et al., 2019).

MISI is thought to be a key driver of possible collapse above 2°C and 3°C atmospheric warming for the West and East Antarctic ice sheets, respectively (Garbe et al., 2020; Golledge et al., 2015; Lipscomb et al., 2021; Pattyn, 2018). The IPCC (Fox-Kemper et al., 2021) states that "the observed evolution of the ASE glaciers is compatible with, but not unequivocally indicating an ongoing MISI" (Fox-Kemper et al., 2021).

Another, more uncertain tipping process that could push both the East and West Antarctic Ice Sheets into unstable retreat is marine ice cliff instability (MICI). The MICI theory posits that ice shelves with ice cliffs taller than ~100m are theoretically unstable due to the stress of the overlying ice exceeding the ice yield strength (Bassis and Walker, 2011). Therefore, if ice shelf disintegration produces cliffs of this height, it may potentially trigger a self-sustained collapse and retreat of the grounding line (Pollard et al., 2015).

MICI has never been observed, with only indirect palaeo evidence (e.g. (Wise et al., 2017), and is a highly uncertain process (Edwards et al., 2019). Rates and duration of this self-sustained collapse are poorly known. The IPCC (Fox-Kemper et al., 2021) states that there is *low confidence* in simulating MICI. Models that invoke MICI processes present higher sea level rise projections than most other studies (DeConto *et al.*, 2021). Under 2°C warming, (DeConto et al., 2021) project the rate of mass loss to 2100 as similar to present day, but at 3°C, this jumps by an order of magnitude, increasing further for more fossil fuel intensive scenarios

MICI's drivers are similar to MISI, as both can be preceded by ice shelf disintegration from ocean thermal forcing. Atmospheric temperatures can also influence ice shelf collapse through hydrofracture (Trusel et al., 2015; van Wessem et al., 2023).

## 2.2.2 The impacts of SRM

There are few studies which focus on the impact of SRM on the East or West Antarctic Ice Sheet, but there is evidence to suggest that it would cool surface air temperatures around Antarctica (Visioni et al., 2021), which may limit hydrofracturing. SRM may be more limited in its ability to prevent Antarctic tipping points, however, as the ocean takes decades to centuries to respond to a change in atmospheric

forcing. This is seen by (Sutter et al., 2023) who find that committed Southern Ocean warming means that under RCP4.5, SRM would have to be deployed by mid century to delay or prevent a West Antarctic Ice Sheet collapse. Under RCP8.5, however, SRM cannot prevent collapse. Hysteresis experiments find that regrowth occurs much more slowly than mass loss (Garbe et al., 2020). DeConto et al. (2021) and Garbe et al. (2020) show that the ocean's slow response to atmospheric thermal changes means that while implementing Carbon Dioxide Removal (CDR, which may have a somewhat similar thermal effect to SRM) in the first half of this century could reduce sea level rise compared to a 3°C warming scenario it cannot reverse it. SRM may also be less effective at cooling the poles than the tropics as during the polar night where there is limited or no solar radiation, it would have no effect (McCusker et al., 2012).

(McCusker et al., 2015) suggest that sulphate SAI induced stratospheric heating would intensify and shift southern hemisphere surface winds poleward, increasing CDW upwelling and therefore basal melting. This finding, however, may be injection strategy dependent as injection of a different aerosol may not cause the stratospheric heating observed (Keith et al., 2016). In addition, the poleward shift seen from tropical injection location (McCusker et al., 2015) is not seen for a southern hemisphere injection where the jet shifts equatorward (Bednarz et al., 2022); (Goddard et al., 2023). Goddard et al., (2023) also find that, while the Antarctic response to SRM is strongly dependent on injection strategy, multi-latitude sulphate SAI injection that limits global warming to 0.5°C above preindustrial could prevent possible collapse of much of the Antarctic ice sheet.

In summary, SRM would therefore likely be effective in reducing surface melting and hydrofracturing, but it would not be as effective at reducing basal melt. For sulphate SAI in particular, it is unclear how the resultant stratospheric heating will affect atmosphere and ocean circulation, and therefore also CDW upwelling. In addition, a reduction in atmospheric temperatures would reduce the moisture-holding capabilities of the air, decreasing the amount of precipitation falling as snow on Antarctica. Mid latitude SAI itself would also dampen the hydrological cycle and suppress precipitation (Irvine et al., 2018; Tilmes et al., 2013; Visioni et al., 2021). Therefore, if SRM's effect on reducing basal melt is limited, while simultaneously decreasing snowfall accumulating on Antarctica, it is also possible that it could be more harmful to Antarctica than doing nothing at all: in a warmer, non-SRM world, increasing precipitation may slightly offset some mass loss (Edwards et al., 2021; Stokes et al., 2022).

## 2.3 Mountain Glacier Loss

Current trends of glacier mass balance globally are negative, with glacier mass loss accounting for ~40% of current observed sea level rise from 1901-2018 (Rounce et al., 2023; Zemp et al., 2019).

(Zemp et al., 2019) also show that if present rates of mass loss were sustained, Western Canada, the USA, central Europe and low latitude glaciers would lose almost all mass by 2100. The glaciers in high mountains of Asia are projected to lose their total mass by 60-70% by the end of the century under the RCP8.5 scenario and by 30-40% even if global warming is limited to 1.5°C (Kraaijenbrink et al., 2017). Most glaciers are not in equilibrium with the current climate and so are still responding to past temperature changes. Therefore, it is projected that they will continue to experience substantial mass loss through the 21[st] century, regardless of which emissions scenario is followed (Marzeion et al., 2018, 2020; Zekollari et al., 2019). Sustained warming of 1.5-3°C  is projected to result in glacier mass loss of 40-60%, increasing up to 75% for 3-5°C (*low confidence*, Fox-Kemper e*t al.*, 2021).

### 2.3.1 Drivers and Feedbacks

Mountain glaciers are, like the Greenland ice sheet, subject to the surface-elevation and melt-albedo feedbacks which could lead to unabated retreat (Johnson and Rupper, 2020), but due to their smaller size, they are more sensitive to climatic changes and respond on shorter timescales. They are also affected by additional local drivers and feedbacks such as changing snow patterns and slope instabilities. These local feedbacks are not discussed here as we are focused on the global scale processes affecting mountain glaciers more generally.

(Rounce et al., 2023) see that mass loss in larger glaciated areas is linearly related to global temperature, but that smaller regions are much more sensitive to warming, leading to a non-linear relationship above 3°C.

### 2.3.2 The impacts of SRM

Each individual glacier has its own topographical and climatological conditions affecting mass balance and it is unlikely that SRM would have a uniform effect. Reducing temperatures using SRM would be more effective for low latitude glaciers where an increased proportion of the energy flux is shortwave (Irvine et al., 2018). Zhao et al. (2017) find that though SRM can limit mass loss from all glaciers in high mountain Asia by 2069, retreat is still observed due to their slow response times to temperature changes. Under the G3 and G4 scenarios, glacier area losses in 2089 are 47% and 59% of their 2010 areas, respectively, compared with 73% under RCP4.5. G3 involves a gradual increase in the amount of $SO_2$ injected to keep global average temperature nearly constant at (projected) 2020 levels under an RCP4.5 scenario (Kravitz et al., 2011).

SRM counteracts hydrological changes to different extents (both on a global and, more pertinently, regional level) to how it counteracts temperature change (Ricke et al., 2023), so while melt may be reduced, surface mass balance could be decreased overall through reduced snowfall in the accumulation zone. Idealised experiments using a reduction of the solar constant to halve the warming resulting from doubled $CO_2$ indicate that negligible amounts of the planet would see substantially reduced precipitation compared to preindustrial (Irvine et al., 2019), but precipitation changes from SRM specifically are unlikely to be uniform. (Zhao et al., 2017) highlight that, for Himalayan glaciers, this precipitation decrease may be much less important compared with whether the precipitation is falling as snowfall in the accumulation zone or as rainfall, in which case SRM-induced cooling might prove valuable. Outside of the Himalayan region, there is a lack of research on precipitation impacts.

## 2.4 Land Ice Further Research

Currently, there are large gaps in the literature and high model uncertainty with regards to how SRM will affect land ice, particularly Antarctica. There is a need for multi-model ensembles forced by various SRM scenarios, including aerosols other than sulphate and methods other than SAI. As suggested in Irvine, Keith and Moore (2018), the inclusion of GeoMIP scenarios in the Ice Sheet (Nowicki et al., 2016) and Glacier (Hock et al., 2019) Modelling Intercomparison Projects (ISMIP and GlacierMIP, respectively) would allow direct comparisons with standard emission scenarios.

The GeoMIP SAI scenarios are fairly simplistic as they prescribe only an equatorial injection and do not take into account the equator-to-pole temperature gradient. As SRM impacts the polar regions differently compared with the rest of the globe, targeted SRM injection at specific latitudes could be more effective, though it could yield different results depending on location. For example, (Bednarz et al., 2022) find that a northern hemisphere SAI injection with sulphate drives a positive southern annular mode, whereas southern hemisphere injection results in a negative southern annular mode response. This area therefore requires more research. Running ice sheet and glacier model ensembles forced by the Geoengineering Large Ensemble project (GLENS, (Tilmes et al., 2018)) simulations would aid further exploration of the effects of targeted SAI, as these experiments inject at 30°N, 30°S, 15°N and 15°S. Seasonal SAI has also been shown to be more effective for Arctic sea ice than year round injection (Lee et al., 2021): expanding this to land ice would also be an important avenue for future research.

## 2.5 Sea Ice

Sea ice is frozen seawater, typically 10s of cm to several metres thick, and at any one time covers around 7% of the earth's surface, although this coverage is decreasing at around 10% per decade (Fetterer, 2017). The annual Arctic sea-ice minimum extent has declined by 50% since satellite observations began in the late 1970s (Fetterer, 2017). The Arctic is expected to be seasonally ice-free by mid-century; a majority of CMIP6 models have ice-free periods during the Arctic summer by 2050 under all plausible emissions scenarios (Notz and SIMIP Community, 2020). CMIP6 models project a decline in Winter sea ice which is linear in both cumulative $CO_2$ and warming (Notz and SIMIP Community, 2020).

Despite substantial warming, there was a slight increasing trend in Antarctic sea ice through the observational record until around 2014 (Parkinson, 2019), likely due to natural variability (Meehl et al., 2016). However, in recent years, a series of low sea-ice extents have occurred; Antarctic sea ice was at the lowest extent on record in 2022, only to be surpassed by a new record low in February 2023 (Fetterer, 2017). Projections of Antarctic sea ice response to climate change have lower confidence than for the Arctic, due to poorer model representation (Masson-Delmotte *et al.*, 2021). CMIP6 models predict a decline over the 21[st] Century of 29-90% in summer and 15-50% in Winter, depending on the emissions scenario (Roach et al., 2020).

### 2.5.1 Drivers and Feedbacks

On decadal time-scales, Arctic sea-ice area has declined linearly with the increase in global mean temperature over the satellite period in all months (Notz and Stroeve, 2018). Local radiative balance at the sea-ice edge may also be an important control on Arctic sea ice extent (Notz and Stroeve, 2016), and large scale modes of atmospheric variability, such as the Arctic Oscillation, also contribute strongly to interannual variability (Mallett et al., 2021; Stroeve et al., 2011). Unlike in the Arctic, almost all of the Antarctic sea ice is seasonal, disappearing each summer. Wind patterns, modulated by large scale modes of atmospheric circulation such as the Southern Annular Mode, are a key driver of Antarctic sea ice extent on inter-annual to decadal timescales (Masson-Delmotte et al., 2021).

Sea ice under global warming is subject to the ice albedo feedback (Serreze et al., 2009), whereby the loss and thinning of sea ice reduces the surface albedo so increases the absorption of solar radiation, leading to additional warming, and further sea-ice loss. As a result, it has been posited that sea ice loss could be subject to tipping points (Merryfield et al., 2008; North, 1984). However, there are also stabilising feedbacks. Open ocean during the polar night can rapidly vent heat to the atmosphere (e.g.

Serreze et al., 2007), thin ice grows faster than thick ice (Bitz and Roe, 2004), and later forming ice has
a thinner layer of insulating snow cover on entering the winter months and so can grow more quickly
(Hezel et al., 2012; Notz and Stroeve, 2018)

These mechanisms likely prevent tipping-point behaviour from arising for summer Arctic sea ice; GCM
simulations find that arctic sea ice is expected to recover to an equilibrium state associated with the
large scale climate forcing within 1-2 years of complete removal (Tietsche et al., 2011), and the
observed time-series of summer sea-ice extent has a negative 1-year lag autocorrelation, that is, years
with low summer sea-ice extent are typically followed by years with above average extent and vice
versa (Notz and Stroeve, 2018). Both satellite observations (Notz and Marotzke, 2012; Notz and
Stroeve, 2018) and modelling studies (Tietsche et al., 2011) concur that the stabilising feedbacks
outweigh the destabilising ice-albedo feedback to mean that summer sea ice loss is not
self-perpetuating, such that the overall sea ice-extent is expected to remain tightly coupled to the
external driver, i.e., temperature rise, throughout its decline (Stroeve and Notz, 2015). For Winter Arctic
sea ice, there is a potential for abrupt areal loss at a threshold warming (Bathiany et al., 2016). This is
because once the arctic is seasonally ice free, sea ice coverage drops to zero wherever the ocean is too
warm to form sea ice in a given year, and if warming is spatially uniform, this transition can happen
rapidly over a large area at a threshold warming level (Bathiany et al., 2016). Local positive feedback
processes may also contribute to the abrupt winter Arctic sea-ice loss seen in some models (Hankel and
Tziperman, 2021).

**2.5.2 The impacts of SRM**

There is broad agreement across models that SRM would cool both the Arctic and Antarctic (Berdahl et
al., 2014; Visioni et al., 2021)**.** As expected given this cooling, various models have shown a reduced
loss of both Arctic (Jiang et al., 2019b; Jones et al., 2018; Lee et al., 2020, 2021) and Antarctic (Jiang et
al., 2019b; McCusker et al., 2015) sea ice under SRM. Under the GeoMIP scenarios G3 and G4, SAI
delays the loss of sea ice but this is not sufficient to prevent the loss of almost all September sea ice in
most models (Berdahl et al., 2014). However, it is likely that this is due to insufficient cooling, and that
a world at the same global mean temperature without SRM would also lose all September sea ice in
these models (Duffey et al., 2023).

Under equatorial or globally uniform injection, SRM likely cools the Arctic less strongly than the global
mean and thus results in greater arctic amplification, and loss of Arctic sea ice at a given global mean
temperature (Ridley and Blockley, 2018). This effect is reduced with greater injection in the mid and
high latitudes. For example, the Geoengineering Large Ensemble simulations in CESM (Tilmes et al.,

2018), which use injection at multiple latitudes to hold global temperature at its 2020 value, while also controlling the meridional temperature gradient, show a 50% increase in Arctic September sea-ice extent relative to present day (Jiang et al., 2019b). Similarly, several studies have modelled SAI with high latitude injection and found that such strategies can effectively halt declines in Arctic sea ice under high emissions scenarios (Jackson et al., 2015; Lee et al., 2021, 2023), potentially more efficiently per unit $SO_2$ injection than low latitude injection strategies (Lee et al., 2023).

Winter arctic sea ice is restored less effectively than summer sea ice in modelling of SRM scenarios (Berdahl et al., 2014; Jiang et al., 2019b; Lee et al., 2021, 2023). For example, one SRM scenario sees 50% more sea-ice extent at the September minimum than the control case (at the same global mean temperature without SRM), but 8% less extent at the March maximum (Jiang et al., 2019b). This is linked to a general under-cooling of the polar winter by SRM, and an associated suppression of the seasonal cycle at high latitudes (Jiang et al., 2019b; Duffey et al., 2023). However, modelling of SRM shows at least partial effectiveness at increasing winter sea ice and reducing local winter near-surface air temperatures relative to the same emissions pathway without SRM (Berdahl et al., 2014; Jiang et al., 2019b; Lee et al., 2021, 2023). As such, it is likely that SRM would decrease the probability of passing any potential thresholds to more abrupt winter Arctic sea-ice decline.

The literature on Antarctic sea-ice response to SRM is more limited than for the Arctic case. The modelling of volcanic eruptions suggests an asymmetric response to hemispherically symmetric aerosol forcings, with Antarctic sea ice extent increasing much more weakly than Arctic under volcanic cooling (Pauling et al., 2021; Zanchettin et al., 2014). A similar result is found in the Geoengineering Large Ensemble simulations in CESM (Tilmes et al., 2018, Jiang et al., 2019b). Antarctic sea ice is less well preserved than Arctic sea ice under this SRM simulation, particularly in austral winter, with a 23% reduction in maximum extent relative to the baseline. However, while several modelling studies show only incomplete preservation of Antarctic sea ice under SRM relative to the target world, in all cases the extent of sea ice is increased relative to the warmer world without SRM (Jiang et al., 2019b; Kravitz et al., 2013; McCusker et al., 2015).

Sea-ice loss is expected to be reversible were temperatures to reduce (Ridley et al., 2012; Tietsche et al., 2011). As such, we would expect sufficient SRM cooling to be capable of restoring sea ice after the onset of ice-free conditions.

### 2.5.3 Further Research

There has been little study of the impact of SRM on Antarctic sea ice. Given the potential hemispheric asymmetry in response to aerosol forcing discussed above, and in the context of concerns over the ability of SRM to arrest Antarctic change (Sect. 2.2), this is an important research gap. Additionally, there has been little work- (Ridley and Blockley, 2018) is a notable exception - assessing the different impact of SRM versus avoided emissions on Arctic and Antarctic climate and sea ice under SRM, at a given global mean temperature. Such assessments would aid in making a fully quantitative statement on the effectiveness of SRM strategies for sea-ice restoration (Duffey et al., 2023).

## 2.6 Permafrost

Permafrost is perennially frozen soil which stores around 1500 GtC in the form of organic matter, roughly twice as much carbon as is found in the atmosphere (Meredith et al., 2019). As the earth warms, permafrost thaws and subsequent decomposition of thawed organic matter releases $CO_2$ and methane, further warming the planet. As such, permafrost thaw is a positive feedback on global temperature, known as the permafrost carbon feedback. The permafrost carbon feedback is estimated to add-roughly 0.05 °C per °C to global temperature increase (Schuur et al., 2015). The strength of the permafrost carbon feedback depends, not only on the reduction in permafrost, but also on the proportion of carbon emissions released as $CO_2$ versus methane, and on the degree of offsetting by increased plant biomass in current permafrost regions (Wang et al., 2023).

Over the 21[st] century, greenhouse gas emissions from thawing permafrost are expected to be similar in magnitude to those of a medium sized industrial country, with estimates from ESMs putting emissions at order of magnitude 10 $GtCO_2$e per °C global warming by 2100 (Masson-Delmotte et al., 2021). For a rapid decarbonisation scenario limiting warming to under 2°C by 2100, permafrost GHG emissions are expected to use up perhaps 10% of the remaining emissions budget (Comyn-Platt et al., 2018; Gasser et al., 2018; MacDougall et al., 2015).

### 2.6.1 Drivers and Feedbacks

Gradual permafrost thaw occurs due to vertical thickening of the active layer in response to warming at rates of centimetres per decade (Grosse et al., 2011; Turetsky et al., 2020). However, locally, permafrost is also subject to abrupt thaw, which refers to deep thaw occurring on rapid timescales of days to several years due to processes such as the physical collapse of the surface caused by ice melt and the formation of thermokarst lakes (Schuur et al., 2015; Turetsky et al., 2020). Such abrupt thaw may increase the

strength of the permafrost carbon feedback substantially relative to that modelled in ESMs, which do not include these processes. For example, Turetsky et al. (2020) report an increase in estimated permafrost carbon release by 40% and an increase in global warming potential by 100% when abrupt thaw is taken into account in addition to gradual thaw by active layer thickening.

Soil temperature is the fundamental control on permafrost thaw, and this in turn is principally controlled by annual mean near-surface air temperature (Burke et al., 2020; Chadburn et al., 2017). Earth system models predict an approximately linear decline in permafrost area with air temperature increase over the current permafrost regions (Slater and Lawrence, 2013). Various other factors also impact soil temperature however, including vegetation cover, precipitation type and amount, and wildfire (Grosse et al., 2011). For example, summer rainfall fluxes sensible heat into the soil, increasing thaw (Douglas et al., 2020), and snow cover over winter insulates the soil, increasing its annual mean temperature (Zhang et al., 1997).

Armstrong McKay et al. (2022) suggest with low confidence a potential threshold behaviour at >4°C global warming or 9°C of local warming for near-synchronous and rapid thaw of large areas of permafrost, particularly Yedoma deposits (Strauss et al., 2017), driven by an additional local positive feedback on thawing due to heat production from microbial metabolism. The self-accelerating permafrost thaw driven by this additional feedback is driven in part by large local rates of warming (Luke and Cox, 2011). Others, however, have suggested that no such global mean temperature threshold applies, with global permafrost loss being quasi-linear in global warming throughout its decline (Nitzbon et al., 2024). If a global temperature threshold at 4°C exists, Armstrong McKay *et al.* (2022) estimate that passing it might lead to a pulse of one-off GHG emissions over 10-300 years equivalent to a rise in global mean temperature of 0.2-0.4 °C. This potential global tipping element is in addition to the occurrence of localised abrupt thaw which becomes more widespread at warming above approximately 1.5°C (Armstrong McKay et al., 2022).

Considering the total land carbon feedback, rather than just the permafrost carbon feedback, the increase in net primary productivity in current permafrost regions will offset at least some of the loss of permafrost carbon over this century (Schuur et al., 2022). Some simulations even show the permafrost regions as net carbon sinks under warming, due to warming and $CO_2$ fertilisation increasing the productivity of vegetation (McGuire et al., 2018).

## 2.6.2 The impacts of SRM

There is good inter-model agreement that SRM would reduce mean annual air temperature over the permafrost regions (Berdahl et al., 2014; Visioni et al., 2021), so we expect it to reduce permafrost thaw relative to warming scenarios without SRM. Modelling studies support this expectation; only a handful of modelling studies have assessed the permafrost response to SRM, but all find reduced loss of permafrost carbon with deployment of SRM (Chen et al., 2020, 2023; Jiang et al., 2019b; Lee et al., 2019, 2023; Liu et al., 2023).

The inter-model spread in permafrost projections is large and can be larger than the difference between SRM and non-SRM scenarios (Chen et al., 2020), so multi-model assessments are desirable. Three studies have assessed the permafrost response to SRM in a multi-model context using the GeoMIP simulations (Chen et al., 2020, 2023; Liu et al., 2023). These studies show that SRM avoids a large fraction of the permafrost loss projected under warming scenarios without SRM. For example, using equatorial SAI to bring global temperatures in line with a medium emissions scenario (SSP2-4.5) under a high emissions scenario (SSP5-8.5) is modelled to mitigate most (>80%) of the extra permafrost carbon loss associated with the high emissions scenario (Chen et al., 2023).

However, global SRM strategies typically under-restore permafrost relative to their impact on global mean temperature because they see residual warming in the permafrost regions (Chen et al., 2020, 2023). It is likely that SRM strategies targeted at restoring polar climate, by injecting more aerosols outside of the tropics, could largely avoid this effect. For example, almost all the 21$^{st}$ century permafrost loss under the high emissions scenario RCP8.5 is avoided under an SAI scenario which modifies injections to target the equator to pole gradient, as well as global mean temperature (Jiang et al., 2019b)

While there has been no modelling study assessing the potential for SRM to avert the widespread and rapid decline envisioned under the permafrost 'collapse' scenario of Amstrong-McKay *et al.* (2022), the fundamental driver of this tipping behaviour is surface temperature, and as such, we expect that reducing local temperatures using SRM would reduce the likelihood of this scenario. However, as it is driven by internal heat production, it seems unlikely that SRM could substantially help reverse tipping once this 'collapse' scenario had begun, were the near-synchronous onset across a large part of the permafrost regions, assumed by Amstrong-McKay et al. (2022), to take place. Similarly, while SRM might reduce the onset of localised abrupt thaw processes, it would be unlikely to reverse these processes once begun.

Emissions from thawed permafrost are irreversible on centennial timescales (Schaefer et al., 2014; Schuur et al., 2022). SRM would not be able to reverse the increased atmospheric GHG concentrations once permafrost thawing had occurred.

### 2.6.3 Further Research

The permafrost response in ESMs does not include the feedback processes leading to abrupt thaw and local tipping behaviour (Turetsky et al., 2020), so the quantitative assessments above principally apply to the gradual thaw component; further development of ESMs to include such processes would allow more robust quantitative assessment of the impact of SRM (Lee et al., 2023). Additionally, the broader study of the high latitude land carbon feedback under SRM would benefit from the attention of scientists from a range of backgrounds, including soil science and ecology, to quantify the impact of simultaneous changes in temperature, hydrology and $CO_2$ concentration expected under SRM.

Greater understanding is also required of the degree and cause of under-cooling of Northern Hemisphere high latitudes under SRM, and the dependence of such under-cooling on the injection strategy. This would facilitate quantification of the expected permafrost carbon feedback under different SRM strategies.

## 2.7 Marine Methane Hydrates Release

Marine methane hydrates are methane trapped in water ice in sea floor sediments. These hydrates contain a large amount (1000s of GtC) of methane and are vulnerable to melt over millennia given several degrees of ocean warming, and so represent a positive climate feedback that may have contributed to past warming events on geological timescales (Archer et al., 2009). However, globally significant methane emissions from hydrates on decadal or centennial timescales are very unlikely (Masson-Delmotte *et al.*, 2021; Schuur *et al.*, 2022). There is no expected threshold warming level associated with methane hydrates as a whole and thus they are typically considered a threshold-free feedback rather than tipping element (Armstrong McKay *et al.*, 2022) and at moderate warming levels (e.g. 2°C) they likely exert a negligible impact on surface temperature (Wang *et al.*, 2023).

## 2.7.1 The impacts of SRM

There is no literature which we are aware of which evaluates the impact of SRM on methane hydrates.
The reduction in surface temperature under SRM, if maintained over the multi-centennial timescale of
deep-ocean heat uptake, might be expected to reduce ocean-floor temperatures and thus the rate of melt.
However in the curve-flattening scenarios without SRM (i.e. an overshoot scenario), the overshoot may
not be long enough (MacMartin et al., 2018) for its impacts to be felt by the methane hydrates in the
deep ocean (Ruppel and Kessler, 2017), meaning SRM may have little benefit over such scenarios.
Moreover, there is no consensus yet amongst models on the large-scale ocean circulation response to
SRM (Fasullo and Richter, 2023).

# 3. Oceans

This section treats three possible tipping elements, all part of the Atlantic and Southern Ocean
circulation (see Fig. 4): The Atlantic Meridional Overturning Circulation (AMOC; Fig. 4 process 1-4),
deep convection in the north Atlantic Subpolar Gyre (SPG, Fig. 4 process 3), and Antarctic Bottom
Water formation (Fig. 4 process 5-6).

669 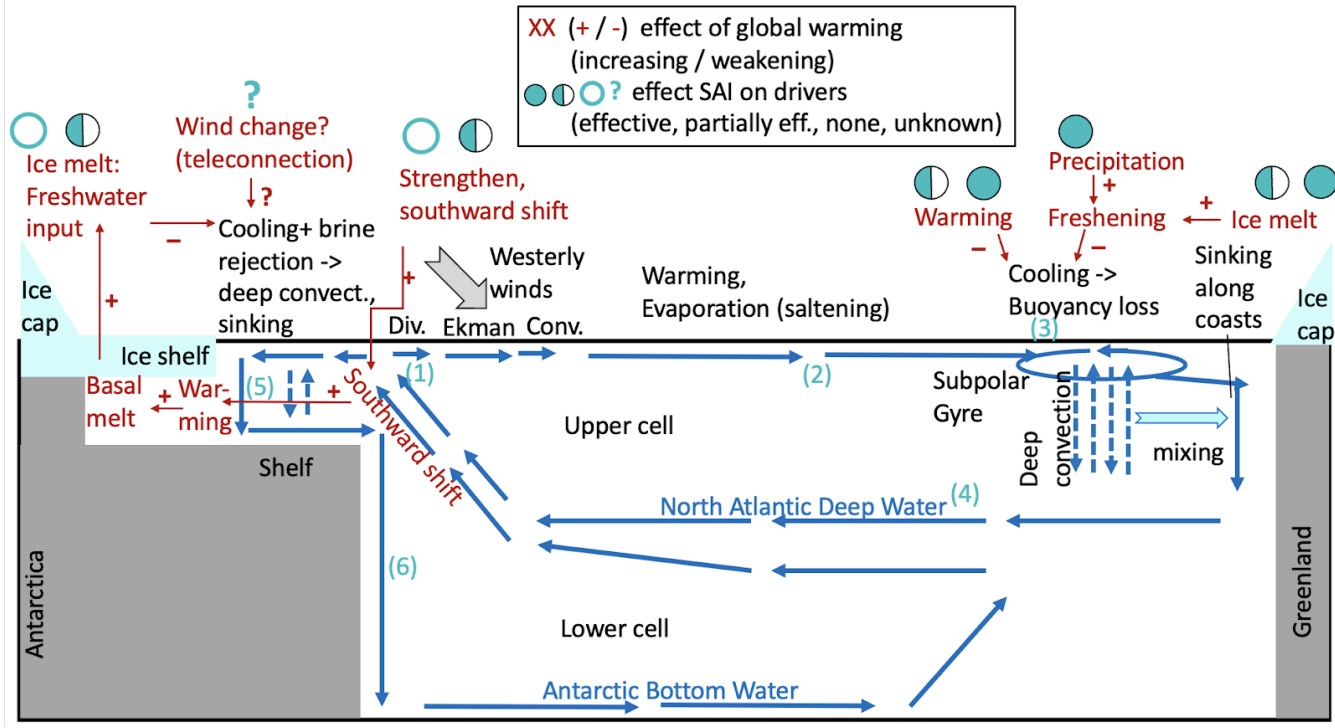

*Figure 4: Schematic of the Atlantic circulation. (1) Westerly winds around 40ºS drive a northward Ekman transport, south of which divergence enables the upwelling of North Atlantic Deep water. (2) To the north, water moves northwards, warming and saltening through evaporation. (3) In the subpolar gyre, water moves counterclockwise, aided by the cold core of the gyre and thermal wind effects. Winter cooling drives deep convection, thereby cooling the water inside the gyre over great depths. Cold water mixed into coastal currents (e.g. along Greenland) helps to drive sinking there. (4) The resulting North Atlantic Deep Water returns to the South. (5) Very dense Antarctic Bottom Water (AABW) is formed in sea-ice-free stretches around Antarctica, where water is exposed to cold air and salinification through brine rejection. It sinks along the shelf edge (6) and feeds the lower circulation cell. Global warming may warm and freshen surface water in the North Atlantic, reducing deep convection and weakening the Atlantic Meridional Overturning Circulation and the Subpolar Gyre (3); SRM is likely partially effective to effective. In the South, global warming can affect Antarctic meltwater input by increasing the upwelling of warm water onto the shelf, hindering densification and hence Antarctic Bottom Water formation (5). SRM is likely not fully effective (Sect. 3.3). The effect of other drivers, e.g. wind change, on AABW formation is uncertain.*

## 3.1 Atlantic Meridional Overturning Circulation (AMOC) Collapse

The upper branch of the Atlantic Meridional Overturning Circulation (AMOC) transports salty, warm water towards the subpolar North Atlantic, where it sinks and returns to the south (Fig. 4). In order to sink, this water must be sufficiently dense compared with the deeper water, therefore surface warming or freshening inhibits sinking. North-Atlantic sinking is at least partly compensated by water rising in the Southern Ocean, due to an interplay of Ekman-driven upwelling and eddy flow (Johnson et al., 2019; Marshall and Speer, 2012).

Climate models project AMOC to weaken under global warming, but in general models do not predict collapse for SSP scenarios extending to 2100 (Weijer et al., 2020), although some models show collapse for extreme hosing (Jackson et al., 2023; van Westen and Dijkstra, 2023) or warming (Hu et al., 2013). Climate models might underestimate AMOC stability, and whether AMOC actually can tip (collapse) under present conditions is still an open debate (see SI). Note that a prolonged quasi-stable shutdown or strong reduction in AMOC strength could have severe climate impacts lasting for decades or more (Fig. 4 of Loriani et al., 2023), even without actual tipping.

### 3.1.1 Drivers and Feedbacks

In the North Atlantic, global warming could cause buoyancy forcing, i.e. reduce surface water density (and hence weaken and potentially tip AMOC) through surface warming and freshening. Freshening could stem from an increase in precipitation minus evaporation, sea ice melt, or meltwater flux from Greenland melting.

Gregory et al. (2016) found that for forcings derived from doubling $CO_2$ gradually over 70 years (1pctCO2), only heat flux changes lead to significant AMOC weakening, whereas freshwater flux other than ice sheet runoff has no significant impact. However, Madan et al. (2023) suggests that for instantaneous $CO_2$ quadrupling in CMIP6, freshwater forcing from sea ice melt weakens AMOC. Liu et al. (2019) also suggested that changes in sea ice cover may impact AMOC through changes in freshwater input (freezing, advection and melting of ice floes) and heat flux (e.g., shielding ocean water from atmospheric influences); they find that sea ice retreat eventually weakens AMOC. Using an intermediate complexity model, Golledge et al. (2019) found that future freshwater fluxes from Greenland (and Antarctica) derived from ice sheet models under RCP8.5 forcing might weaken AMOC by 3-4Sv. If AMOC can indeed tip, then icemelt would likely increase the probability. Atmospheric circulation changes, e.g. North Atlantic Oscillation (NAO), may also affect AMOC, for example by introducing heat flux anomalies (Delworth and Zeng, 2016).

In the Southern Ocean, climate change might influence the position or strength of the westerly winds potentially affecting AMOC's upwelling branch. However, changes in eddy fluxes might (partly) compensate for the change in westerlies (Marshall and Speer, 2012).

It is uncertain if tipping into an off-state can be reached with climate forcings that can occur under anthropogenic global warming. If so, buoyancy forcing, either from heat flux changes or freshwater changes, is likely the key driver, as is the case for AMOC weakening.

Whilst the classic view is that a gradual change in forcing would eventually tip AMOC (Fig. 1a), random fluctuations in buoyancy forcing might push AMOC into the off-state even if the tipping point is not reached ("noise-induced tipping", Fig. 1c; Ditlevsen and Johnsen, 2010). In addition, it has been suggested that fast changes in the buoyancy forcing may lead to rate-induced tipping (Fig. 1d; Lohmann and Ditlevsen, 2021).

### 3.1.2 The impacts of SRM
SRM is likely to reduce most drivers of AMOC weakening. Using GeoMIP (Kravitz et al., 2011) data, Xie et al., (2022) found that in the highly idealised G1 experiment, where the GMST effect of instantaneous quadrupling of $CO_2$ is compensated by instantaneous solar dimming, the GHG effect on

heat flux in North Atlantic deep convection regions is Partially to Effectively compensated (3 models), while the effect on precipitation minus evaporation is Effectively compensated to Overcompensated (6 models) and September sea ice loss is Effectively compensated (6 models). SRM is expected to Partially to Effectively prevent Greenland tipping (Sect. 2.1), which suggests it may reduce freshwater input from ice melt.

Several studies directly modelled the effect of SRM (or analogues) on AMOC weakening without separating the effect on various drivers. Hassan et al. (2021) showed that anthropogenic aerosols, in absence of Greenhouse forcing, increased AMOC by about 1.5Sv in the 1990s, with surface heat flux dominating over freshwater flux. Xie et al. (2022) used simulations of various SRM methods, including SAI, solar dimming, increasing ocean albedo (a rough proxy for Marine Cloud Brightening (MCB) or for placing reflective foam on the water), and increasing cloud droplet number concentration (a simple representation of MCB), and the strength varies from a modest reduction to complete elimination of greenhouse-gas-induced warming. They found that in all cases, SRM reduces GHG-induced AMOC weakening. If global mean surface temperature change is fully compensated (experiment G1), AMOC strength is Effectively restored in the multi-model mean, with solar dimming performing slightly better and MCB slightly worse than SAI. Note that in G1 there is no period of global warming, as solar dimming starts simultaneously with CO2 increase, while in reality, AMOC changes may be locked in before SRM starts. Using the CESM2-WACCM model, Tilmes et al. (2020) found that if SRM is used to cool RCP8.5 forcing back to 1.5 degrees from 2020, AMOC weakening is roughly halved compared to RCP8.5 forcing without SRM compared to year 2020. In a previous model version, AMOC weakening was even overcompensated by SRM, leading to AMOC strengthening (Fasullo et al., 2018; Tilmes et al., 2018). This suggests that SRM's overall effect on AMOC weakening is Partial compensation to Overcompensation. Given the similarity in drivers for AMOC weakening and tipping, we assess the effect of SRM on AMOC tipping to be Partial to Overcompensation, too.

The potential rate-dependency of AMOC tipping (Lohmann and Ditlevsen, 2021) may imply that strategies where SRM is used to reduce the rate of warming before being phased out may reduce the risk of tipping the AMOC. However, it also implies that termination shock may increase the risk of tipping compared to the same temperature rise without SRM. However, rate-dependent AMOC tipping remains uncertain, so the possible effects of SRM on this mechanism remain uncertain too. As for noise-induced tipping, it is unclear whether SRM would affect the amplitude of buoyancy forcing noise. However, SRM may help to keep AMOC further from the tipping point, which would reduce the susceptibility to noise-induced tipping.

It is difficult to understand to what extent SRM could restore the AMOC once tipping has begun, as no model simulations exist. An extension of sea ice cover after AMOC tipping (or weakening) may shield the ocean from surface cooling (van Westen and Dijkstra, 2023), rendering SRM less effective or potentially counterproductive. Even if SRM can restore AMOC, very strong SRM might be required if AMOC shows hysteresis, and this forcing may have to be applied for many decades, with potentially detrimental consequences. Schwinger et al. (2022) demonstrate this by simulating the effect of instantaneous CDR, and hence instant cooling, on a weakened (i.e. not even tipped) AMOC. AMOC recovered, but during the transition period, the North Atlantic region was severely overcooled, as the cooling effect of CDR already manifested itself, while AMOC was still weak. Pflüger et al. (2024) simulate an abrupt SAI onset in 2080 and find that AMOC weakening is halted, but not reverted, by 2100, leading to prolonged overcooling in the North Atlantic. Attempts to restore a tipping or fully tipped AMOC might lead to even more severe and extended overcooling. Conversely, potential attempts to minimise overcooling by slowly ramping up SRM may conflict with requirements for preventing other tipping points.

### 3.1.3 Further Research

Ongoing efforts of the AMOC research community may help to better understand AMOC instability and its susceptibility to SRM. Improving climate models may reduce biases, in particular potentially excessive AMOC stability, and hopefully eventually enable us to directly simulate SRM's impact on AMOC tipping. Meanwhile, qualitative insights on SRM's effect on potential AMOC tipping might be gained by using simulations with extreme forcings (warming and/or freshwater) which actually tip AMOC, and investigate whether SRM can postpone or revert tipping.

Another research avenue could be to chart more systematically the impact of SRM on AMOC drivers, including in the South. This requires disentangling the direct effect of SRM forcing from AMOC feedbacks (Hassan et al, 2021). Impacts on drivers likely depend on the SRM method (e.g. SAI or alternatives) and strategy (e.g. timing, intensity and location of injection points). Note that even if AMOC can not tip, SRM's impact on AMOC weakening remains an important research subject.

## 3.2 North Atlantic Sub-Polar Gyre Collapse

There are indications that deep convection in the subpolar gyre (SPG) in the North Atlantic may collapse without full AMOC collapse, although it is uncertain whether the SPG is a tipping element (see SI).

### 3.2.1 Drivers and Feedbacks

As is the case for AMOC, the main drivers are surface warming and processes leading to surface freshening. Sgubin et al. (2017) and Swingedouw et al. (2021) leaning on Born and Stocker (2014), suggest the following mechanism for SPG collapse: First, the SPG gradually freshens due to enhanced precipitation and runoff caused by intensified hydrological cycle under global warming; meltwater from Greenland could provide additional freshening, and surface warming might further reduce surface density. Once threshold stratification is reached, deep convection is strongly reduced in the (western) SPG, preventing winter cooling and further reducing the density in the interior of the gyre. Less dense water in the interior of SPG means weaker gyre circulation because of thermal wind effects; this in turn leads to reduced salt import from tropics and hence additional freshening. SPG collapse can occur without AMOC collapse, but the two may influence each other.

### 3.2.2: The impact of SRM

SRM's effect on the drivers is similar to the discussion in Sect. 3.1, although the relative importance of these drivers may differ.

Direct simulations of SRM's effect on the SPG are extremely scarce, with Pflüger et al. (2024) being the only study at date - to the authors' knowledge - to analyse the impact of SRM on SPG tipping. They show that in CESM2, the SPG collapses under an RCP8.5 scenario, but deep convection is preserved in the eastern part of the SPG if SRM is used to stabilise GMST at 1.5ºC above pre-industrial. We conjecture that SRM might at least partially counteract SPG collapse by reducing or reverting buoyancy forcing in the subpolar North Atlantic.

To our knowledge, no study has explicitly simulated SPG recovery due to SRM. Plüger et al. (2024) find that, when cooling an RCP8.5 scenario down to 1.5ºC from 2080 using SAI, SPG convection remains in the collapsed state at least for several decades.

### 3.2.3: Further Research

Some possible research avenues overlap with AMOC (sect 3.1.3), including improving process understanding in the North Atlantic and quantifying SRM's impact on drivers there. As opposed to AMOC weakening (Xie et al., 2022), to our knowledge SPG changes have not been systematically reviewed in GeoMIP data. As some climate models actually simulate SPG tipping, targeted

experiments could be performed in these models, e.g. applying SRM some time before the tipping to test SRM's preventative potential, and after the tipping, to assess reversibility.

## 3.3 Antarctic Overturning Circulation and Bottom Water formation

Antarctic Bottom Water (AABW) is a very cold and moderately salty water mass that forms around Antarctica by ocean heat loss (especially in ice-free areas, where water is exposed to very cold katabatic winds from Antarctica) and brine rejection during sea ice formation. It sinks to great depth, filling the abyssal ocean and constituting the lower branch of the lower Atlantic circulation cell (Fig. 2, process 5). Process understanding is still limited, as most climate models do not resolve small-scale processes such as circulation in ice shelf cavities, and meltwater input from Antarctica is typically not included (Fox-Kemper *et al.*, 2021). Observational and modelling evidence suggest a future weakening of AABW formation, and AABW formation collapse has been listed as a potential tipping point (Armstrong McKay et al., 2022; Loriani et al., 2023; see also SI).

### 3.3.1: Drivers and Feedbacks

A modelling study by Li et al. (2023a) finds that the major driver of AABW formation decline is meltwater input from Antarctica, which freshens the surface water flowing towards Antarctica (point (5) in Fig. 4) and inhibits sinking. In contrast, another modelling study (Zhou et al., 2023) finds that AABW formation in the Weddell sea has declined due to a decrease in southerly winds near the ice shelf edge, which push sea ice away from the shelf edge, thereby enabling surface cooling in the open water and sea ice production and hence brine rejection, both of which help increase density. The study suggests that the local wind changes are at least partly driven by natural variability over the Pacific, transferred through teleconnections. In addition, global warming is predicted to cause an intensification and southward shift of the westerlies around Antarctica (Goyal et al., 2021), leading to intensified upwelling of warm water around Antarctica. Dias et al. (2021) suggest that this may reduce sea ice cover and enhance surface cooling, convection and ultimately AABW formation, although this may be overestimated in models with overly large stretches of open ocean. Note that ocean warming around Antarctica is also expected to accelerate ice loss (Sect. 2.2) and hence freshwater input, which would again reduce AABW production (Q. Li et *al.*, 2023).

### 3.3.2: The impact of SRM

To our knowledge, no dedicated studies exist on the effect of SRM on AABW tipping. We conjecture that SRM's effectiveness to mitigate AABW tipping depends on its ability to counter drivers, especially

melting of land and sea ice (Sects.. 2.2 and 2.5). As outlined in Sect. 2.2, depending on the injection strategy, SAI may have limited effects on preventing the intensification and southward shift of the westerlies. It may thus fail to revert land ice melt, which exacerbates AABW loss, but also sea ice loss, which allows wider open stretches for convection and AABW formation (Sect. 3.3.1).

SRM's influence on secondary drivers, including Antarctic wind changes through teleconnections, may modify the outcome and is hard to predict; we currently do not have modelling of the impact of SRM on these winds. Given large uncertainties and the fact that SRM may affect various drivers in ways that may counteract each other, we cannot predict the sign of the overall effect. We also have no evidence as to whether SRM could reverse AABW tipping once started.

### 3.2.3: Further Research

Better understanding of processes determining AABW formation, and reducing model uncertainty, is key. Given the dependence on Antarctic ice melt, as well as its relation with the AMOC, understanding the impact of SRM on both of those tipping elements is also important. Finally, understanding the impact of SRM on Antarctic winds and the teleconnections that drive them may also be important if these prove to be influential in driving long-term trends of AABW formation.

# 4: Atmosphere

## 4.1: Marine Stratocumulus Cloud

Marine stratocumulus clouds are low-altitude clouds that form primarily in the sub-tropics, covering approximately 20% of the low-latitude ocean or 6.5% of the Earth's surface. Due to their location, high albedo and low-altitude they produce a very substantial local forcing of up to -100 Wm$^{-2}$ (Klein and Hartmann, 1993). Recent work has shown that these clouds exhibit multiple equilibrium states and that at sufficiently high Sea-Surface Temperatures (SST) or $CO_2$ concentrations they can transition from a cloudy to a non-cloudy state (Bellon and Geoffroy, 2016; Salazar and Tziperman, 2023; Schneider et al., 2019). The break-up of these cloud decks would be associated with substantial local and global temperature increases, with Schneider et al. (2019) finding a 10 °C warming within the affected domain and an enormous 8 °C global warming in response in their highly idealised setup.

### 4.1.1: Drivers and Feedbacks

Unlike most types of clouds, the convection that produces marine stratocumulus clouds originates at the cloud-top and is driven by longwave radiative cooling (Turton and Nicholls, 1987). If this longwave cooling is sufficiently strong, air parcels from the cloud top descend all the way to the ocean surface producing a well-mixed boundary layer that connects the cloud layer with its moisture source (Schneider et al., 2019). These cloud decks will break up if this longwave cooling weakens to such an extent that the descending air parcels can no longer reach the ocean surface (Salazar & Tziperman, 2023). This can occur if the longwave emissivity of the overlying atmospheric layer increases sufficiently, i.e., if GHG concentrations or water vapour content rise sufficiently (Schneider et al., 2019). It can also occur if too much of the warm, dry air from the overlying inversion layer is mixed into the cloud as this would dehydrate the cloud, reducing its emissivity and hence the longwave cooling that sustains it (Bretherton and Wyant, 1997).

Using a cloud-resolving Large Eddy Simulation of a patch of marine stratocumulus coupled to a tropical atmospheric column model, Schneider et al. (2019) found that if $CO_2$ concentrations rose above 1200 ppm there was a sudden transition from a cloudy to a non-cloudy state and a substantial local and global warming. As the feedbacks associated with this warming make it more difficult for these clouds to form, this transition exhibited considerable hysteresis, with $CO_2$ concentrations needing to be brought back below 300 ppm for the system to return to the cloudy state. Salazar and Tziperman (2023) reproduced this hysteresis in an idealised mixed layer cloud model, finding multiple equilibria between 500 and 1750 ppm.

### 4.1.2: The impact of SRM

In a follow-up study, Schneider et al. (2020) found that whilst reducing insolation to offset some of the warming from elevated $CO_2$ concentrations did not eliminate this hysteresis, the critical threshold for marine stratocumulus break-up is raised from >1200 ppm in their $CO_2$-only runs to >1700 ppm. The increase in global temperatures is reduced from ~8 °C to ~5 °C, though $CO_2$ concentrations must still be brought below 300 ppm to restore the clouds.

However, the reduction in insolation that they imposed in their simulations only offset roughly half of the warming from their elevated $CO_2$ concentrations. While simulations by the GeoMIP found that a reduction of between 1.75 and 2.5% was needed to offset each doubling of $CO_2$ concentrations (Kravitz et al., 2013), Schneider et al. (2020) applied only a 3.7 $Wm^{-2}$ reduction for every doubling of $CO_2$ to the 471 $Wm^{-2}$ of incoming sunlight in their sub-tropical domain, i.e., a 0.8% reduction. As warming

increases the latent heat flux from the surface that leads to greater cloud-top turbulence and the dehydration of the clouds, and it leads to increased water vapour in the overlying inversion layer, the residual warming in these SRM simulations substantially weakens the longwave cooling that sustains the clouds. This may suggest that if Schneider et al. (2020) had reduced incoming sunlight sufficiently to eliminate the residual warming in their simulations they would have found a much higher critical $CO_2$ threshold in their SRM case.

Some support for this conclusion on the effects of this residual warming can be found in the sensitivity tests of Salazar and Tziperman (2023). In one case (in Fig. 4, row 2 in Salazar and Tziperman (2023)) they eliminate the water vapour feedback from their model, breaking the association between temperature and emissivity in the inversion layer, and find that the critical $CO_2$ threshold for marine stratocumulus collapse is more than doubled from 1750 to >4000 ppm. However, in this case they still have elevated sea surface temperatures, and so a greater latent heat flux from the surface than would be the case if SRM fully offset the warming.

While SRM would not address the reduction in longwave cooling caused by elevated GHG concentrations, it would be effective in lowering temperatures, reducing the water vapour feedback and the increase in turbulence caused by increased latent heat flux from a warmer ocean surface. As such SRM would substantially raise the critical $CO_2$ threshold for marine stratocumulus from a very high $CO_2$ concentration to an extremely high $CO_2$ concentration.

**4.1.3: Further Research**

To date there has been very little research into this potential tipping point, as such further research in a wider range of models is needed to determine whether it is a robust feature of marine stratocumulus decks. As the CO2 concentrations and temperatures required to produce this tipping point may have occurred at certain points in the past, e.g., the Paleocene-Eocene Thermal Maxima (Schneider et al., 2019), future research could address whether observations and model simulations of this period are consistent with this potential tipping point.
To assess SRM's potential to address this tipping point more fully, a wider range of SRM simulations than those in Schneider et al. (2020) could be conducted. For SAI, such simulations should include the effects not present in sun-dimming experiments, such as stratospheric heating, and should cover a range of scenarios with different levels of GHG forcing where SAI offsets all warming. Studies assessing MCB's potential to address this tipping point would also be particularly worthwhile as MCB would

directly modify marine stratocumulus clouds, changing the cloud microphysics in ways which may
affect the threshold for collapse.

# 960 5: Biosphere

## 961 5.1: The Impacts of SRM on ecological systems in general

Tipping points have been extensively discussed in the ecological literature (Jiang et al., 2019a), and
ecological systems in the tipping literature (Lenton et al., 2023). Ecologists refer to tipping points for
complete system changes either in the dominant, foundational or keystone species, in the life forms or
functional types of the plants (e.g. from trees to grasses), to large changes in the community of
organisms present (e.g. diverse native species community to monocultures of an invasive species), or in
the physical structure of an environment (wetland or aquatic to dry land, deep soil to eroded rock
substrate). Moreover, the ecological literature refers to tipping points not only with respect to such
changes at the system level (which we focus on here), but also to the point at which the extinction of an
individual species becomes inevitable (Osmond and Klausmeier, 2017). Such changes may be driven by
self-sustaining drivers and positive feedbacks, or to sudden or persistent drivers without positive
feedbacks (Fig. 1).
The losses of biodiversity locally, regionally and globally in the last half century, accelerating in recent
975 years, has particularly focused attention on tipping points resulting in biological losses. Ecological
systems are typically driven over tipping points by a complex series of drivers - including non-climatic
drivers (Lenton *et al.* 2023) - rather than single dominant drivers from local to global spatial scales, and
SRM is likely to change many environmental factors affecting these systems (Liang et al., 2022).
Greater uncertainty of knowledge of climate impacts at local and regional scales can make
understanding the impacts of particular climatic changes difficult, and exploitation and land-use change,
amongst other anthropogenic factors, can interact to make these systems more susceptible to
climate-driven tipping.
There has been very little research on the impacts of SRM on complex ecosystems. The clearest clues as
to whether SRM can prevent ecological tipping points lie in its central role of reducing global average
warming (albeit with regional uncertainties), and thus those ecological systems that suffer most from the
direct impact of increased temperatures might potentially benefit from SRM-induced cooling and evade
temperature-forced tipping points. However, responses such as species distributions, species interactions
(e.g. pollination), and ecosystem processes such as net primary productivity may be more affected by

specific aspects of weather and climate that directly impact organisms. These may include reductions in precipitation or changes in seasonality of precipitation relative to temperatures,  increases in peak extreme  temperatures, which are generally reduced by SRM (Kuswanto et al., 2022), reductions or loss of freezing temperatures and increase in nighttime temperatures, which are reduced substantially, but not fully, by SRM (Zarnetske et al., 2021), and other factors including growing season duration, and consecutive days of extreme temperatures. Some factors affected by temperature may drive ecological effects in opposite directions as well;  for example cooling may suppress photosynthesis due to a drop in productivity or increase it if the suppression of heat stress is more significant (Zarnetske et al., 2021). Thus even for the factor where we best understand the climatic effects of SRM, the effects on pulling them back from, or pushing them over, tipping points, remain challenging to predict.

Changes to the hydrological cycle under SRM are central to plant productivity, growth, survival and reproduction. However, large uncertainties in the simulated hydrological consequences of different SRM schemes (Ricke et al., 2023) preclude a simple answer as to whether a SRM scheme would alleviate or exacerbate hydrological-related drivers of tipping. It will be critical to understand both observed and modelled ecological responses to changes in precipitation and atmospheric drought (e.g. vapour pressure deficit) for SRM scenarios to better anticipate changes that can drive or prevent ecological tipping.

SRM would also affect other factors in novel ways when compared to climate change. Whilst temperatures would be kept artificially low, $CO_2$ levels may remain high or rise, with profound impacts on terrestrial and marine ecosystems (Zarnetske et al., 2021). Diffuse to direct light ratios would be enhanced under SRM, potentially enhancing or otherwise altering photosynthesis for photosynthetic organisms (Xia et al., 2016).

Other factors besides average global temperatures are sensitive to the exact configuration of the deployment scheme of SRM. Changes in SRM scenarios may have profoundly different impacts on ecosystems. For example, if SRM were to continue for decades and then be suddenly terminated while $CO_2$ continued to increase, the termination effects on ecological systems (Ito, 2017; Trisos et al., 2018) would be so disruptive that tipping points would almost certainly be precipitated for many ecological systems, as many of these are examples of rate-dependent tipping (Fig. 2). The latitude(s) of injection sites would influence many aspects of climate relevant to potential ecological tipping points, including movement of the Hadley cells and the arctic-to-tropic temperature gradient (Cheng et al., 2022; Smyth et al., 2017).

# 5.2: Tropical Forests: Amazon Rainforest Collapse

The Amazon basin is a region of many different tropical forest ecological systems and high biodiversity. It is a key Earth system component (Armstrong McKay et al., 2022), regulating regional and even global climates (Wunderling et al., 2024) by cycling enormous amounts of water vapour and latent heat between land and atmosphere, by storing around 150–200 Pg carbon above and below ground, though this is in decline (Brienen et al., 2015). As such, it is perhaps better to see the Amazon basin as a combined ecological-climatic system.

It is predicted that 2-6$^{\mathrm{O}}$C of global warming (relative to preindustrial), and even less when considering interactions with other human activities such as clearcutting and fires, might force a tipping point for the Amazon basin to the replacement of tropical forest with systems without trees or with fewer, scattered trees and without continuous canopies (Lenton et al. 2023). Indeed, whilst the Amazon has a series of local tipping elements within it, these can be considered to be connected by the atmospheric moisture recycling feedback, where intercepted precipitation and transpiration allows evapotranspiration from the forest to be recycled into precipitation elsewhere. This spatially connects the different local tipping points together, potentially allowing for tipping cascades through each of the local elements (Wunderling et al., 2022b).

## 5.2.1: Drivers and Feedbacks

As is the case for most highly diverse tropical forests globally (e.g., the Dipterocarp forests of Southeast Asia, SI), the forests of the Amazon are affected by multiple interacting factors that together may precipitate tipping. The major climatic driver behind this tipping point is drought caused by decreasing precipitation and increasing evaporation in this region during the dry season under global warming, whilst annual precipitation changes seem of limited importance (Wunderling et al., 2022b). Secondary drivers related to warming include more widespread and frequent occurrence of extreme heatwaves (Costa et al., 2022; Jiménez-Muñoz et al., 2016) that cause tree and animal mortalities either directly or indirectly through increased wildfires and droughts. Feedbacks are likely to cause or accelerate such a tipping point because as global climate change induced drought kills areas of forest, the precipitation those trees had cycled back to the atmosphere disappears, furthering drought and killing more forest. Studies have found that vegetation-climate feedbacks in the Amazon could be significant in tipping. For example, Zemp et al. (2017) illustrated a feedback loop of reduced rainfall causing an increased risk of forest dieback causing forest loss induced intensification of regional droughts that self-amplifies forest loss in the Amazon basin. Staal et al. (2020) further delineated a bistable state of forests in the southern

Amazon, which are most susceptible to the drought-dieback feedback loop that would tip these forests to a savanna-like non-forested state.

Fire is another major driver of tipping, driven by climatic and non-climatic sources, which is raised in significance if micro-climatic inertia is important (Malhi et al., 2009). The increase in human activity and forest fragmentation increases the proximity of much of the forest to anthropogenic ignition points, which as the forest dries is the limiting factor in fire frequency, increasing the likelihood of tipping (Malhi et al., 2009). The impact of deforestation and degradation is the final significant driver of tipping (Lenton et al., 2023), which not only causes increased vulnerability to other tipping drivers (Wunderling et al., 2022b), as well as definitionally causing localised state changes, but via cascades may itself be a key driver of changes to the combined ecological-climatic system in the Amazon basin (Boers et al., 2017).

Some researchers have suggested that ecosystems capable of developing Turing patterns might have multistability with many partly vegetated states, which may enhance resilience and lower irreversibility (Rietkerk et al., 2021); it is unknown how SRM would enhance or detract from this resilience, so these will not be discussed further.

Some changes in oceanic and atmospheric circulations due to climate change could also have indirect, beneficial effects on the resilience of Amazon forests. For example, the possible AMOC collapse with elevated warming (Sect. 3.1) is projected to shift the Intertropical Convergence Zone southwards (Orihuela-Pinto et al., 2022) and cause increased rainfall and decreased temperature in most parts of the Amazon, which would stabilise eastern Amazonian rainforests (Nian et al., 2023) by mitigating the above-mentioned drought-dieback feedback loop.

## 5.2.2: The impact of SRM

Limited research makes predicting the effects of SRM on Amazon tipping deeply uncertain, given that it is highly dependent on a number of factors, some poorly understood, and that some of the conditions created by SRM are novel. In addition, large areas of the Amazon are poorly studied, and the climatic drivers are not fully understood (Carvalho et al., 2023). We know that Amazon forests are highly dependent on regional precipitation and are particularly sensitive to drought. GCMs can be used to provide insight to understand the large-scale impacts of SRM, but tropical forests commonly depend not only on global circulation patterns, but also may depend on regional changes including monsoon dynamics and thus the movement of the Hadley cells, and on convection-forest interactions, which are often inadequately captured in models (indeed, GCMs often disagree on even the sign of these regional

precipitation change). Moreover, the effects are likely to depend on the specifics of the particular SRM scenario, and different SRM approaches may have very different regional and local meteorological and ecological consequences even if they aim for similar global average temperatures (Fan et al., 2021). Changes in relative humidity and vapour pressure deficit are also important for forest function (Grossiord et al., 2020), with vapour pressure deficit generally decreasing under SRM and thus alleviating atmospheric aridity and stomatal stress even with reduced precipitation (Fan et al., 2021). Whether global warming is increasing land aridity or not is a highly debated topic (Berg and McColl, 2021) and in light of this, whether SRM would alleviate or exacerbate aridity (including Amazon drying) is likewise highly uncertain. Moreover, effects may be in different directions; for example, given SRM could stabilise the AMOC (Sect. 3.1.2), this would aid the tipping process, even when other effects may help prevent it. Because SRM would not reverse climate change but would create novel environmental conditions, predicting the consequences beyond lowered temperatures in Amazon forests is extremely difficult. For example, in contrast to same-temperature conditions obtained by $CO_2$ reduction, SRM would result in lower temperature but elevated $CO_2$ levels, and changes in direct/diffuse light ratio, with currently poorly understood vegetation responses.

Jones et al. (2018) used models of SAI deployment to keep temperature to $1.5^O$C above preindustrial, and found that Amazon drying is very imperfectly compensated for by the deployment, although it is reduced relative to same-emission scenarios. The compensation is better in the East Amazon, where tipping concern under climate change is the greatest, than the West Amazon. They suggest that this is because much of the hydrology of the Amazon is controlled by changes to annual-mean photosynthetic activity and stomatal conductance, which are driven by elevated atmospheric $CO_2$ levels as well as temperature. These may also be impacted by the type of light, although this was not explored in the study. Simpson et al. (2019) see precipitation reductions over the Amazon in GLENS that are equal to that of the comparative non-SAI scenario (RCP8.5), although soil moisture is greater under SRM than RCP8.5, as evapotranspiration is suppressed. This P-E reduction was also seen in Jones et al (2018). However, this analysis is limited as it looks at annual precipitation rather than droughts, with the latter a much stronger driver of Amazon tipping. Touma et al. (2023) uses an SAI scheme to keep temperature close to $1.5^O$C above pre-industrial, and sees increases in drying and fires in the West Amazon when compared to SSP2-4.5, whilst a reduction in fires in Northeast Brazil, which includes part of the East Amazon. However, drought severity is found to increase slightly for both regions under SRM when compared to SSP2-4.5. In general, the East Amazon is the area of greatest concern for tipping behaviour under climate change (Malhi et al., 2009), so in our overall judgement we have weighted the impact of SRM on this region higher, although the possibility of cascades through the

atmospheric-moisture recycling feedback means that the drying in the West Amazon cannot be ruled out as precipitating regional tipping.

Whilst this may give some indication of possible regional climatic effects, the reliability of these results in such a complex system which GCMs struggle to represent is questionable so the effect SRM has on Amazon tipping remains highly uncertain. Moreover, SRM does not affect deforestation or the proximity of the rainforest to ignition sources, which are key drivers of tipping.

### 5.2.3: Further Research

In light of the complexity of the ecological system and regional- to micro-climatology in the Amazon, more research is needed to better represent bioclimatological (vegetation-climate interaction) processes in GCMs and their land surface models in order to constrain future projects of the impact of SRM on Amazon forest tipping. Better monitoring of and incorporating spatial data on land use change in the Amazon basin and more widely in tropical forests globally is essential for realistic predictions; increasing the number of monitoring stations  and continued archiving of satellite imagery of the Amazon microclimate and forest health status is critical for enriching empirical knowledge of this unique system to support model development (Carvalho et al., 2023). Better understanding of the relationship between phylogenetic diversity and plant functional traits, and their heterogeneity across the Amazon Basin will facilitate more accurate predictions of responses to climate change and the effects of SRM in promoting or reducing incipient tipping points. The contrasting effects of SRM on hydrological aridity (precipitation and soil moisture) and atmospheric aridity (vapour pressure deficit), and their competing effects on forest health is also worth attention in assessing the overall effect of SRM on the Amazon system. Furthermore, better understanding the importance of droughts and fires in different regions to overall Amazon dieback, may allow us to constrain the effect of the differential regional impacts of SRM on the tipping element as a whole.

## 5.3: Shallow-Sea Tropical Coral Reefs

Corals are invertebrate animals belonging to thousands of species in the phylum Cnidaria, living in a range of marine environments. A reef is built up by the excretion of calcium carbonate from millions of coral polyps, which keep building up toward the light, leaving the coral reef structure underneath. The structure created by the corals creates a massive habitat for many other organisms. Tipping in shallow-water tropical coral reefs results in the establishment of an entirely different biotic and physical community space, often dominated by macroalgae without these hard skeletons (Holbrook et al., 2016).

More recent work has highlighted the presence of multiple stable states if fish are considered alongside benthic functional groups (Jouffray et al., 2019).

### 5.3.1: Drivers and Mechanisms

Ocean warming is a primary driver of shallow-sea tropical coral reef tipping, normally via sustained high temperature events causing coral bleaching (Fox-Kemper *et al.*, 2021). During these events, corals will expel their symbiotic photosynthetic dinoflagellates; if they are bleached for extended periods of time, this can result in death (Wang et al., 2023). If the corals are then replaced by other organisms, chiefly macroalgae, then a transition to an entirely new stable state can occur (Schmitt et al., 2019). It sometimes may be possible for the scleractinian coral to reestablish themselves after mass mortality events. However, warming is projected to outpace the adaptive capacity of corals with recurrent bleaching events making recovery very difficult, causing transitions to a second stable state to be more likely (Hughes et al., 2017). Other interactions such as a drop in herbivory may make it easier for the macroalgae to become established, further promoting tipping (Holbrook et al., 2016).

Acidification is a secondary driver of tipping. As more CO2 dissolves in ocean water aragonite saturation levels drop, so calcification by the polyps decreases, leading corals to either reduce their skeletal growth, keep the same rate of skeletal growth but reduce skeletal density increasing susceptibility to erosion, or to keep the same skeletal density and rate of growth whilst diverting resources away from other essential functions (Hoegh-Guldberg et al., 2007). Dead coral structures are also dissolved or eroded at a faster rate in more acidic water, further reducing reef functioning. Nonetheless, the relationship between increased acidification and decreased calcification is complex with studies equivocal over how strong this relationship is, as well as how important non-pH factors are in changes to calcification rate (Mollica et al., 2018). The response of coral calcification to acidification is generally linear and highly species specific, so a simple 'coral acidification tipping point' does not exist. Other factors, such as internal pH regulation, may have physiological tipping points, but manifest as linear decreases at an ecosystem-wide level. However, coral reefs are complex communities with non-coral species playing important roles, and whilst most acidification impacts are linear, there does seem to be some evidence of tipping on a local scale due to the indirect effects of acidification on the overall health of the community in specific habitats, particularly those with an already high $pCO_2$ (Cornwall et al., 2024). Nonetheless, these are unlikely to manifest as a global, near-synchronous, tipping point.

Other factors may also contribute to coral tipping. Storm intensity is expected to increase under warming, causing physical damage to the reef which recovery may be difficult from (Gardner et al.,

2005; Mudge and Bruno, 2023). Sea level rise, if it outpaces the coral's ability to track, which may be the case due to the other factors mentioned, can promote increases in sedimentation. However, (Brown et al., 2019) find sea level rise promotes reef growth, likely by allowing space for the reef to grow, reducing aerial exposure and exposure to turbid waters. A variety of non-climatic or $CO_2$ related anthropogenic factors are also important. Jouffray et al. (2019) identified a number of different stressors on Hawaiian coral reefs, including fishing and pollution, and finds in certain regime shifts this has been a more important driver than climatic factors. Moreover, diseases (Alvarez-Filip et al., 2022) and invasive species (Pettay et al., 2015), often associated with warming and global trade, also have negative impacts on the structure, functioning and stability of coral reefs such as those found in the Caribbean.

### 5.3.2: The impact of SRM

SRM would help to reduce coral reefs tipping by reducing ocean temperatures (Couce et al., 2013), thus likely reducing the frequency of bleaching events. SRM may increase acidification somewhat by decreasing pH and aragonite saturation relative to the same emissions pathway without SRM, due to cooler water having a higher $CO_2$ solubility (Couce et al., 2013). However, Jin et al. (2022) argues that it is more complex; temperature decreases tend to increase pH and aragonite saturation for a given $pCO_2$ (Cao et al., 2009), whilst cooler temperatures generally reduce calcification and thus lead to lower pH and aragonite saturations. Their results suggest that whilst pH is slightly increased under SRM, aragonite saturation, the key variable of interest, is negligibly affected; thus we should expect SRM to have a close to negligible impact on the acidification driver of coral tipping.

SRM is likely to decrease the intensity of tropical storms, although with low confidence (Moore et al., 2015). Wang et al. (2018) find that SRM decreases the number of tropical cyclones relative to the same emissions pathway without SRM, although it does increase in the South Pacific, and so its overall impact on coral reef tipping is unclear. The impact is also heavily scenario dependent (Jones et al., 2017; Wang et al., 2018).

The impact of SRM on the incoming radiation, both by reducing the amount of direct radiation and increasing the diffuse radiation, is also likely to impact photosynthesis but any effect on tipping behaviour of photosynthetic organisms is likely to be minimal due to the cancellation effects between direct and diffuse radiation changes induced by SRM (Durand et al., 2021; Fan et al., 2021; Shao et al., 2020). These studies, however, were carried out in terrestrial environments, so the effect on zooxanthellae algae may be different. Non-climatic or $CO_2$ related anthropogenic drivers will be unaffected by SRM.

Couce et al. (2013) finds that suitability for reef conditions are improved under SRM when compared to same emission pathway scenarios, although worse than same temperature scenarios generated through mitigation. However, conditions in much of the Pacific improved relative to present day. Zhang et al., (2017) specifically look at Caribbean coral reefs, and find that coral bleaching is significantly reduced by SRM due to its effect in allowing temperature to remain below the critical threshold for corals. Moreover, SRM is seen to reduce the frequency of Category 5 hurricanes, and whilst the recurrence time is increased, this is not enough to fully offset the impacts of climate change. Relative to the same emission pathway scenarios, both studies see SAI as reducing the likelihood of coral reef tipping, although they both report an undercompensation for the changes seen due to climate change.

There has also been interest in the use of MCB in combating bleaching, particularly short-term use around bleaching events (Tollefson, 2021). Theoretically, such a programme ought to reduce bleaching on the corals, although full analysis of the limited field experiments carried out have not yet shown if the technology is capable of attaining the necessary cooling.

### 5.3.3 Further Research

Given the high level of temperature dependence of the climatic drivers, our understanding of the direction of the impact of SRM on coral reef tipping is quite strong, and so further research is here less of a priority than other tipping elements. Nonetheless, the lack of modelling studies, combined with the presence of uncertainties (such as the difference in SRM impact across regions) and co-drivers alongside temperature (such as bleaching) might indicate that up-to-date ESM studies of SRM's impact on coral reefs would be useful. Studies of how much SRM might be necessary and what deployment design is needed to keep below critical thresholds of Degree Heating Week and recurrence times, as well as the impacts on storm intensity would be useful too. We also lack the understanding whether reducing the temperature driver is sufficient to stop tipping if other drivers of tipping are severe enough. The interest in regional MCB to avoid tipping would also require further research to test if proposed schemes are feasible. Similarly, better research with how other reef restoration strategies may interact with SRM to reduce the probability of tipping, or may reduce its counterfactual impact, may also be important for the most realistic assessment.

## 5.4: The Himalaya-to-Sundarbans (HTS) Hydro-ecological System

There is a vast region that extends from the glaciers of the Himalaya through their foothills, to a riparian network of the Ganges, Brahmaputra and Meghna Rivers with their extensive river basins, ending in the

enormous wetlands of the Sundarbans in the Bay of Bengal. It includes areas partially or entirely within five different nations (India, China, Nepal, Bangladesh and Bhutan) with between 400 -750 million people (depending on how one defines its boundary). This large system includes a range of glacial and contrasting ecological realms, and the different parts of this system have typically been treated separately and viewed as being independent components. Consequently it has been assumed that while there might be localised tipping in these different components (for example, in the glaciers of the Himalaya) resulting from different drivers in response to climate change (as for sea level rise for the Sundarbans), there would be no systemic response and no generalised tipping of the entire system.

Here we suggest, for the first time, that the HTS hydro-ecological system is a plausible candidate as a single, integrated regional impact tipping element, according to our definition of tipping process in multi-dimensional systems (Fig. 1f), although this tipping process may appear different from the better-known and possibly simpler forcing-driven tipping processes (Fig. 1a,b) in other more familiar tipping elements as established by (Armstrong McKay et al., 2022; Lenton et al., 2008). We present this as an alternative hypothesis to that of the independent tipping of its components, and present an argument that the systemic tipping hypothesis proposed here bears more investigation. The ecological and socio-cultural importance of the HTS hydro-ecological system means that the impact of SRM on tipping in this system, regardless of the scale of said tipping, should be seriously evaluated, and we suggest that this subcontinental system, while poorly understood and understudied,  may possibly be an integrated if underappreciated component of the Earth System.

The diverse ecological systems in the HTS are dependent on the interconnections between the glacial-riparian network originating from Himalayan glaciers, the monsoon, and on the interface between the marine and terrestrial environments at the deltas of the Ganges, Brahmaputra and Meghna Rivers in the Sundarbans. The HTS as a whole includes important biodiversity hotspots, including the eastern Himalaya/southwestern China (Sharma et al., 2009) and the Sundarbans. The Sundarbans are the largest and most biodiverse mangrove wetlands in the world. Analogous to coral reefs, the mangroves form a living physical structure that creates habitat that supports many other species and complex species interactions (Raha et al., 2012; Sievers et al., 2020). We chose to highlight the HTS system to bring attention to the potential for SRM to impact this ecologically and socially important system. We also hope to illustrate how our approach can allow for evidence informed hypotheses on the effects of SRM of systems where the possibility of systemic tipping is very uncertain and under-evidenced, and to illustrate how other complex and multidimensional ecological systems might plausibly show broad systemic tipping.

We hypothesise that changes to water variability and availability due to climate change might be a plausible trigger of systematic tipping to multi-dimensional alternative stable states (Fig. 1f) in this

potentially integrated system. This mosaic of habitats and biomes is interconnected and interdependent on the water that originates in the glaciers of the Himalaya and feeds the river systems which are essential to the living systems of the HTS. Glacial melting (Sect. 2.3) to a critical level (Kraaijenbrink et al., 2017) and subsequent decline or seasonal failure of river flow and groundwater recharge (Nie et al., 2021; Talukder et al., 2021; Whitehead et al., 2015) could act as a potential driver or trigger other drivers (Sect. 5.4.1) of tipping for the whole system, and the joint dependence on the monsoon exacerbates the likelihood of potential system-wide state changes, albeit of a highly uncertain nature and threshold. We posit that these different but connected ecological systems are not independent, and that climate change will not affect them independently but rather that state changes in subsystems may potentially be linked at the system level. As temperature change and associated glacial-hydrological changes and monsoon changes pass possible thresholds (Mall et al., 2022; Mishra et al., 2021; Swapna et al., 2017) they could possibly tip the whole system to multidimensional new states (Fig. 1f). That is, we are positing that the potential drivers are hydrological, linking the HTS via the behaviour of the monsoon and from the Himalayan glaciers feeding a network of major river systems. It is at present difficult to define a clear and specific threshold, but it seems plausible that the entire system would be affected by these hydrological changes in a linked manner. Tipping to alternative states for parts of the HTS system is already occurring and is likely to accelerate with climate change, with system alteration to different habitats or even biomes and degradation of native and endemic species diversity (Negi et al. 2022), changes in species distribution (Telwala et al., 2013), increasing dominance of invasive pan-global species adapted to high levels of disturbance, and global decreases in cold-tolerant and cold-adapted species. Human responses to climate change or to SRM in this densely populated hydro-ecological system, including land use change and human migration, would have unpredictable effects on tipping.

These system changes may be integrated with biogeophysical and biogeochemical changes, with implications for future climate through complex feedback mechanisms involving albedo, hydrological cycles, changes to salinity in the Bay of Bengal, soil nutrients and microbial processes, ecosystem dynamics, and other factors.

It is not known what alternative states would be should this complex and hydrologically integrated system be driven by climate change past a tipping point, but one speculation is low diversity mixed shrublands and grasslands, possibly dominated by invasive species, if high variability of water availability associated with monsoon changes combine with systematic drought after glacial melting and warming-induced increased evaporative demand. Whether SRM would cool sufficiently to prevent the loss of the Himalayan glaciers is discussed earlier (Sect. 2.3).

## 5.4.1: Drivers and Mechanisms

There are a number of potential climate change-induced drivers of tipping in the HTS system, including melting montane glaciers, changes in mean and extreme river flows, changes in the seasonality and intensity of the monsoon and behaviour of the Hadley cells, sea level rise, droughts and extreme high temperatures (Kraaijenbrink et al., 2017; Mall et al., 2022; Mishra et al., 2021; Swapna et al., 2017). Among these drivers, we posit that systemic changes in the water cycle and declining water availability after unsustainable glacier melt or monsoon changes could be the dominant driver that force systemic tipping in HTS. Global warming is melting high elevation glaciers rapidly worldwide (Sect. 2.3) (Hugonnet et al., 2021), with accelerated ice loss observed across the Himalayas over the past 40 years (Maurer et al., 2019) and a likely non-linear increasing trend with greater than $3^{O}C$ warming (Rounce et al., 2023). Glacial melting in the Himalaya (Potocki et al., 2022; Kraaijenbrink et al. 2017) would result in tipping in the immediate area below the glaciers, and also for the vast areas of the HTS system, including the Ganges-Brahmaputra-Meghna basin below dependent on these glaciers as a source of water. Recent studies already show that the accelerated melting of Himalayan glaciers and Tibetan Plateau snowpacks are triggering downstream hydrological changes (Nie et al., 2021), and increasing agricultural risks (Qin et al., 2020). Changes in the distribution, intensity and timing of tropical monsoonal rains in the HTS are also potential drivers of in tipping the ecological, agricultural, and human systems that depend on them. For example, climate change has been implicated in the weakening of Indian summer monsoon in recent decades (Mall et al., 2022; Mishra et al., 2021; Swapna et al., 2017), which would cause catastrophic change to some natural and agricultural systems if future monsoon changes intensify. Severe and extended heat in this region in recent years, exacerbated by drying, is likely to directly affect organism survival, species abundances and lead to extinctions, pushing some natural systems over tipping points (Mishra et al., 2020). Im et al. (2017) predicted that extreme heatwaves would exceed the human survivability limit ($35^{O}C$ wet-bulb temperature) at a few locations in the densely populated agricultural regions of the Ganges and Indus river basins and would approach the survivability limit over most of South Asia under the RCP8.5 scenario by the end of the century (i.e., about $4.5^{O}C$ warming relative to preindustrial). Climate induced sea level rise, exacerbated by extensive river damming, is contributing to the tipping of the vast coastal mangrove systems that are an integral part of the HTS system. There also exist significant non-climate related drivers of tipping in this system, particularly deforestation (Pandit et al., 2007). Finally, it could be possible that a multitude of these drivers are likely to interact and reinforce each other to force ecological tipping at the system level, although further studies are needed to test this hypothesis.

## 5.4.2: The impact of SRM

Climate-related drivers of tipping for the complex HTS system that would be affected by SRM are glacial melting and other monsoonal change, rising sea levels, drought and extreme heat. First, SRM would partially slow the melting of Himalayan glaciers (Sect. 2.3), reducing the probability of drying out in the river systems that would drive systemic tipping of the HTS system. While SRM might relieve the likelihood of hitting tipping points caused by glacial melting, changes to the movement of the Hadley cells predicted from some SAI scenarios might result in changes in the seasonality and predictability of the monsoons, leading to drought-induced tipping of the entire HTS system by removing the rainfall needed to sustain all of the coordinated components of the system (Cheng et al., 2022; Mishra et al., 2021; Smyth et al., 2017). Eventual and partial reductions in sea level rise due to cooling from SRM, and restoration of riparian freshwater from restoration of glaciers, might have some restorative effects in pulling the mangrove forests ringing the Bay of Bengal back from tipping. However, the anthropogenic effects of damming and other land use changes might reduce these potential reversals of tipping for this part of the HTS system, or alter their probability in an unpredictable manner. Finally, reduction of the extent and severity of extreme heat and likelihood of compound drought and heat extremes from the implementation of SRM could act directly to prevent region-wide drought-heat-related deaths and extinctions of keystone species and others, preventing catastrophic changes in ecosystems and therefore pulling back system tipping points from occurring.

## 5.4.3: Further research

Research directions to better understand the potential impact of SRM on the HTS earth system element largely overlap with progress in research on mountain cryosphere, sea level rise and extreme events. While aspects of this system have been studied, much more work on the nature of the complex integrated networks that comprise this system will be critical not only for understanding the HTS, but as a model for understanding other large systems that integrate major climatic, biological, and human dimensions. Moreover, understanding if systemic tipping is possible will require establishment of the extent to which the proposed mechanisms actually act to unify this diverse system, and whether this integration is sufficient for synchronous tipping. Ecological tipping in these regions may happen before climate-driven tipping in Himalayan glaciers, sea level, and Indian monsoons because the functions of these biodiversity hotspots depend not only on external drivers in climate and hydrology but also on their internal feedbacks and human disturbance (such as damming). These human actions could exacerbate the risks of collapsing or tipping. Therefore, the timing and thresholds of tipping in these biodiversity hotspots and how these will respond to climate change and SRM requires collaborative research between climatologists, ecologists and biologists. Far greater awareness of this overlooked but major earth system element among scientists and the general public is also critically needed.

## 5.5: Northern Boreal Forests

The northern coniferous forest, is the largest of Earth's biomes, and although low in biodiversity with many circumboreal species and genera, also is a major reservoir for carbon. Anthropogenic warming is greatest in these northern regions due to Arctic amplification (Serreze and Barry, 2011), and warming nights and extended periods of extreme heat are directly and indirectly forcing major structural changes in some parts of this biome, potentially precipitating tipping points, perhaps from forests to shrublands or grassland due to biotic and abiotic disturbances (Seidl et al., 2017) or from shrublands or grasslands to forests due to temperature-driven northern migration of boreal trees (Berner and Goetz, 2022). Rao et al. (2023) found that climate change is predicted to expose a foundational and dominant tree species across the entire region, *Larix siberica*, to temperatures that result in irreversible damage to photosynthetic tissue in the near future, leading to widespread and abrupt synchronous tree mortality. Tree mortality at this extent would be likely to cause a tipping point for the entire southern boreal forest system to a grassland-steppe system, as has been already observed in some areas (Li et al., 2023b). They suggest that an abrupt tipping point may be reached within the next decades which would "fundamentally and irreversibly alter the ecosystem state at regional to sub-continental spatial scales" for hundreds of km along an extensive area in the southern Eurasian boundary of the northern coniferous forests.

## 5.5.1: Drivers and Feedbacks

Warmer temperatures, increased evaporative demand, increased droughts, lower water availability and reduced snowpack and duration of snowpack under climate change all directly stress the coniferous forest (Ruiz-Pérez and Vico, 2020) and in doing so makes them more vulnerable to other stressors such as insect attack. Northern expansion of bark beetles (Singh et al., 2024; Venäläinen et al., 2020) and reduced generation times for these and other pests have killed large expanses of northern coniferous forests, and the dead and dying trees combined with warmer temperatures and drought have drastically reduced fire return intervals in many areas and greatly increased the scope and severity of fires (Bentz et al., 2010). The effects on feedbacks to climate are complex and difficult to predict. Reduced duration of snow cover reduces albedo, potentially increasing surface absorption of direct radiant energy from sunlight by the dark canopies of these trees. A tipping point leading to a shift from boreal forest to grassland/steppe might potentially increase albedo, at least during the growing season. Extensive fires and decomposition of soil carbon stores resulting from thawing of permafrost would greatly decrease carbon storage and contribute to increases to atmospheric carbon and global warming (Ruiz-Pérez and Vico 2020). Thus dieback can have opposite regional (cooling by increased albedo) and global (warming by carbon release) climatic effects. These dynamics could interact in complex stochastic

ways, with potential for positive feedbacks. Other climate elements that can lead to tipping in this system include thawing of permafrost (Sect. 2.6).

### 5.5.2: The impacts of SRM

As far as the authors know, there are no specific studies on the impact of SRM on boreal forests. By cooling average temperatures, it is possible that the consequences of SRM for the driving forces that either promote (northern migration of trees) or suppress (fires and insect attacks) northern coniferous forests might all be lessened and the system pulled back from such tipping points in either direction. On the one hand, cooler temperatures are likely to slow or stop the migration of trees into tundra and preserve the original biome configuration. On the other hand, extending periods below freezing by SRM might limit the northward spread of destructive insect outbreaks, extend snow cover, and possibly reduce drought and vapour pressure deficit, enhancing the resilience of these forests and pulling them back from a tipping point. Preservation of cold temperatures and prevention of extreme heat events could prevent widespread mortality of Larix and other foundational tree species in the boreal forest, likewise pulling it back from a tipping point from forest to steppe. By reducing the frequency and extent of boreal forest wildfires, reductions in heat could also reduce the positive feedbacks between loss of carbon stores in living trees and soil organic matter and the carbon in the atmosphere. Furthermore, given complex eco-hydrological mechanisms in boreal forest dynamics, the large uncertainty in simulated regional precipitation changes under SRM might complicate the above temperature-driven mechanisms of tipping dynamics (see more discussions on this aspect in Sects. 1.1 and 5.2).

### 5.5.3: Further research

Research explicitly of the impact of SRM on boreal forests is needed. The migration of northern coniferous forests to higher mountains and higher latitudes is creating new ecological systems that demand more research to understand their tipping points. Further advancement in the monitoring and/or prediction of abiotic (fires, drought, wind, snow and ice) and biotic (insects, pathogens, invasive species) disturbance agents and their interactions (Seidl et al. 2017) under global warming are key to predict future disturbance and resilience of both existing and expanding northern coniferous forests under novel climates of SRM.

# 6: Discussion

## 6.1 Conclusions

Our review suggests that for 9 out of 15 tipping elements considered, spatially homogeneous peak-shaving (Sect. 1.2) SRM using an SAI deployment would be at least partially effective in reducing the overall effect of their drivers, while for 4 we could not determine the sign of SRM's impact due to low process understanding (Table 1, Fig. 3). AMOC was the only tipping element where we judged SRM to possibly overcompensate the effect of climate change on the drivers (its range being partial compensation to overcompensation). For 2 of the tipping elements, the Sub-Polar Gyre and Amazon Rainforest Collapse, the effect of SRM at minimum provided no compensation for the effect of climate change. For none of the tipping elements was it expected that SRM may worsen the overall effects of the drivers, although for some their drivers were worsened (Table 1, Fig. 3). Moreover, regional heterogeneities may be significant; for example, for the Western Amazon, the overall effect was Worsening-Partial compensation, but this is less significant for overall Amazon Rainforest tipping than the effect on the Eastern Amazon, hence the overall judgement of the effect on tipping was No compensation-Partial compensation. Uncertainties are considerable to very large for the vast majority of tipping elements, particularly those where the drivers were less strongly coupled to global temperature. Moreover, our analysis has largely relied on qualitative judgement based on process understanding, so these should mostly be considered as evidence-backed hypotheses needing further research. Furthermore, our 'overall judgements' were based on our assessment of the relative importance of different drivers, and for many tipping elements this is not fully known.

Although rate-dependence effects could play a role for some ecological tipping elements and potentially AMOC, for most tipping elements the level and (for slowly-evolving systems like ice caps) the duration of drivers, rather than their rate of change, determines whether the system tips. This implies that preventing tipping would require SRM to be in place until other measures, such as negative emissions, can reduce the strength of the tipping drivers - merely slowing down the rate of warming would at most postpone tipping. Absence of rate-dependence may also imply that a "termination shock" from discontinuation of SRM would not affect tipping probability for most tipping elements.

Deliberately using SRM to reverse self-sustained tipping dynamics, once started, may be more difficult than reducing drivers preventatively, for several reasons. First, it may require stronger forcing, which may not be physically possible for many tipping elements (Table 1), or reversal may still exhibit considerable hysteresis. Second, process understanding is weaker than for drivers, making it harder to

judge the correct dose, or timing, of the intervention; in particular, reliable early-warning-signals may not be available for most tipping points. Whilst it may be possible for some tipping elements to be 'pulled back from the brink' by 'emergency deployment' of SRM soon after tipping has begun, this strategy appears risky and ill-advised. Thus, we conclude, like Lenton (2018), that such a strategy ought not to be relied upon to reduce the tipping risk, and instead we suggest that the most feasible role (if any) for SRM would be preemptive deployment preventing hitting tipping elements rather than reversal once they have been hit.

## 6.2 Uncertainties

*Physical uncertainties* for individual tipping elements were discussed in specific sections above. Some stem from limited process understanding of tipping elements involved, e.g. regarding threshold values for driver intensity and duration, the relative importance of and possible interaction between drivers, and the dynamics of the tipping process once initiated. Climate models notoriously struggle to represent tipping behaviour, partly because relevant processes and/or subsystems are not included in models, partly due to model uncertainties and biases.

SRM introduces an additional layer of uncertainty, namely, regarding its effect on tipping drivers and feedbacks. It is often possible to obtain a reasonable estimate of SRM's effect on drivers, especially if they are temperature-driven, although sometimes the drivers less coupled to temperature (e.g. precipitation in the Amazon) are much harder to predict, and introduce much more uncertainty into our estimates. Feedbacks are often even less well understood, and the estimate for the effect of SRM on these are often even more uncertain. Direct climate simulations are typically lacking, either because the tipping process itself is not well represented, or because dedicated simulations with SRM have not been performed. In some cases, proxies can be used (e.g. modelled AMOC weakening for potential AMOC tipping).

*Strategy and scenario uncertainty* arises because the effect of SRM is most likely dependent on the implementation strategy (e.g., type and location of SRM) and it's time trajectory. Our assessment is based on a spatially fairly homogeneous peak-shaving scenario, but spatially inhomogeneous cooling and associated circulation changes may have strong beneficial or adverse local impacts, while delaying SRM use may mean that some tipping points are already breached.

*Political uncertainties* are arguably the most concerning uncertainties around SRM. We will only highlight a few that might affect SRM's ability to prevent tipping - the discussion of whether a potential

reduction in tipping risk (or other climate risks) is worth incurring political risks from SRM is important, but beyond the scope of this study. Mitigation deterrence (McLaren, 2016), if it actually occurs (Cherry et al., 2023), might mean that SRM leads to higher GHG concentrations than if it had never been deployed. This could exacerbate tipping risks, especially if negative emissions turn out to be difficult, and/or if SRM cannot be sustained at the required intensity for long enough to avoid temperature overshoot. International disagreement on SRM may lead to inconsistent or inappropriate implementation that could be delayed, of variable or insufficient intensity, or include a host of local to regional measures that interact with tipping points in potentially unpredictable ways. Moreover, large scale CDR required to achieve the CO2 concentration reductions needed in a 'peak-shaving' scenario may put significant pressure on ecosystems. In those scenarios, whilst SRM may help avoid tipping in the ecosystem, the effect of the overall SRM and CDR package may be more equivocal.

## 6.3 Research recommendations

The wider climate science community will hopefully continue to work towards better process understanding of tipping, including better representation thereof in models. In the short run, a systematic assessment on (the relative importance of) tipping drivers may be helpful. Where applicable, this can be done with subsystem models (e.g., ice sheet models) if relevant processes are not included in global Earth System Models.

For many non-SAI techniques, uncertainties regarding their effectiveness and/or technical feasibility (including the time of earliest possible deployment) remain large, yet those parameters are vital for potentially suppressing tipping. The SRM community should continue to address these questions. In addition, SRM's effect on relevant tipping drivers, especially those less closely coupled to temperature, should be systematically assessed in existing and new SRM simulations.

For tipping points that are reasonably well represented in models, dedicated simulations of SRM's effect on preventing or reversing tipping should be performed. If model uncertainties are still large, strong SRM and GHG forcing can be used to explore whether certain processes are possible "in principle", whereas in the course of time, more modest and/or realistic forcing scenarios can be studied. Direct simulation of preventing or reversing tipping may not yet be feasible for tipping elements that are not well represented in models.

A challenge is the huge number of possible SRM scenarios, which may vary on background GHG trajectories, SRM method (SAI or other; possibly combinations) and location, starting year, intensity,

and so on. The choice of scenario may depend on the underlying research question, for example: Can (and should) SRM be optimised, and with which objectives? Are there low-regret options? Can (ill-coordinated) implementation exacerbate tipping risks? Communication with social scientists and stakeholders can help prioritise research questions.

Our preliminary assessment suggests that well-implemented SRM may have an overall beneficial effect on many Earth System tipping elements, although uncertainties are still very large. Whilst tipping concerns are important and ought to be a part of any assessment of the benefits and risks of SRM, such an assessment must be holistic and consider tipping concerns alongside other climatic, environmental, social and political factors that are affected by SRM.

## Author Contributions

Overall lead and coordination: GF with input from CW

Conceptualisation and methodology: GF with input from CW

Introduction: CW with assistance of GF and JG

Section 2.1 to 2.4: MA under the supervision of PI

Section 2.5-2.8: AD under the supervision of PI

Section 3: CW

Section 4: PI

Section 5: YF and JG (and GF on Section 5.2 and 5.3)

Discussion: GF and CW

Reviewing of all sections: GF

## Competing Interests

The authors declare that they have no conflict of interest.

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
