# Peer review of "The interaction of Solar Radiation Modification with Earth System"

_EGUsphere, 2023_

## Author Comment (AC1)

**Reviewer 1**

***We thank the reviewer for the helpful comments. The reviewers comments are in grey, whilst our responses are in black and italicised.***

The paper "The interaction of solar radiation modification with Earth System Tipping Elements" by Futerman et al., explores the current state of the literature of the effects of solar radiation modification (SRM) on climate tipping elements. This is a very timely review that should indeed be carried out in order to inform future research on SRM; therefore, I believe that the authors have a very good reasoning for their review article at this critical point in time. While I can see the time and effort that the authors put in this review article, I found the rationale of the article difficult to grasp and think that several changes are necessary before this article can be considered for publication in Earth System Dynamics. I apologize for my critical feedback and I hope the authors will find the comments below helpful for their decision on how to proceed.

*We thank reviewer 1 for these helpful comments. As we discuss in response to the key points, we agree with this reviewer that more clarity needs to be provided with regards to the rationale, and we believe that the edits we have made and the replies here should help with this clarity.*

Key points:

Streamlining of the article: Throughout the article, each of the tipping elements is split up into four parts: (i) Current state of the tipping element; (ii) Drivers and Feedbacks; (iii) Impacts of SRM; (iv) Further research. While I think that the core of the paper is in (iii), I do not think that another review on the current state of the tipping elements as well as drivers and feedbacks is necessary (not the main topic of this paper, is it?) or helpful (the current state of the tipping elements is done in the global tipping points report and the many other papers of this special issue). Therefore, I suggest to strongly cut those parts (maybe put the long version in an SI if the authors would like to keep this information). For instance, the section on Mountain Glaciers (2.3) had a good balance between the subsections in my view.

This is not to say that I didn't find this information valuable and interesting to read, but it distracts from the main purpose of this article (SRM) in my view.

Rationale of the article: The above (my point 1) is partially reflected in the abstract where the authors state that they review the literature on SRM but given that there is not a lot of literature they state that additionally a review of the current state of tipping elements as well as their drivers and feedbacks was carried out. I think this rationale is invalid and needs to be sharpened. In my view, a review on SRM on tipping elements is warranted even if the current state of literature is still immature and sparse in some places. This should be reflected in the abstract and the text overall. It can, however, not be an excuse to add only partially related parts to a paper because literature evidence of a certain aspect is weak.

*We agree with this reviewer that this piece should be cut down, and we have in parts cut down particularly part i each section, and moved some of this to the supplement. However, we (agreeing with Reviewer 2) believe those earlier parts are still useful, although we have clarified our rationale for doing this in response to this comment. Our revised rational may be stated as follows:*

*We review the existing literature of the impact of SRM on various tipping elements. However, because the evidence that attempts to give direct indications of this (ie explicit modelling studies) are exceptionally sparse, we attempt to make a qualitative assessment of the possible impact of SRM on these tipping elements using the literature of the impact of SRM on what has been identified as key drivers and dynamics of the tipping elements. To do this, we first review the possible drivers and dynamics to establish what parts of the earth system we need to assess the impact of SRM on (eg Amazon drying, North Atlantic Surface Ocean Temperature etc). Then, using that information, we come up with first order, evidence backed indications of what one would expect the impact of SRM to be on the tipping element by assessing its impact on these drivers, and the possibility of reversal (once tipping has started but not concluded) by its impact on the dynamics. For some tipping elements, overall judgement is feasible, as SRM may push all the important drivers in the same direction with reasonable uncertainty (for example those elements that are mainly temperature driven). For others, the impact of SRM may be different on different drivers, so either no overall judgement can be made, or a judgement with a large range of resultant impacts and a high range of uncertainty (ie those that are primarily precipitation driven) This is why part ii (as identified by the reviewer) of each section is necessary; without reviewing the drivers and dynamics, this method that attempts to use our fundamental understanding of the tipping elements to establish the impact of SRM is not possible. This is also now reflected in the table, with the impact of SRM on each individual driver stated, as well as the overall judgement; this should further clarify the rationale.*

*Given this justification, whilst we disagree with the reviewer that part ii is unnecessary, we agree that part i can be cut down. We have done this, although still have kept some information in to allow readers with perhaps less context on tipping (eg those who have come from the SRM community) to understand at least what each tipping element refers to and why it may be significant. We have done this as we hope this paper will be useful to a wide range of readers from different scientific communities who may be interested in the topic.*

Main results/Table 1: I think this is the main outcome of the paper and I like it very much (with some smaller suggestions below). This table should be placed in the beginning and then elaborated in the specific sections. This gives the reader a clear picture of the main results early on in the article instead of after 47 pages of text – I believe this table is also not announced before so that the reader could anticipate. I also strongly recommend to add at least one further figure that graphically represents the table, e.g. on a world map populated with the tipping elements and their reaction to SRM.

*We fully agree on both of these points. We have moved the table to the introduction, and have created a figure showing a world map and the impacts of SRM on the tipping elements. This figure shows our overall judgement of the impacts of SRM on the*

*prevention of tipping in each of the tipping elements (this is decomposed by driver in the table) as well as a sense of our uncertainty in this judgement.*

Overall, I believe that the manuscript would profit a lot from a clearer focus on SRM.

*We hope the increased clarification of our method in particular should help this, allowing the explanations of the drivers and dynamics to be seen as only there in service of understanding the impact of SRM on the tipping elements. The foregrounding of the table in the abstract (in response to your helpful suggestion), should further show this to the reader. We have also added in discussion of the impacts of SRM on every driver we mention in the table within the appropriate sections, further increasing the focus on SRM. The cutting of much of part i of each tipping element should also further enhance the focus on SRM.*

Some concrete points below.

Major points:

I find that section 1.3 (Solar Radiation Modification and Tipping Elements) could be streamlined strongly.

*We have edited this section to attempt to streamline it, as well as to explain what we mean by various terms of the impact of SRM (ie 'prevent tipping' 'reverse tipping'). We have added a figure to further aid with clarity in this section; this figure also may aid in further clarifying the justification discussed above.*

While figure 1 is illustrative, I am unsure whether this is really useful in this manuscript because it is not a result from this paper but a reproduction.

*We feel figure 1 is useful to the reader to understand the various feedback processes that can allow for land ice tipping to happen. If you accept that, under our newly clarified rationale, understanding the drivers and dynamics is key to understanding the plausible impact of SRM, then we think this figure may still be useful to the reader.*

In my view, this is similar for figure 2 as long as there are no studies that discuss how SRM directly alters Atlantic Ocean circulations; if there are, please add them to the figure.

*We have edited figure 2 to include the impact of SRM on the Atlantic Ocean circulations in the figure.*

Figure 3 is a general figure of S-shapes that allows for different types of tipping (forcing, noise, rate, …). As such, this figure is not specific to the AMOC and should either be removed or moved up to the introduction. It may actually be a good figure to discuss threshold-free feedbacks as opposed to different types of tipping in the introduction.

*An updated version of this figure including threshold-free feedbacks has been moved to the introduction.*

Page 12, l 322-346: Can be strongly shortened in my opinion. In particular if there are not many studies that discuss SRM, then this should be stated, and additional CDR

studies (e.g. Garbe et al., 2020) should be kept very brief as this is not the main contribution of the paper as I understand.

*This has now been shortened.*

Pages 32-35: The impact of SRM on ecological systems in general: As opposed to the other sections which are excellently referenced, this section is not. Further, my overall feeling is that this section can be condensed to around 20-35% of text lengths.

*This section has been cut down considerably.*

Section 5.2: Dipterocarp Forests: This is not a global tipping element. Why is it discussed in this manuscript? Do the authors suggest that it should be considered a climate tipping element because it is relevant on the global scale? For me, it sounds regionally very relevant but more like a super-regional regime shift rather than a global tipping element (e.g. see Rocha et al., 2018, Science: 1126/science.aat7850). Maybe this section can be moved to the SI or removed.

*The section on Dipterocarp forests has been moved to the Supplement and some of the text was clarified.*

Section 5.5: Indian subcontinent biodiversity hotspots: It is unclear to me why this section is included because (i) it is not a climate tipping element and (ii) there are also hotspots of biodiversity in Africa, Indonesia and particular in South America. Therefore, I suggest to move this section in the SI of the paper or remove it.

*We focus in the biosphere section on tipping elements (and their tipping points) in ecological systems that could well qualify as global or regional impact climate tipping elements (or we called Earth system tipping elements). We have attempted to clarify that what we mean by ecological tipping elements may be different from other climate tipping elements covered in other sections in this paper (in section 1.1: " In ecological systems, the concept of tipping elements may be somewhat different, with tipping behaviour is not only seen for large, complex systems, but also on the level of species, and events leading to species extinction can be considered a tipping point. "), which may involve feedbacks to climate, or not, or the feedbacks to climate may not be well understood. We agree that this is not a climate element, and while we agree that (fortunately) there are biodiversity centres elsewhere, this system (now called "the Himalaya-to-Sundarbans (HTS) Hydro-ecological System" in the revised section) is unusual and highly relevant to the discussion here. This comment was critically helpful in allowing us to realise that we needed to clarify what is unique about this system and why it may have the potential to qualify as an Earth System tipping element according to existing definitions. According to Lenton et al. 2008 and Armstrong McKay et al. 2022, climate tipping elements should be "at least subcontinental in scale (of the order of 1000 km, i.e., ~1 M km2) and could pass a tipping point as a result of actions this century" and they "either (i) contribute significantly to the overall mode of operation of the Earth system (such that tipping them modifies the overall state of the whole system), (ii) contribute significantly to human welfare (such that tipping them affects >~100 million people), or (iii) have great value in themselves as a unique feature of the Earth system", and "Global core tipping elements must meet criterion (i) whereas regional impact tipping elements must meet criterion (ii) or (iii) but not (i)."*

*From these definitions, we believe the Himalaya-to-Sundarbans (HTS) hydro-ecological system may be well qualified as a regional impact tipping element as it meets both criterion (ii) and (iii) as well as the subcontinental scale. If low-latitude coral reefs are considered "regional impact tipping elements" as explained in Armstrong McKay et al. 2022: "Given regionally synchronized tipping dynamics with significant human but indirect climate impacts, we categorize warm-water coral reefs as a regional impact tipping element (high confidence)", there is no reason why the HTS should not be considered. In fact, the list of climate or Earth system tipping elements is growing in recent studies and additional tipping elements are continuously been proposed driven by deeper understanding of new complex systems. Thus, we think it is valuable to use the HTS as an example tipping element in the biosphere here, even if the literature on it is sparse and we are uncertain to what degree it shows tipping behaviour. Importantly, also, we stress the speculative nature of this tipping element in the section, but nonetheless consider it an illustrative example of the impact of SRM on systems like these, as well as highlighting the advantages of the more qualitative method we have used to come up with first order indications (ie that it is feasible to establish some evidence-backed hypothesise in the absence of modelling data).*

*We have now rewritten the section to first make the case that the entire system from the Himalayan glaciers to the wide subtropical and tropical plains to the Bay of Bengal is a single integrated system and a unitary Earth System tipping element, albeit one that is underrecognized and understudied as such (we haven't found previous literature that argues for this specific point, and therefore argue for it in this section). It is integrated by connections made by water: the river networks that emanate in the Himalaya and terminate in deltas that richly sustain biodiversity. These river networks support much of the life on the subcontinent, together with the integral and integrating effects of the monsoon, that controls and nourishes human and natural systems. We realize that emphasizing separate elements of this system in the previous version did not clarify this, and we have modified the text accordingly. Like the many different forests and ecological regions of the Amazon Basin, which reaches from the foothills of the Andes to the Atlantic and encompass river networks and rainfall regimes that determine ecological drivers and feedbacks to climate, the Himalayan-to-Sundarbans system is a huge, diverse system that interacts with natural systems, human systems, and climates, and is properly regarded as an important Earth System tipping element as explained above. In contrast to the Amazon, it is barely studied as such. The Indian subcontinent is a region with growing bodies of SRM research, and as such, bringing attention to a hardly studied, potential tipping element and the impact of SRM was considered by us to be valuable enough to include in the paper.*

Comments to Table 1, which is really helpful:

*We have significantly edits table 1. It now has 4 columns. The 1st column contains the name of the tipping element. The second has the 'effect of SRM on drivers' where we decompose the impact of SRM on each of the drivers, and contain our overall judgement (and a representation of our uncertainty) of the impact of SRM on the drivers overall (ie how useful may SRM be in preventing tipping). The third column contains a description of SRM's ability to reverse tipping once tipping has started but before the system is in a*

*stable alternative state. Finally, we have a column commenting on the strength of the evidence base.*

1. Column: Overall confidence of what is meant here? Overall confidence of SRM being able to reverse tipping? Overall confidence (=agreement) of the literature on column "Can SAI reverse tipping"

*We have modified the table significantly. We no longer have an overall confidence column, instead we have a) a representation of our uncertainty in our overall judgement with regards to the effect of SRM on being able to prevent tipping (in the drivers column) and on the strength of the evidence base. This means the 'overall confidence' is decomposed further, and clarified with regards to the different things it means.*

2. Rows "Dipterocarp Forest" and "Indian Subcontinent Biodiversity Hotspots": How can SRM help once tipping is completed (as noted in "b. Likely" in column 4). Once biodiversity is lost or a forest has died back, SRM cannot help to restore these systems as they have developed and adapted over millions of years.

*Unfortunately this is true, and seems to have been an unintentional error on our parts, so we thank the reviewer for noticing this in such a crucial part of the paper. We have removed Dipterocarp from the table. As reversibility has been clarified to refer to reversal once tipping has started but before it is completed, it is possible for SRM to reverse it. However, the evidence base on reversibility for the HTS is very sparse (partially due to little modelling of tipping behaviour so far, and due to the complexity of ecological systems), and so we no longer state it is 'likely' that reversibility could occur, but rather that it is unlikely with high uncertainty.*

Section 6.1 (Further Research): I suggest to keep these sections short because for each of the tipping elements, it is already discussed where future research can broaden knowledge.

There is a number of smaller points - Minor points:

Abstract: Please write SAI in the abstract out at first occurrence
*This has been done*
Page 2, line 45: Also cite Levermann et al., 2012:
https://doi.org/10.1007/s10584-011-0126-5
*This has been done.*

On page 846 is a definition of tipping elements
Page 2, line 53-56: The definition of ecological tipping elements is unclear. In Armstrong McKay, tipping of ecosystems means a large-scale state shift of the Amazon rainforest, boreal forests, or coral reefs. The death of single species does not constitute a tipping in the Earth system sense. Please clarify.

*We agree with the reviewer here that according to the McKay22 definition, extinction wouldn't constitute tipping. However, it is important to clarify that*

*McKay22 discusses climate tipping points, and whilst we focus on climate tipping points, we do include broader earth system tipping points as well. Agreement on definitions is an inherent difficulty in interdisciplinary collaborative efforts, however, in the ecological and evolutionary literature, rapid population loss can constitute a tipping point (eg https://doi.org/10.1111/evo.13374). Whilst it is true that the scale is not that of an earth system tipping element (and we have clarified that we do only focus on the system level), we have kept a brief mention of this to highlight the difference in language use in the fields. We understand how the inclusion and current phrasing could cause confusion, and we hope that our clarification that "the ecological literature refers to tipping points not only with respect to such changes at the system level (which we focus on here), but also in to the point at which the extinction of an individual species becomes inevitable (e.g. Osmond and Klausmeier 2017)."*

Page 2, l 59: Can you directly here give an example of a threshold-free feedback? (maybe this is a good spot for figure 3)

*We agree with the reviewer, and have added figure 3 here, as well as the example of methane hydrates*

Page 3, l 79: "… has been exceeded for sufficiently long times" What you mean are so-called *overshoots*. Replace citations by

1. Ritchie et al., 2021: https://doi.org/10.1038/s41586-021-03263-2
2. Wunderling et al., 2023: https://doi.org/10.1038/s41558-022-01545-9

*This has been done*

Page 4, l 99: Irvine et al., 2019 does not exist in the reference list but only Irvine et al., 2018. Please check.

*This edit had been carried out*

Page 7, l 184, cite Levermann, Winkelmann, 2016, The Cryosphere: https://doi.org/10.5194/tc-10-1799-2016

*This edit has been carried out*

Page 13, l 369: Did the authors really mean 2089, not 2099?

*Yes, the study cited goes up to 2089*

Page 26/27: Formatting changes twice, please check.

*This has been sorted*

Page 31, l 894: The Schneider et al., 2019 paper on a 10°C warming due to cloud changes is speculative (given its huge temperature feedback). This should be stated somewhere around these lines.

*At the first mention of this temperature projection, in the previous sub-section, we now note that this was a highly idealized study. For this second mention we have edited the text to remove the specific reference to the magnitude of the warming.*

Page 37, chapter 5.3. Amazon basin: In the introductory paragraph, the combined adverse influence of deforestation, human-made fires, and climate change on the Amazon rainforest could be discussed more directly.

Somewhere in this section, the new Bochow et al., 2023, Science Advances paper should be cited: 1126/sciadv.add9973

*We have added text that includes a citation for Bochow et al (2023)*

I believe MCB (probably Marine Cloud Brightening) is only used as an abbreviation

*This has now been edited*

Page 46: I think this study by Rao et al., 2023 in Communications Earth and Environment should be discussed briefly in this section (https://doi.org/10.1038/s43247-023-00910-6)

*This has now been added, and we are thankful for the reviewer for making us aware of this paper.*

---

## Author Comment (AC2)

**Reviewer 2**

*We thank the reviewer for the helpful comments. The reviewers comments are in grey, whilst our responses are in black and italicised*

Paper summary:

In this paper, the authors present a review of scientific literature around the potential for solar radiation modification (SRM) to prevent tipping points (TPs) in the Earth system, and in particular climate tipping points (CTPs). While the risks posed by CTPs have been suggested as a motivation for considering SRM, studies on the specifics of how SRM might interact with tipping dynamics in parts of the climate system liable to tip (i.e. tipping elements/systems) have only recently started to be undertaken. In many cases no such specific studies exist, although more general research on how SRM affects feedbacks involved in tipping might be available instead. The authors collate direct or indirect studies relating to potential SRM impacts on tipping systems and their ability to mitigate CTPs, covering commonly proposed tipping systems in the cryosphere, ocean, atmosphere, and biosphere. They then assess whether the collated evidence supports SRM being able to prevent tipping in those systems to some extent, as well as whether tipping can be countered once in progress. They find that while many systems have some evidence that tipping drivers can be partly compensated for using SRM, confidence is generally low and research is currently lacking in many areas, for which future research avenues are suggested for improved study scenarios and models.

*We would like to thank the reviewer for this very accurate summary of our work*

General comments:

In general this is a timely paper, given increasing attention on the risks posed by climate tipping points and proposals that this necessitates SRM, but only a small (but recently growing) pool of targeted research in this area. Reviewing current literature and identifying gaps to target with further research is therefore a welcome exercise that will hopefully spur on more research in this area, and easily falls within the scope of this journal.

*We thank the reviewer for their confidence in the topic of the paper.*

Tipping systems and their hypothesised dynamics are in general accurately portrayed, though in places could variously do with more detail or being more concise (see specific comments for details). There are some places too where the latest papers should be included (e.g. on Antarctica) as the pace is picking up in this field now with some relevant papers published since submission. In several places tighter writing, reducing repetition, and more consistent structuring would help improve flow, which would also make the paper more concise (it's not overly long at present, but ideally it shouldn't get longer, and shorter is preferable for readability if possible).

*We have attempted to add these paper and tighten up the paper, and hope in this regards it is improved. We have also made the paper more concise, by shortening some sections, and by removing the section on Dipterocarp Forests.*

It's good to have put SRM discourse in context upfront, and the complexities around SRM not cancelling out all climate change as easily as GMST. However, I think there could be more discussion of how applying SRM in ESMs (particularly for slightly older SRM studies) that lack key tipping-relevant processes and feedbacks means we may be missing some key SRM-CTP interactions, such as the effects of SRM-induced precipitation or ecological changes. For example, ESMs are likely biased towards AMOC stability (per IPCC), lack spatial heterogeneity in ecohydrological dynamics in the Amazon (where many ESMs disagree on even sign of precip change too), or lack abrupt thaw or interactions with surface vegetation in permafrost. In some cases this could offset part of the SRM compensation of tipping, and in some cases potentially even worsen tipping risk (e.g. that one study shows worsening of Amazon droughts with SRM vs. emission scenario without SRM). To me this means the conclusion in the abstract that "*We find that SRM mostly reduces the risk of hitting tipping points relative to same emission pathway scenarios without SRM*" needs additional clarificarp that this is mostly relative to temperature, with other climatic factors that are not so well understood likely complicating and potentially undermining this risk reduction.

*We thank the reviewer for the useful stressing of this, and agree that this is not clarified enough upfront. To this end, we have made a number of changes. Firstly, uncertainty has now been stressed in the abstract after this conclusion, with words to the following effect: Where temperature is the key driver (which is the case for a large number of tipping elements), we find that SRM mostly reduces the risk of hitting tipping points relative to same emission pathway scenarios without SRM. Nonetheless, deep uncertainties remain, particularly when drivers less strongly coupled to temperature are important. Considerably more modelling is necessary before many of these uncertainties are resolved.*

*Secondly, we have added a figure (mentioned in our response to reviewer 1), putting each of the tipping elements on a map, and stressing our overall judgement of the impact of SRM on that tipping element, and a visual representation of our uncertainty prominently alongside this. Furthermore, we have now edited the table so it includes a statement of our uncertainty on the impact of SRM on each individual driver (by highlighting the range of possible effects), and importantly now has a column dedicated to the 'strength of evidence base' where we discuss the uncertainty around the evidence. In response to reviewer 1, the table is earlier in the paper and much more prominent, and so the statement of our uncertainty is as well.*

*Thirdly, we have stressed this more in the introduction and the discussion.*

Beyond the often discussed biospheric tipping systems of Amazon rainforest, boreal forests, and warm-water coral reefs, the authors also select dipterocarp forests in

Southeast Asia and various ecosystems in South Asia as potential tipping systems. However, for some examples such as dipterocarp forests, eastern Himalayas, or the Western Ghats their potential for self-sustaining regime shifts is not currently clear from the presented evidence. Additionally, it's not clear why for example the Sundarbans are highlighted but not the mangrove biome more widely, or why other biomes like savannahs/grasslands or non-coral ocean ecosystems aren't analysed. If this is because these are not covered by existing studies then that's OK but should be made clear, otherwise it seems a few examples of highly biodiverse ecosystems have been picked and the possibility of tipping in them speculated. I suggest being clearer on reasons for inclusion and their potential tipping dynamics, or adjusting review framing to be explicitly about whether SRM can prevent tipping dynamics plus biodiversity loss in some selected examples. The latter could be a whole other (and interesting) paper though, so I would suggest focusing on where tipping dynamics can be shown to be possible or likely.

*The section on parts of the Indian Subcontinent has been rewritten (see response to Reviewer 1) and renamed the Himalaya-to-Sundarbans (HTS) Hydro-ecological System, to emphasize that this is a single, integrated system from the Himalayan glaciers to the Sundarbans in the Bay of Bengal. The HTS system well qualifies as a (very uncertain) example of regional impact tipping element, following the same rationale behind defining low-latitude coral reefs as regional impact tipping element in Armstrong McKay 2022, That is, they both have significant human but indirect/regional impacts (please see the response posted to Reviewer 1 for more details). Because the natural and human systems within the HTS are interconnected by water and temperature, they are highly vulnerable to climate change induced tipping points, and to SRM potentially pulling them back from tipping points or leading to unpredicted state changes. However, while highly plausible, there is little data to support this whole system perspective, because this enormous system is greatly understudied as a unified system vulnerable to climate tipping points. We think that making this point may possibly result in greater attention, particularly given the proposed significance that some has suggested for the states on the Indian Subcontinent in shaping the future of SRM (as envisioned, for example, in Stanley Robinson (2020) and scenario work that one of us (Futerman) carried out in 2022-23). We think that a single section at this point, given the dearth of data, may be sufficient to stimulate more work on this, but it is probably insufficient for an entire paper.*

*The relevance of the Sundarbans system to other mangrove forests has now been added, but it is not just a random example. As the largest mangrove system in the world, and arguably the most biodiverse, it is worth bringing attention to this specific understudied and underrecognized system, and given space constraints, we chose for this to be the mangrove system example we use. It is now discussed in the context of the larger system of which it is an integral part.*

*We agree that grasslands/savannahs and other ecological systems are vulnerable to climate change and land use change induced tipping points and that understanding the potential of SRM to exacerbate or ameliorate climate tipping in these systems is important! But we had to stop somewhere, so, we chose some very familiar and*

*well-studied systems, and a less-familiar one to stimulate thinking, awareness, and hopefully further study.*

Specific comments:

Line no.

12-13:   Is preventing CTPs considered a key benefit of SRM that often in literature? My reading of literature is that it has increasingly been suggested in some papers/proposals, but I'm not sure preventing CTPs is so often made central yet.

*It is true that it is not central, although it is being increasingly suggested. This has been amended to reflect this fact.*

16:      SAI not yet defined.

*This has now been changed*

20-22:   Given the large uncertainties include non-temperature factors such as precipitation changes potentially countering or even cancel out tipping-compensation by SRM (particularly for biosphere, where hydrological effects makes SRM very uncertain and e.g. some experiments show worsening of Amazon droughts with SRM), I think this statement could do with some clarification. To me, this review shows that while direct SRM-CTP studies are few or missing for many systems, where studies have been done it appears SRM can reduce tipping risk specifically with respect to regional or global temperature (with high uncertainties), but current model limitations mean SRM's influence on tipping dynamics via other climatic factors could reduce or even nullify risk reductions.

*We agree with the reviewer, and thank him for this clarification. As mentioned, we have edited this to better clarify and foreground the uncertainties. With respect to the last statement the reviewer makes, it is important that we apply the implication of uncertainty symmetrically; just as the high uncertainty due to  our poor quality of evidence could mean that the risk reductions from SRM are reduced or nullified, it also could mean they are enhanced. Moreover, whilst we agree with the reviewer that there do exist tipping points where this could be the case (or at least it is possible this is the case), for the majority of tipping points, SRM doesn't seem likely to push any key driver in the 'worsen' direction when compared to anthropogenic climate change. So whilst reducing risk reductions from the temperature effect seems plausible, for most elements nullification seems unlikely (of course, if SRM is only partially effective, it may remain easy for human impacts to tip the tipping elements, even if SRM made such tipping harder).*

29:      AM22 is specifically about climate TPs rather than wider Earth system TPs, which is likely the case here too given regional/global temperature trigger focus.

*Whilst we do mostly focus on climate tipping elements, the hope was to expand this slightly to some of the biosphere tipping elements that, whilst having climatic drivers, may or may not even mainly be driven by climatic factors. Whilst AM22 does deal with*

*the biosphere, we hoped to explicitly treat the biosphere in its own terms, rather than solely as a CTP.*

50:     I think self-sustaining or self-perpetuating is more accurate - the changes can be quite steady for a long time rather than accelerating in some cases (e.g. ice sheet collapse).

*This has been amended.*

55:     Is the TP here the extinction itself or events leading to it as TP? I'm not sure the former could be seen as a TP, more a general threshold response, as there's no clear self-sustaining change dynamic there (beyond maybe a population bottleneck / functional extinction effect in final stages). Extinction can certainly lead to ecosystem regime shifts though, which can be a TP if the regime shift is self-sustaining in some regard (e.g. such that hysteresis occurs).

*The reviewer is correct that it is the rapid population loss preceding extinction (beyond which extinction may be near inevitable) that is generally considered the tipping point, as per Osmond and Klausmeier (2017). This study does highlight that such rapid population loss has been considered a tipping point in the ecological/evolutionary literature. Of course, the scale of these tipping points is much smaller than what we are concerned with, and we simply wished to note this discrepancy between the broader understanding of TPs in the ecological literature with the use of it in the climate literature. The wording of the section has been amended to highlight this. Of course, extinction could also precipitate tipping as well.*

73-75:   This is a key point (along with e.g. lines 143-146), and links to my hesitation elsewhere about confidence in SRM's compensation for CTPs being somewhat temperature-centric.

*We agree with the reviewer here. We also hope the visual uncertainty representation that will be in the figure (containing a world map) will help to foreground this uncertainty.*

79:     A more specific reference here for tipping overshoot would be Ritchie et al (2021) [https://www.nature.com/articles/s41586-021-03263-2].

This has been added.

105-107:            Good to have put SRM in wider context upfront.

*Thank you for this remark.*

113-114: Presumably there's not so much research out there for attempts at regional cooling.

*These do exist, including the recent regional marine cloud brightening study (https://doi.org/10.1029/2023GL104314) which we now briefly discuss at around this point in the paper; however, we chose to mostly focus on global SRM deployment schemes as these are the more commonly discussed.*

115: Is "resembling" here these references, or is this a sentence fragment?

*This is related to the references; the brackets have been changed to make this clear*

183: "Greenland Ice Sheet " would be clearer than "Greenland tipping element".

*This has been edited*

200-203: I think this would be better rearranged so explanation of rebound comes first.

*We have rearranged this paragraph to explain rebound before going on to explain how it may counteract surface lowering.*

208-210: I suspect the ice sheet components in the models likely used for these studies were lacking in terms of representing enough ice sheet dynamics to capture tipping, which is a limitation (and taps in to my general comment that SRM studies using ESMs that lack key tipping dynamics won't be getting the full picture of how SRM effects tipping dynamics).

*Yes, most of these studies use simple ice sheet models. Irvine et al 2009 use a fully coupled AOGCM for the climate scenarios, but the ice sheet model is relatively simple and doesn't capture fast flowing ice streams or calving in great detail. Moore et al. (2010) do not use an ice sheet model, but a linear model to model sea level rise using radiative forcing. Applegate and Keller (2015) include more feedbacks than the previous studies but also uses a simple ice sheet model.*

227-228: Specify by 2100 – much more sea level rise beyond this even on low pathways.

*This has been specified.*

230-231: Recent research from Li et al (2023) [https://www.nature.com/articles/s41586-023-05762-w] would be a good reference inclusion here.

*This has now been added.*

232: "currently driven" – air temperature could become more important in future, depending on circulation changes.

*This has been amended*

238:     East Antarctica is conventionally split in to marine basins and land-based, as they face different dynamics that lead to quite different tipping thresholds and timescales.

*We have not split up East Antarctica specifically here to keep the section shorter. A little more detail is given on the marine basins of East Antarctica being affected by marine ice sheet instability. Land based tipping points of East Antarctic are not discussed here again to shorten the section as generally it is quite stable and the threshold is high for a tipping point.*

240-243:              This sentence is a bit convoluted to me, consider rearranging.

*This has been reworded*

251-252:              Missing here is the sense of where the tipping point might be, which is generally the point at which retreating grounding line reaches main retrograde slope leading to accelerating discharge (subject to buttressing, pinning, etc.).

*We have clarified where the tipping point is within the MISI mechanism.*

276-278:              Probably a bit higher warming needed for East Antarctic basins than WAIS based on e.g. Garbe et al. 2020 (more like 3-6C for Wilkes land for example, though I'd agree the risk starts growing from 2C).

*We have clarified this to say that 2C is for West Antarctica and that East Antarctica is more like 3C+*

287:     A key reference on issues around MICI is Edwards et al 2019 [https://doi.org/10.1038/s41586-019-0901-4]

*This has been added.*

290:     Clearer to say "both are preceded by" ice shelf disintegration.

*This has been edited to say the above.*

209:     32cm sea level rise by when? It would also be useful to put these values in context by stating WAIS total sea level equivalent (3+m) earlier.

*We have added some context earlier to give WAIS's total sea level equivalent, but the line referencing the 32cm sea level rise has been removed due to restructuring and shortening of the section.*

308:     Given e.g. Goddard et al. 2023 is discussed below and there are some systems with fewer studies, "relatively few" seems more accurate now. Another recent release to consider in this section too is Sutter et al. (2023) [https://www.nature.com/articles/s41558-023-01738-w].

*This has been amended and Sutter et al added.*

309-310:            Presumably this result is specific to surface air temperature – useful to specify this in context of limited short to medium term impact on ocean warming.

*Yes, is specific to surface air temperature, this has been clarified in the text. The fact that cooling surface air temperatures does not have a large impact on the ocean in the short to mid term (ocean intertia) is discussed in this subsection.*

326:    Garbe et al. (2020) is not specifically a CDR experiment – might need to rephrase above for clarity.

*This has been rephrased and moved earlier in the subsection, as part of a shortening of this section to reduce the content on CDR, as it is not the focus of the paper.*

329:    True, but how close it recovers by present day levels depends on experiment type (equilibrium vs "quasi-static"). Also, do you mean current warming of ~1.2C or pre-industrial by present levels?

*We have cut most of the CDR section to shorten as these experiments are not the main focus, but agree that recovery does depend on equilibrium vs quasi static experiments. Though both experiments show the ice sheet can't recover to present day, it is worse under quasi static than under equilibrium.*

334-335:            Feels like this should be discussed earlier in this subsection given it's such a critical point, before moving on to what changing SAT might be able to do as a second-order driver and then what reversal/CDR studies show.

*Agreed, this has been moved to the first paragraph of the subsection.*

358-359:            Could probably do with a little more detail on and beyond melt-elevation and melt-albedo feedbacks here (these two can signpost back to ice sheets for more info), e.g. role of changing snow patterns and black carbon/dust on albedo, thermokarst interaction, slope instabilities, negative feedbacks from retreat to higher altitudes or debris insulation, etc., as well as that unlike ice sheets these feedbacks happen on a largely local to regional scale.

*We have included the point that due to feedbacks happening on a more local scale, glaciers are more sensitive to climatic changes and have additional smaller scale drivers. We have also added in information on black carbon/dust when we discuss the melt albedo feedback in the Greenland section. We have not gone into detail on these smaller scale processes given limited space, and because many of these are small-scale processes that have a more limited impact on the glacier as a whole, and can differ widely between glacier regions. We are more focused on the large-scale. We have chosen not to include retreat to higher altitudes as a negative feedback as though it can stop the glacier from retreating any further, it doesn't necessarily lead to glacier readvance. Additionally, usually retreat would have to be quite substantial to reach*

*altitudes high enough for warming from climate change to not be the dominant influencer.*

368-369:          Are G3 and G4 defined before this point (for readers not familiar with GeoMIP terminology or context)?

*G3 and G4 are not defined before this point and so we have briefly defined them here.*

375-377:          A Himalayan example is given here, but can anything be said so far about how heterogenous SRM precipitation impacts may affect different glacier regions, e.g. from GeoMIP? If not, then can be explicit in that.

*We have explicitly said that there does not appear to be any studies beyond the Himalayan region.*

397:   SAM not defined before now.

*This has now been defined.*

422:   Specify what sort of temperature, e.g. surface air, sea surface, regional/global etc.

*Edited to be more precise. Sentence now reads: "On decadal time-scales, Arctic sea-ice area has declined linearly with the increase in global mean temperature over the satellite period in all months (Notz and Stroeve, 2018) and climate models project a continuation of this relationship over the 21st century (Niederdrenk and Notz, 2018)"*

447-448:          Could clarify here that while some positive feedbacks exist that have been suggested could drive tipping dynamics (e.g. https://link.springer.com/article/10.1007/s10584-011-0126-5; https://journals.ametsoc.org/view/journals/clim/34/11/JCLI-D-20-0558.1.xml) the explanation given here is generally more favoured.

*Agreed, we have added reference to Hankel and Tziperman (2021) here.*

476-479:          Presumably this is relative to a scenario of continued warming without SRM, rather than versus no further warming in above example? (otherwise this statement could be seen as conflicting with a reduction in March extent above.)

*Yes, we have added the phrase " relative to the background warming scenario without SRM" here to clarify this.*

485:   Missing "which" after citation

*This has now been edited.*

498-500: I think this could do with clarification - compared with continued warming on same emissions pathway but without SRM?

*Agreed, and clarification added. Sentence now reads: "Additionally, there has been little work (Ridley and Blockley (2018) is a notable exception), assessing the different impact of SRM versus avoided emissions on Arctic and Antarctic climate and sea ice under SRM, at a given global mean temperature. Such assessments would aid in making a fully quantitative statement on the effectiveness of different SRM strategies for sea-ice restoration (Duffey et al., 2023)."*

512: Clarify as (presumably) soil temperature.

*The value, from Biskaborn et al. (2019), refers to the permafrost temperature at the depth where there is zero mean annual amplitude in temperature (defined as <0.1°C), as measured in boreholes. We have cut this sentence from the paragraph as distracting from the main message.*

523: Perhaps useful to clarify this as melt of ice blocks/wedges, and add thermokarst/talik development (which is not necessarily ice melt dependent).

*Agreed. We have expanded this sentence to give a broader explanation, as below: "..to abrupt thaw, which refers to deep thaw occurring on rapid timescales of days to several years due to processes such as the physical collapse of the surface caused by ice melt and the formation of thermokarst lakes (Turetsky et al., 2020; Schuur et al., 2015)."*

536: Perhaps worth clarifying that in AM22 this was an additional threshold beyond the more general (and higher confidence) widespread localised abrupt thaw above ~1.5C.

*Additional sentence added to clarify this: "This potential global tipping point is in addition to the widespread occurrence of localised abrupt thaw which could occur at warming above approximately 1.5°C (Armstrong McKay et al., 2022)"*

549: Missing in this section (reflecting the point made under further research) is a discussion of how current generation models don't capture non-gradual thaw processes, so currently these conclusions primarily relate to gradual, non-tipping thaw only, or how changes in e.g. precipitation interact with abrupt thaw or eco-hydrological processes.

*Sentence added to highlight this point at the start of the 'further research' section. "The permafrost response in ESMs does not include the feedback processes leading to abrupt thaw and local tipping behaviour (Turetsky et al., 2020), so the quantitative assessments above principally apply to the gradual thaw component; further development of ESMs to include such processes would allow more robust quantitative assessment of the impact of SRM (Lee et al., 2023)."*

580-581: SRM is also unlikely to reverse abrupt thaw processes like thermokarst/talik development once started, but can limit formation of new taliks.

Sentence added to reflect this, above (after 578 of first submission): "Similarly, while SRM might reduce the onset of localised abrupt thaw processes, it would be unlikely to reverse these processes once begun."

604: Could specify very long timescales is necessary because of the centennial-millennial timescale of ocean heat uptake and circulation.

*We have added this specification as suggested. The sentence now reads: "The reduction in surface temperature under SRM, if maintained over the very long (multi-centennial) timescale of deep-ocean heat uptake."*

616: A useful figure, but why no equivalent for cryosphere or biosphere systems? A generic schematic showing e.g. elevation/albedo feedbacks might help illustrate key feedbacks there. There could also more focus specifically on how SRM might intervene in those feedbacks to make figures more specific to this article (rather than textbook-style background).

*We have added the effect of global warming, and effect of SAI on drivers.*

622: Deep convection should also be marked in Southern Ocean, as it's not separate to sinking there.

*This has been done.*

628: Might need to clarify deep convection/sinking, which is illustrated separately in figure but are really part and parcel (with "sinking" a simplification of deep convection).

*As shown by Katsman in several papers, convection is not the same as (net) sinking: Convection being a mixing process with some water parcels moving up and some down, close to each other, but no net movement, whereas sinking is understood as the net downward movement of water. Sinking requires the proximity of a coast to break geostropic balance (see also Spall 2001), as opposed to (deep) convection which can take place in the interior. Therefore we depict them as separate processes, also in the figure.*

640-641: Relevant here is the IPCC's assessment of collapse unlikely before 2100, but also that CMIP models are biased towards stability and lack key aspects (e.g. ice sheet meltwater) that likely lead to under-estimates of weakening and collapse. This is why unrealistic hosing is used in some studies (like Jackson et al. 2023 below) to overcome this tendency. This is mentioned in next paragraph, but think it's useful context within this paragraph.

*This paragraph has now been moved to the supplement, so we feel repeating the context there is likely no longer relevant, as the reader of the supplement is likely to have read the paper (and thus the mentioning of this) before reading the supplement.*

652: Jackson et al. 2022 is the preprint, can update to final 2023 paper now.

*Thank you, corrected.*

660:    Liu et al 2017 relevant here too
[https://www.science.org/doi/10.1126/sciadv.1601666]

*Thank you, reference added.*

662:    Another angle beyond palaeo and model evidence not covered here is observations, with the debate over current slowdown (with IPCC AR6 having low confidence in some weakening) and of potential early warning signals suggestive of destabilisation [e.g. https://www.nature.com/articles/s41558-021-01097-4; https://www.nature.com/articles/s41467-022-32704-3; https://www.nature.com/articles/s41467-023-39810-w]

*Good point, we included some studies on observational evidence - however, also some studies which criticise some of the papers mentioned here. The part ended up in the supplement.*

677-678:            Ice sheet runoff is quite the important missing factor though, which is missing from most ESM simulations of AMOC weakening/collapse (and therefore SRM studies using those too).

*A study on the effect of ice melt on AMOC weakening was already mentioned a few lines further down. We have added a sentence to mention that ice melt could also increase tipping probability.*

700:    One way of exploring possible SRM effects on AMOC would be to explore the role of past/current aerosol emission effects on AMOC, which have been suggested to generally reduce AMOC tipping likelihood (and is being partly reversed over North Atlantic over recent years) and so could effectively be a smaller-scale SRM analogue.

*One paper doing this is mentioned in the next paragraph already (Hassan et al). We expanded this comment to clarify that according to that paper aerosol forcing indeed contributed to AMOC (≈1.5Sv).*

705:    There's a question mark over whether precip changes from SRM would help or hinder though (though admittedly it seems to be a small factor, model limitations permitting).

*We have now clarified this in a rewritten section on the impact SRM on drivers, that 1) it seems as if the impact of SRM on precip - evap is helpful (i.e. less freshening; see Xie et al., 2022) and 2) precip-evap is not the dominant factor.*

721-723:            This is a key point, and relates to my general comment on uncertainty inherent in using models with limited tipping dynamics representation to project SRM effects on CTPs.

*Thanks for the remark. No direct action taken at this point of the text.*

728-729: This is a very good point, and may be applicable to a other tipping points too, as while many have generic warming level thresholds estimated several are likely to have rate-dependent too [e.g. Ritchie et al. 2023 https://esd.copernicus.org/articles/14/669/2023/]. Could be something to highlight need for further study of in discussion.

*Thanks for the remark. No direct action taken at this point of the text.*

733-738: This paragraph largely repeats previous one.

*Agreed, duplication was removed.*

748-749: Indeed – a subset of this risk is that if SRM was initiated to protect the AMOC but it turns out tipping has already commenced (as exactly when thresholds are crossed is uncertain and there's a time lag in tipping dynamics) then SRM might have to be phased down or out to avoid over-cooling North Atlantic region, but at potential cost of unprotecting other regions and CTPs. Relates to potential tricky balancing of how to target mitigating multiple CTPs and other climate impacts without worsening some over others.

*This is an interesting point and a brief remark has been added.*

761: Missing parenthesis after "injection points".

*This has been corrected.*

808: Given similar issues to AMOC could merge with further research there for a clearer structure, in a similar way to how land ice further research is grouped together.

*We have decided not to merge as it seemed to overcomplicated things. However, we have referred to AMOC section for overlapping matters and treated here only the non-overlapping ones*

823: Missing fullstop before citation?

*Indeed, thank you.*

831: Relevant here is new paper Li et al (2023) [https://www.nature.com/articles/s41586-023-05762-w] which projects strong weakening with further warming (RCP8.5 in that study, but likely similar magnitude decline for RCP4.5 on longer timescales)

*Indeed this paper is very useful and has been added.*

834: Sentence would be clearer if what the mechanism is for is specified (overturning weakening? collapse?)

*Corrected (note: the whole paragraph has now been rewritten)*

839:    Contrast Li et al, who suggest wind has marginal effect on projected weakening (though possibly model dependent).

*Li et al is now discussed.*

862:    Only marine stratocumulus clouds are tackled in this section, and not e.g. monsoons which are sometimes considered atmospheric TPs, though possibly more due to aerosols than climate change proper. Given the relation of monsoon tipping to aerosols, and recent work on interhemispheric AOD difference as a key safe boundary (Rockström et al. 2023), their relation to SRM would be an interesting avenue to explore if resources permit.

*Given the complexity of this topic, doubts about whether it constitutes an atmospheric tipping point, and most importantly limited space, we decided to exclude monsoons from this paper. As mentioned elsewhere, this paper is not a comprehensive review of all possible tipping elements, but just a diverse range that we thought may be considered useful.*

894:    Warming values repeated from past paragraph (not bad, but unnecessary to make hysteresis point).

*The reference to specific values has been removed and the sentences rephrased.*

920:    What does this mean - the feedback is fully counteracted, or literally turned off?

*The text now clarifies that this is a highly idealized model and that this feedback was literally turned off in the model.*

948:    This general intro to ecosystem TPs & SRM is useful in general, but is repetitive in parts and could be tightened.

*The introduction has been clarified in places and shortened.*

953-954:            In my experience extinction is not normally thought of as a TP / regime shift in ecology (although extinction or replacement of a keystone species could help trigger a wider regime shift).

> *In the ecological and evolutionary literature, rapid population loss can constitute a tipping point (eg https://doi.org/10.1111/evo.13374). Whilst it is true that the scale is not that of an earth system tipping element (and we have clarified that we do only focus on the system level, and so won't be focusing on the more micro-scale tipping elements), we have kept a brief mention of this to highlight the difference in language use in the fields. We understand how the inclusion and current phrasing could cause confusion, and we hope that our clarification that "the ecological literature refers to tipping points not only with respect to such changes at the system level (which we focus on here), but also in to the point at which  the extinction of an individual species becomes inevitable (e.g.*

*Osmond and Klausmeier 2017).” is useful. Of course, we also agree with the reviewer that wider systemic tipping can also be initiated by such extinction.*

965-967:              Clarify – the impact/event itself was not a TP (just an abrupt exogenous forcing with an equally abrupt response), but it likely drove many localised/regionalised ecosystem regime shifts (some but not all of which might have featured self-sustaining tipping dynamics).

*This has been clarified, the original wording was sloppy language.*

1010-1013:          This sentence can be simplified for clarity.

*Rewritten to clarify the meaning. More sloppy language, sadly, but hopefully better now.*

1024:  To me this is conceptually confused – "forests" and "biodiversity" are very different categories of things (one a collection of objects forming a system, the other a property of such a system).

*Thank you for pointing this out, it has now been deleted.*

1025-1030:          These sentences are a bit confusing and could do with tightening.

*Thank you, these sentences have now been shortened and rewritten.*

1051-1052:          I'd like to see more evidence presented on how these forests are susceptible to tipping (via regeneration failure) that justifies pulling it out as a specific unit to analyse. In analyses of tropical rainforest-rainfall feedbacks, rainforests in maritime Southeast Asia are normally found to be more robust due to plentiful ocean rainfall sources, so it can't be that mechanism, and synchronised seeding occurs in many forest biomes, particularly tropical but including to some extent temperate too. Numata et al. is a key citation here, but it's not apparent that regime shifts per se are projected in that study. An alternative would be too merge in with discussion of other tropical forests.

*This section has been moved to the Supplement, and the text modified. Failure of a foundational/kestone/dominant group of species due to climate change would be likely to result in a fundamental state change of the system, although data are admittedly limited.While these forests may be robust to some climate changes, the massive deforestation, increased fires, and other harms may make them more susceptible to climate-induced tipping. We agree that the evidence is limited, but think these highly diverse and less well recognized systems should gain attention, and therefore hope to retain this in the Supplement.*

1073-1075:          I couldn't find this in the given reference.

*This has been added*

1078:  Clarify where.

*This sentence has been revised to: SRM is predicted to reduce global mean temperatures and create drier conditions (MacMartin et al., 2016) including in southeast Asia (Tan et al. 2023).*

1082-1083:          Assumes there is a TP in this system – if this is uncertain though could reframe this section as about protecting a particular biome that may or may not feature tipping dynamics (but then a question is why this ecosystem and not various others).

*We have followed this suggestion and moved this to the supplement, as well as briefly discussed it with regards to tropical forests more generally. We picked this biome because it is under explored and may (or may not) have unique sensitivity for tipping, and hoped to highlight that it may be possible to have some knowledge of the impact of SRM on systems that may or may not tip, even if tipping is not known (such knowledge may be much harder to get if we were just reliant on models.*

1086:  Missing a "Further Research" subheading here?

*We have added this*

1097:  This surprises me, as Amazon is normally considered to be one of the most biodiverse biomes (when considered as a biome unit). The cited analysis doesn't highlight Amazon because the "hotspots" are a product both of endemism and threat level, with Amazonia seemingly left out because of lower threat rather than lower endemism (i.e. hotspots in Myers et al 2000 don't mean that's where most biodiversity is, but where areas of high biodiversity are most threatened – it also excludes invertebrates, and any newly identified species in past 23 years). Also, this citation is not in the reference list (I'm not sure it adds too much to discussion anyway).

*We agree with the reviewer and have removed it as it is tangential.*

1101-1102:          2-6C is for just climate change without anthropogenic co-drivers like deforestation – including those could lower it further.

*Text modified to make this point.*

1104:  More recent studies highlight that degraded forest is a likely alternative state (possibly more common than savannah).

*Agree. We have rephrased this part: "...might force a tipping point for the Amazon basin to the replacement of tropical forest with systems without trees or with fewer, scattered trees and without continuous canopies." to cover both degraded forest state and savannah-like non-forest state.*

1123-1124:    Some discussion of where dieback is more likely – in the drier south and east, as per bistable areas in Staal et al. 2020 – would be useful too for understanding the feedbacks and areas most at risk.

*Thanks for the suggestion. We have added this sentence and the reference: "Staal et al. 2020 further delineated a bistable state of forests in the southern Amazon, which are most susceptible to the drought-dieback feedback loop that would tip these forests to savanna-like nonforested state."*

1143:  Could make a link here to AMOC collapse too, which is projected to shift ITCZ southwards and so cause drying in Northern Amazonia (likely similar to in Younger Dryas). Also some link to monsoon shifts and global aerosol patterns.

*Thanks for the suggestion, but recent work has suggested  the possibiity of the opposite outcome: A potential collapse of AMOC would stabilise eastern Amazonian rainforests by increasing rainfall and decreasing temperature in most parts of the Amazon (e.g. Da Nian et al. 2023). This may reflect the difficulty of GCMs in predicting regional climate responses of the Amazon. We think it is helpful to mention potential competing/controversial effects of climate change and SRM on the tipping dynamics of the Amazon: "Although the most direct drivers of potential tipping dynamics in the Amazon is due to climate change, some changes in oceanic and atmospheric circulations could also have indirect, beneficial effects on the resilience of Amazon forests. For example, the possible AMOC collapse with elevated warming (see section 3.1) is projected to shift the Intertropical Convergence Zone southwards (Orihuela-Pinto et al. 2022) and cause increased rainfall and decreased temperature in most parts of the Amazon, which would stabilise eastern Amazonian rainforests (Da Nian et al. 2023) by mitigating the above-mentioned drought-dieback feedback loop. Although large uncertainties exist in regional climate predictions, potential competing effects of SRM on the Amazon forests are foreseen due to the complex roles of reduced global warming and prevented AMOC collapse in the regional climate of the Amazon, which warrants further studies."*

1150-1151:    Even here GCMs often struggle to agree on even the sign of regional precip change, which is one of the reasons ESMs tend to disagree on likelihood and scale of Amazon dieback.

*Agree. We have rephrased this sentence to make this point and also mentioned large uncertainties in regional climate predictions in the Amazon elsewhere.*

1165:  "the utility... is of limited utlity" – rephrase to e.g. "Thus, existing studies...".

*This has been rephrased.*

1194:  I think representing land use change, and spatial heterogeneity in plant traits adaptivity across the Amazon is also key for better model representation.

*We agree with the reviewer and have added this.*

1205-1206:        This probably needs a citation [e.g. https://journals.plos.org/plosone/article?id=10.1371/journal.pone.0025026]

*Thank you, this has been added*

1249:  Critical too in some regions like the Caribbean is disease and invasive species spread (often facilitated by warming and globalised trade).

*We agree and have added a sentence about this: "Moreover, diseases (Alvarez-Filip 2022) and invasive species (Pettay et al. 2015), often associated with warming and global trade, also have negative impacts on the structure, functioning and stability of coral reefs such as those found in the Caribbean."*

1268:  Some numbers here would be useful to demonstrate this if possible.

*Rephrased and a citation is added to show the cancellation effects between direct and diffuse radiation changes induced by SRM*

1278:  Recurrence in between hurricanes?

*Yes*

1291:  While the basis for SRM effects on coral bleaching is better understood than for some tipping systems, given the fair few uncertainties explored above relating e.g. to extreme events, complex co-drivers, or variation in SRM impacts between different coral regions makes it seem not quite so strong to me. Not a huge amount of dedicated studies are cited above either. Personally I'd like to see studies with up-to-date ESMs exploring where and how much SRM is necessary to e.g. keep below specific Degree Heating Week or recurrence thresholds, how this affects storms, and the spatial heterogeneity of these.

*We mostly agree with the reviewer, although do think it is useful to contrast the degree of knowledge we have around coral reefs (where we can have decent certainty that SRM would to some degree reduce tipping) compared to most other systems in the biosphere, where our uncertainty is far larger. Nonetheless, we have added words to the effect that the reviewer has suggested, as we do agree more research could be useful.*

1303-1304:        Are these are all vulnerable to tipping? Not all ecosystems have alternative stable states to tip to – they can just become gradually degraded – so it's important to present or cite evidence for self-sustaining regime shifts. The Sundarbans are the clearest example here, with coastal erosion and salinity effects likely to drive tipping dynamic across mangrove biome (GTPR, 2023), but given they're at risk of regime shifts across the tropics why is this the only locality discussed?

*Permanent state change to a "degraded" system, with different dominant functional types of vegetation, fragmented and populated by invasive species, is a common stable end point of state change in many ecological systems. We believe that this change to a profoundly different, stable state fits the definition and concept of a tipping point to an*

*alternative stable state. However, we do except your criticism that we don't highlight what this stable alternative system is, partially because there has been little study of what we now refer to as the HTS in the context of tipping, so the literature base to make such judgements on are rather limited.  A rationale for discussing the Sundarbans as part of a larger, integrated Earth System tipping element (the HTS), is discussed earlier in this response, and we hope that justification is satisfactory. Because mangroves are globally dispersed and affected by many factors, focusing on the largest and most diverse such system as an example seems useful to us, just as we and others use the Amazon as a critically important example and do not just discuss all tropical forests as a single Earth System tipping element.*

1311:  Here as in above in text?

*We have clarified we were referring to the mountain glaciers section.*

1315-1317:          Maybe better discussed in SRM effects subsection below?

*We have now moved it to that section*

1329:  In other subsections the drivers and mechanisms of a specific tipping system is analysed, while here it's for multiple different systems within the same region, which is somewhat inconsistent in paper structuring. Additionally there's some overlap with other sections, e.g. for glaciers.

*We agree with the reviewer here, and have now reframed the Himalaya-to-Sundarbans (HTS) Hydro-ecological System as a single, integrated system which qualifies (with high uncertainty) as a regional impact tipping element (see responses above). When we now describe the drivers and mechanisms of the HTS tipping system, we consistently refer to the whole HTS system or an integral part of the HTS system. There is very minor overlap with the Moutain Glaciers section, and this example focuses on the hydroecological aspect, especially the ecological, agricultural, and human systems that depend on the glacier water and monsoon. We believe all other ecological tipping elements such as the Amazon forests and the tropical coral reefs are of similar complexity, involving different subsystems within the whole system.*

1334-1336:          Specify what system this tipping point is in – downstream ecosystems? Peri-glacial environment around glaciers?

*We have specified the immediate area below the glaciers as well as the downstream Ganges-Brahmaputra-Meghna basin that are subject to tipping driven by Glacial melting.*

1343-1353:          This paragraph reads rather fragmented to me. Additionally, I don't think the TPs on line 1345-1346 have been explicitly described yet.

*We have largely revised this paragraph of section 5.4.2 The impacts of SRM to make it clearer. We have also added a few sentences and references regarding specific numbers of tipping points (in terms of degC warming) related to extreme heat and glacier melting specific to the HTS region.  We cannot however find specific numbers of tipping*

points for all subsystems in the HTS due to the scarcity of research establishing this unique system as a tipping element (which is one reason we introduce it to increase awareness and stimulate further research).

1374-1378:        Possibly worth noting that this hypothesis (or at least magnitude of effect) is debated.

*Agree, we have revised this part to make the point: An intriguing but hotly debated hypothesis has suggested that the extinction of large mammals (e.g. woolly mammoths) was a tipping point in the most recent glacial maxima in which their grazing maintained grasslands which had higher albedo than the coniferous forests, resulting in global cooling because the extent of these systems is so great; other controversial suggestions includes wildlife restoration as a way to reverse that tipping point (Zimov, 2005; Schmitz and Sylvén, 2023).*

1388-1390:        Worth noting that biogeophysical effects can have opposite regional climate impacts to the global climate effect (which could potentially interact with SRM) – e.g. boreal dieback to steppe increasing albedo and regional cooling, versus carbon released by dieback increasing global warming.

*This has now been noted in the text.*

1394:  Given complex eco-hydrological mechanisms in boreal forest dynamics this subsection could do with mention of how (likely uncertain) impacts of SRM on precip might complicate temperature-based considerations.

*Thanks for this comment. We have added in the end of this paragraph: "Furthermore, given complex eco-hydrological mechanisms in boreal forest dynamics, the large uncertainty in simulated regional precipitation changes under SRM might complicate the above temperature-driven mechanisms of tipping dynamics (see more discussions on this aspect in Sections 1.1 and 5.2)."*

1403:  It seems that there's no specific SRM studies re. boreal forests – if so, should highlight that as a knowledge gap.

*The text has been edited to include this.*

1417-1422:        Minor point, but not sure these sentences are necessary as they mostly repeat the Introduction.

*This has now been mostly removed to avoid repetition.*

1426-1428:        Useful point to mention –potentially relevant to highlight in abstract.

*We have changed some of the abstract to highlight the limitations of the method, suggesting that " we give first-order indications of the impact of SRM." to highlight the point made here.*

Table 1:

*We have significantly edited table 1. It now has 4 columns. The 1st column contains the name of the tipping element. The second has the 'effect of SRM on drivers' where we decompose the impact of SRM on each of the drivers, and contain our overall judgement (and a representation of our uncertainty) of the impact of SRM on the drivers overall (ie how useful may SRM be in preventing tipping). The third column contains a description of SRM's ability to reverse tipping once tipping has started but before the system is in a stable alternative state. Finally, we have a column commenting on the strength of the evidence base.*

Is b) in the 4th column linked to a specific timescale? GrIS, AIS, and permafrost here are Not Reversible by SAI once tipping is complete, which is true on centennial timescales but not necessarily very long timescales, while AMOC collapse is Yes with hysteresis, when on long enough timescales that could apply to the former examples too. I'm also not entirely sure if AMOC/SPG collapse can likely be reversed once tipping has begun (depends on feedback dynamics, which we don't have a great hold on right now). Also, is "Uncertain" in last row 4th column for both a and b?

*We have clarified what we mean by reversal now in the introduction, using a diagram. Essentially, we are using the term reversal to mean reversing tipping once self-perpetuating feedbacks have started but (considerably before) tipping is completed, and don't discuss reversal once it has entered into a second stable state, due to the fact that timescales are simply too long. Exactly where the line between these two meanings of reversal falls is fuzzy, but we don't deem such ambiguity as such a vital problem, particularly when there is so much uncertainty over other factors anyway.*

1441-1442:            These partial compensations are mostly with regards to temperature though – I would hazard that greater uncertainty / lower confidence would hold on this partial compensation once accounting for impacts on other climate factors such as precipitation. At the moment the Discussion and Table 1 doesn't highlight this.

*The edits to table 1 make this clearer, and we have added a little more discussion on this as well. We now decompose table 1 into the impact of SRM on each of the individual drivers (this should also now be discussed in each relevant section), as well as an overall judgement with a sense of uncertainty. We also have a column on strength of evidence, which may also be useful at stressing this point.  Whilst it is true that the partial compensation is mostly with regards to temperature, or with regards to those tipping elements that are closely coupled to temperature (and particularly to those that are less strongly related to local precipitation), this is a large number of tipping elements evaluated. We have added in more discussion to clarify the nuances, and we hope this, combined with the edited table, would be sufficient.*

1444-1447:            These points are repeated two paragraphs down.

*This has now been edited to remove the repetition.*

1445: I don't think peak-shaving hasn't been mentioned prior to this point, so could do with an explanatory citation on first instance.

*We have now mentioned it in the introduction.*

1449-1453: This overlaps discussion of emergency use and hysteresis in paragraph below – could probably tighten discussion to be more concise.

*This has been tightened.*

1475: Could specify biosphere/ecosystem TPs here (e.g. biomass plantation based CDR making forest dieback or savannah degradation more likely).

*This has been done.*

1477: A key issue and source of uncertainty for me is that we know most about how SRM might effect CTPs via temperature, but much less about other climatic factors like precipitation, which in many cases could reduce or even overwhelm the compensation delivered by reducing warming (e.g. less precip delivered to ice sheets, or cooling accompanied by droughts in tropical forests). This relates to the research approaches given below – improving modelling of Earth system tipping dynamics is critical to be able to give a fuller assessment of how different SRM schemes might affect tipping processes.

*We essentially agree here, and agree improved modelling of Earth system tipping dynamics is critical, and have clarified this. However, we don't want to downplay our knowledge either, and want to keep the discussion of uncertainty symmetrical (just as the uncertainty over precipitation may make compensation less effective, it also could make it more effective). At present, beyond a brief mention of this justification (which has now been mentioned elsewhere as well), we do not believe any more discussion of this is needed in 6.1*

1478-1481: This is not very clear to me at present – is this saying that these Qs wouldn't be relevant to people who think SRM research shouldn't be pursued at all because of moral hazard issues, or more the issues raised in the penultimate paragraph?

*This paragraph has mostly been removed, and replaced with a statement of the idea that tipping and other physical impacts are not the only relevant factor to take into consideration when carrying out an assessment of SRM and SRM-related research.*

---

## Referee Report (RR1)

**Second Review of "The interaction of Solar Radiation Modification with Earth System Tipping Elements" (egusphere-2023-1753) for Earth System Dynamics**

**General comments:**

The authors have substantially revised this manuscript, tightening the writing & structure and reducing repetition throughout, and in my view, it has improved considerably as a result. In particular, the authors have caveated their summary of overall SRM effectiveness for CTPs more thoroughly, emphasising the uncertainties brought by non-temperature drivers in the abstract, introduction, and discussion (while the counter-point made that this could also make SRM more effective, not only less, is fair enough too). The additional figures and tables are useful, better helping to demonstrate tipping dynamics, how SRM might intervene in these dynamics, and the paper's results. Table 1 has also improved with the addition of more explicit discussion of drivers, effectiveness, and confidence throughout, making it easier to discern key points. Other points raised, for example the ocean temperature focus for marine ice sheets, or current general circulation / Earth system model limitations, have also been clarified, and the authors have justified their selection of systems to consider in this paper.

Beyond minor further suggestions, my main remaining comments concern the subtleties of categorising tipping in the Himalaya-to-Sundarbans (HTS) hydro-ecological system. While I appreciate the value of looking at HTS as an integrated socio-ecological system likely featuring localised tipping points, I am not yet convinced that it can be categorised as a regional/impact tipping element by the rationale of AM22 (even as a highly uncertain one) without a clear mechanism for shared tipping dynamics at the system rather than subsystem level beyond a common threshold. For comparison, warm-water coral reefs are classed as regional elements because there's evidence for a bleaching frequency threshold beyond which recovery is prevented and localised die-off becomes inevitable, with widespread mortality across the same biome/ecosystem functional group occurring at similar warming levels. Similarly, while Amazon rainforest dieback due to moisture recycling failure is also localised, it can trigger further dieback across much wider parts of the same system via that process, and is all one biome/functional group. In contrast, HTS covers multiple biomes/ecosystem functional groups, with the default likelihood being that different habitats that may tip are likely to tip due to different dynamics at different levels of climate change/degradation. It's still fine to include HTS in the paper though, but I think it either needs a little more justification as to how systemic tipping might emerge at the integrated system level, or a little more clarification that it's socio-ecological system with localised tipping, which this paper is suggesting as a potential element but is still a valuable case-study for considering SRM impacts on even if not.

**Specific comments (by line no.):**

23:          I think "could" or "is likely" rather than just "is" is more appropriate here, given most of the evidence comes from simulations.

38:          A comma has escaped.

47:          Should "reverse" be here as well (along with avoid/postpone)?

61:          "stop increasing" would be smoother than "stop to increase".

130:          "tipping elements" would be more consistent terminology than "earth systems" here, or Earth subsystems to differentiate from the Earth system as a whole.

144:          Insufficient SRM possible/used is also highly relevant for only postponing tipping.

175 / Table 1: The table is much improved, but I had to repeatedly look back at caption to remind myself of what each letter meant in the second column. A simpler approach to consider might be pluses and minuses, e.g. - for worsen, ~ for negligible, + for partial compensation, ++ for effective compensation, and +++ for overcompensation (however, that'd mean +/-s for both driver and compensation direction, so that might be too confusing...), or just abbreviations e.g. over, part, etc. Also, I assume bolding for drivers means primary drivers, but I don't think this is stated (and is missing e.g. for MSC driver or HTS driver/reversibility). I was going to ask too whether it'd might make more sense to have Table 1 plus its description (lines 194-205) after Section 5 instead, but on reflection I can see it can be argued either way which makes more sense to the reader. Finally, I broadly agree with the categorisations, though I'd query a few of the overall ones, such as whether HTS should be U-P (as the uncertain effect of SAI on monsoons could be critical), whether AR should be U-P or W-P (as evidence / GCMs remain limited, and while I agree it works better in east than west, there remains an uncertain risk of bringing west to point of bistability instead), whether SPG should be U-E (as no driver has an N or W, but there are few studies), and whether BPF should be P-E (as studies find reduced permafrost loss, but not totally countered, albeit potentially improvable via SAI strategy).

Fig. 3:        I believe the compensations here are the overall compensation judgments in Table 1 (rather than temperature alone), but would be useful to state in the legend or caption for clarity.

222-244 & 286-287:        This is indeed the IPCC AR6 summary, but the nature of confidence language makes it sound less compelling than I think the evidence suggests (especially from palaeo studies) - AR6 for example reported several studies where total loss committed at 2-3°C (ch.9-pg.78). As such, so that it doesn't sound like tipping is unlikely (rather than uncertain) below 3C, if space allows I'd suggest clarifying that this is specifically the IPCC AR6 assessment and briefly mentioning that some studies do find evidence for past collapse within this range.

323:        Double citation.

385:        Pedantic, but "like the Greenland Ice Sheet" to be specific to the icy bit of Greenland.

402-403:        Should clarify the temperature level in G3 for unfamiliar, i.e. 2020 levels I believe. Also, maybe better to have this sentence integrated in to paragraph above rather than free-floating?

574-575:        Some caveats on phrasing here, as while we do refer to this as a "global/core tipping *element*", referring to a "global tipping *point*" in a permafrost context can be misread, as it is often associated with the idea of a runaway warming threshold in the permafrost carbon feedback at the global scale (which the recent assessment by Nitzbon et al. this year [https://www.nature.com/articles/s41558-024-02011-4] is clear in ruling out). Nitzbon et al. also assess the Yedoma scenario as unlikely (although they do support localised abrupt thaw as tipping, albeit without a specific threshold warming level), which is worth mentioning here for context. Lastly, more accurate to say "becomes more widespread at" rather than "could occur at", given some abrupt thaw is already happening (our threshold estimate here is for when localised tipping becomes regionally/globally widespread in a near-synchronous manner, as for coral reefs).

598-599:        Do you mean less effectively than limiting GMT through zero emissions?

655-669 / Fig. 4:            (6) is not explicitly labelled in caption (e.g. before "It sinks along the shelf edge").

680:        I think until after 2100 would be more accurate

709:        Anthropogenic global warming, for clarity (as can't rule out palaeo warming events triggering circulation collapse).

722:        Not a major issue, but is this the first time this compensation language is used in main text outside of table 1? It's useful, but feels like could be more consistent throughout if used.

885:        GHGs are defined in Intro, so no need to redefine here.

1161:        Relevant here is this recent paper, also in this SI: https://esd.copernicus.org/articles/15/671/2024/esd-15-671-2024.html

1196:        Phytoplankton are indeed under-studied on this, but corals & symbiont algae are relevant here (statement is equivalent for all though).

1232-1234:    I think this should be caveated more upfront here, making it clearer that you are advocating this system as being a plausible candidate for a tipping element for the first time here, with this section making the case for it being a tipping element as well as SRM's likely impact on it. Otherwise, as phrased it to me it can be read as saying it's already been established as plausible, which I don't think is the case yet. You should also rephrase the part in parentheses, as to me it implies that we suggest HTS as a candidate regional/impact element in AM22, rather than the category of regional/impact element being proposed in AM22.

1239-1242:    I think this is a good framing and caveat, as much like some threshold-free systems are considered in this paper the HTS doesn't need to be a tipping element per se to be considered, and there is value in considering SRM's affect on tipping cascades in socio-ecological systems. As it stands though, I personally don't think sufficient evidence is presented for tipping to be likely at the integrated system level, but that doesn't mean it shouldn't be discussed.

1246-1251:    The key difference between heterogeneity in HTS vs Amazon/corals is that shallow tropical coral reefs and the Amazon rainforest (NB, we consider the Amazon *rainforest* to be the tipping element, not the whole Amazon *basin*, as it shares a common tipping dynamic within same biome) are all the same biome/ecosystem functional group, whereas HTS integrates quite different biomes. Furthermore, the evidence presented for alternative states emerging are to me all evidence of ecosystem state change under continued pressure, but insufficient for full regime shifts to a new and self-sustaining alternative state. A degraded ecosystem can be quasi-stable under sustained pressure, but to prove it's in an alternative state / system attractor in a dynamical systems sense (the theoretical underpinning to tipping points), one would have to demonstrate that feedbacks exist that would maintain this new state even if the pressure relented, instead of allowing recovery to an original-like state (albeit with some adaptive differences between original and recovered state, as per Holling's ecological rather than engineering resilience). Invasive species are a candidate driver in this sense (e.g. cited in GTPR23 as playing a role in localised tipping in savannah/dryland ecosystems), though whether it could act as a driver at the integrated system level for the HTS rather than some specific habitats within it is harder to see.

1259-1261:    Good to be thinking about the alternative state would be (bearing in mind need for additional dynamical evidence mentioned above) – would low diversity grasslands be expected across all topography though, given the spatial and climatic heterogeneity?

1264-1266:    For HTS to be considered one integrated tipping element (in line with AM22/GTPR definitions, at least), I think the tipping dynamic for it either needs to be a process spanning and involving all of these sub-systems, or be the same tipping dynamic in each of them but with near-synchronous thresholds. As it stands it feels more like tipping is possible within each HTS ecological,

agricultural, and human sub-system without a clear common tipping dynamic or threshold across them all, with correlation in their degradation/tipping more likely to be due to sharing common drivers in warming and habitat loss/degradation rather than direct causation between them. This is to some extent true of coral tipping too, as tipping dynamics are localised to each reef rather than spanning the whole element, but the reason we grouped those as a tipping element anyway is that they share a common specific tipping dynamic (i.e. bleaching recurrence leading to mass mortality) with likely regionally-to-globally similar warming thresholds (i.e. ~1.5C) leading to near-synchronous tipping across the whole biome/functional group. Ideally then there'd be some discussion of whether there's a specific process that could feasibly span across the whole HTS system and precipitate tipping in its subsystems at approximately the same level of climate change, or highlight this as a gap to explore in future before HTS can be considered as a regional tipping element.

1276-1278:     As above, are these drivers for tipping in each HTS sub-system, or for integrated HTS system as a whole?

1294:          Are the Western Ghats part of HTS as defined? The west coast of India seems a bit far removed from Himalaya-to-Sundarbans. (Eastern Ghats would make more sense, but even they don't quite make it to edge of GBM drainage basin).

1321:          Suggest changing "Earth System" to "climate", as the Earth system view is that climate, biological, and human dimensions are all aspects of the overall Earth system they are all a part of.

1351:          AM22 not necessarily the best citation on this – we compiled more evidence & papers on boreal tipping dynamics in the GTPR23 biosphere chapter: https://global-tipping-points.org/section1/1-earth-system-tipping-points/1-3-tipping-points-in-the-biosphere/

1359, 1363 & Table 1:          Permafrost "melting" should be permafrost thawing for accuracy.

1394:          Cite Table 1 here as discussing results it summarises (could arguably have table here rather than up top too, but I can see arguments for placement at either place).

1398-1399:     Sentence a bit fragmented here, and probably need to clarify for readers why AMOC can overcompensate in previous sentence but not compensating here.

1400:          And for some elements, we're not fully sure of the relative importance of different drivers yet either.

1448:          I like this uncertainty typology, makes discussion very clear.

Table 1 & Supp. Info.:          SPG is misspelled as SGP in several places (e.g. SI line 18, 74; Table1 SPG row evidence strength column), so need to check through for that.

Dr. David A. McKay

---

## Author Response (AR2)

**Responses to Reviewers**

**Reviewer 1 Comments**

The authors have carefully carried out the recommendations by the reviewers and fully covered my concerns. In particular, the presentation improved much (Figures, Table). Still, if the authors see potential to streamline the text, I would appreciate these efforts as well as even clearer figures. In particular, Fig. 3 would profit from less abbreviations, number of question marks, etc.. This is similarly valid for Table 1. Apart from these technical corrections (that I leave to the authors to work over or not), I am very happy to recommend this paper for publication in ESD. And I am happy to congratulate the authors to this very large and important review paper.

We thank the reviewer for their very kind comments, and generally wish to thank the reviewers previous set of comments in helping to substantially shape the paper.

Unfortunately, we have not been able to streamline the text, particularly given the need to address reviewer 2 comments about the HTS.

To make Fig. 3 clearer, we have added the meanings of the abbreviations in the text description. We have reduced the abbreviations used in Table 1, using full words for the overall judgements, and clearer abbreviations (Worse for Worsening, No for No compensation, Part for Partial Compensation, Eff for Effective Compensation, Over for Overcompensation and Unk for Unknown).

**Armstrong-McKay Comments**

General comments:

The authors have substantially revised this manuscript, tightening the writing & structure and reducing repetition throughout, and in my view, it has improved considerably as a result. In particular, the authors have caveated their summary of overall SRM effectiveness for CTPs more thoroughly, emphasising the uncertainties brought by non-temperature drivers in the abstract, introduction, and discussion (while the counter-point made that this could also make SRM more effective, not only less, is fair enough too). The additional figures and tables are useful, better helping to demonstrate tipping dynamics, how SRM might intervene in these dynamics, and the paper's results. Table 1 has also improved with the addition of more explicit discussion of drivers, effectiveness, and confidence throughout, making it easier to discern key points. Other points raised, for example the ocean temperature focus for marine ice sheets, or current

general circulation / Earth system model limitations, have also been clarified, and the authors have justified their selection of systems to consider in this paper.

We thank the reviewer for his kind words and helpful feedback in both this, and the previous, round of reviews.

Beyond minor further suggestions, my main remaining comments concern the subtleties of categorising tipping in the Himalaya-to-Sundarbans (HTS) hydro-ecological system. While I appreciate the value of looking at HTS as an integrated socio-ecological system likely featuring localised tipping points, I am not yet convinced that it can be categorised as a regional/impact tipping element by the rationale of AM22 (even as a highly uncertain one) without a clear mechanism for shared tipping dynamics at the system rather than subsystem level beyond a common threshold. For comparison, warm-water coral reefs are classed as regional elements because there's evidence for a bleaching frequency threshold beyond which recovery is prevented and localised die-off becomes inevitable, with widespread mortality across the same biome/ecosystem functional group occurring at similar warming levels. Similarly, while Amazon rainforest dieback due to moisture recycling failure is also localised, it can trigger further dieback across much wider parts of the same system via that process, and is all one biome/functional group. In contrast, HTS covers multiple biomes/ecosystem functional groups, with the default likelihood being that different habitats that may tip are likely to tip due to different dynamics at different levels of climate change/degradation. It's still fine to include HTS in the paper though, but I think it either needs a little more justification as to how systemic tipping might emerge at the integrated system level, or a little more clarification that it's socio-ecological system with localised tipping, which this paper is suggesting as a potential element but is still a valuable case-study for considering SRM impacts on even if not.

We believed we have addressed many of the reviewers issues with the HTS section, with the comments related to this by the appropriate Specific Comment. The HTS section as a whole has been substantially revised to focus on what may make it a single integrated tipping element, whilst there is now also greater acknowledgement of the uncertainty of this hypothesis

Specific comments (by line no.):

23: I think "could" or "is likely" rather than just "is" is more appropriate here, given most of the evidence comes from simulations.

This has been done, and changed to 'could be'. Line 23.

38: A comma has escaped.

This has been deleted.

47: Should "reverse" be here as well (along with avoid/postpone)?

This has been added, line 48-49

61: "stop increasing" would be smoother than "stop to increase".

We agree, this has been changed, line 62.

130: "tipping elements" would be more consistent terminology than "earth systems" here, or Earth subsystems to differentiate from the Earth system as a whole.

This has been changed to "earth sub-systems (tipping elements)", line 131

144: Insufficient SRM possible/used is also highly relevant for only postponing tipping.

We have mentioned this now, although only discuss it here as it is different to the peak shaving scenario we use. We say the following on line 145-148:

"Moreover, if insufficient amounts of SRM were used - maintaining, for example, a constant SRM forcing rather than the constant Global Mean Surface Temperature (GMST) assumed in the peak shaving scenario - SRM may also only postpone tipping."

175 / Table 1: The table is much improved, but I had to repeatedly look back at caption to remind myself of what each letter meant in the second column. A simpler approach to consider might be pluses and minuses, e.g. - for worsen, ~ for negligible, + for partial compensation, ++ for effective compensation, and +++ for overcompensation (however, that'd mean +/-s for both driver and compensation direction, so that might be too confusing...), or just abbreviations e.g. over, part, etc. Also, I assume bolding for drivers means primary drivers, but I don't think this is stated (and is missing e.g. for MSC driver or HTS driver/reversibility). I was going to ask too whether it'd might make more sense to have Table 1 plus its description (lines 194-205) after Section 5 instead, but on reflection I can see it can be argued either way which makes more sense to the reader. Finally, I broadly agree with the categorisations, though I'd query a few of the overall ones, such as whether HTS should be U-P (as the uncertain effect of SAI on monsoons could be critical), whether AR should be U-P or W-P (as evidence / GCMs remain limited, and while I agree it works better in east than west, there remains an uncertain

risk of bringing west to point of bistability instead), whether SPG should be U-E (as no driver has an N or W, but there are few studies), and whether BPF should be P-E (as studies find reduced permafrost loss, but not totally countered, albeit potentially improvable via SAI strategy).

We have changed to shortened versions of the words (Worse (for Worsen), No (for No Compensation), Part (for Partial Compensation), Eff (for Effective Compensation), Over (for Overcompensation) and Unk (for Unknown). We also include the full words for the overall judgement of each tipping element.

We have included the initialisms in the figure caption to aid readability.

Bolded does mean primary drivers and we have added these.

We have amended the HTS system to Unknown, because of the possibly decisive influence of the monsoon.

For the Amazon Rainforest, we have clarified that in the West it is W-P, but do believe that the Eastern Amazon tipping would be more significant if the entire tipping point were to tip, so feel confident that we could say, with the clarification we have in place, that the Amazon is No compensation-Partial compensation.

For the SPG, whilst there are few studies, hence the maximum uncertainty, the studies that do exist don't suggest that it is likely to Worsen it however.

For the BPF, the reason we are comfortable saying it is Effective is that the Effective range is 75%-125% compensation. Both studies that allow percentages to be calculated (Chen et al., 2023; Liu, Moore and Chen, 2023) exceed our 75% threshold.

Fig. 3: I believe the compensations here are the overall compensation judgments in Table 1 (rather than temperature alone), but would be useful to state in the legend or caption for clarity.

This has been clarified, we have added "*The compensation and uncertainty judgements is our assessment for the overall effect on drivers from Table 1.*"

222-244 & 286-287: This is indeed the IPCC AR6 summary, but the nature of confidence language makes it sound less compelling than I think the evidence suggests (especially from palaeo studies) - AR6 for example reported several studies where total loss committed at 2-3°C (ch.9-pg.78). As such, so that it doesn't sound like tipping is unlikely (rather than uncertain) below 3C, if space allows I'd suggest clarifying that this

is specifically the IPCC AR6 assessment and briefly mentioning that some studies do find evidence for past collapse within this range.

We have edited this, adding in sentences in both paragraphs. We clarify that the uncertainty judgements are from the IPCC AR6 report, cite paleo evidence, and highlight that Lenton et al. 2023 puts the critical threshold at lower than the IPCC for both.

323: Double citation.

This has been removed.

385: Pedantic, but "like the Greenland Ice Sheet" to be specific to the icy bit of Greenland.

This has been changed

402-403: Should clarify the temperature level in G3 for unfamiliar, i.e. 2020 levels I believe. Also, maybe better to have this sentence integrated in to paragraph above rather than free-floating?

We have clarified that G3 is projected 2020 levels and the paragraph has been integrated.

574-575: Some caveats on phrasing here, as while we do refer to this as a "global/core tipping element", referring to a "global tipping point" in a permafrost context can be misread, as it is often associated with the idea of a runaway warming threshold in the permafrost carbon feedback at the global scale (which the recent assessment by Nitzbon et al. this year [https://www.nature.com/articles/s41558-024-02011-4] is clear in ruling out). Nitzbon et al. also assess the Yedoma scenario as unlikely (although they do support localised abrupt thaw as tipping, albeit without a specific threshold warming level), which is worth mentioning here for context. Lastly, more accurate to say "becomes more widespread at" rather than "could occur at", given some abrupt thaw is already happening (our threshold estimate here is for when localised tipping becomes regionally/globally widespread in a near-synchronous manner, as for coral reefs).

We have edited the text, including adding "Others, however, have suggested that no such global mean temperature threshold applies, with global permafrost loss being quasi-linear in global warming throughout its decline (Nitzbon et al., 2024). If asuch a global temperature threshold at 4°C exists"

598-599: Do you mean less effectively than limiting GMT through zero emissions?

No, we meant that the fractional restoration of permafrost is modeled to be smaller than that of GMT. The same logic as with the % compensations in table 1. I.e. when cooling from 2 to 1C, the permafrost might change from that of a 2C GHG-warming world to a 1.2C GHG-warming world. We have edited the text to make this clear, saying "However, global SRM strategies typically under-restore permafrost relative to their impact on global mean temperature"

655-669 / Fig. 4: (6) is not explicitly labelled in caption (e.g. before "It sinks along the shelf edge").

This has been added there

680: I think until after 2100 would be more accurate

We have edited it, as these scenarios also don't expect post 2100 collapse. Rather, it now reads "in general models do not predict collapse for SSP scenarios extending to until 2100 (Weijer 2020), although some models show collapse for extreme hosing (Jackson 2023, van Westen 2023) or warming (Hu et al., 2013). "

709: Anthropogenic global warming, for clarity (as can't rule out palaeo warming events triggering circulation collapse).

This has been changed

722: Not a major issue, but is this the first time this compensation language is used in main text outside of table 1? It's useful, but feels like could be more consistent throughout if used.

Whilst we somewhat agree, time constraints in this review iteration has precluded us from doing so

885: GHGs are defined in Intro, so no need to redefine here.

Edited.

1161: Relevant here is this recent paper, also in this SI:
https://esd.copernicus.org/articles/15/671/2024/esd-15-671-2024.html

We have added in discussion of this paper. In particular, we have said

"The response of coral calcification to acidification is generally linear and highly species specific, so a simple 'coral acidification tipping point' does not exist. Other factors, such as internal pH regulation, may have physiological tipping, points, but manifest as linear decreases at

an ecosystem-wide level. However, coral reefs are complex communities with non-coral species playing important roles, and whilst most acidification impacts are linear, there does seem to be some evidence of tipping on a local scale due to the indirect effects of acidification on the overall health of the community in specific habitats, particularly those with an already high $pCO_2$ (Cornwall et al. 2024). Nonetheless, these are unlikely to manifest as a global, near-synchronous, tipping point."

1196: Phytoplankton are indeed under-studied on this, but corals & symbiont algae are relevant here (statement is equivalent for all though).

We agree and have changed phytoplankton to "zooxanthanae algae"

1232-1234: I think this should be caveated more upfront here, making it clearer that you are advocating this system as being a plausible candidate for a tipping element for the first time here,

Acknowledged as suggested

with this section making the case for it being a tipping element as well as SRM's likely impact on it. Otherwise, as phrased it to me it can be read as saying it's already been established as plausible, which I don't think is the case yet.

Thank you for this suggestion, we changed the text as recommended here.

You should also rephrase the part in parentheses, as to me it implies that we suggest HTS as a candidate regional/impact element in AM22, rather than the category of regional/impact element being proposed in AM22.

Changed as suggested in this section

1239-1242: I think this is a good framing and caveat, as much like some threshold-free systems are considered in this paper the HTS doesn't need to be a tipping element per se to be considered, and there is value in considering SRM's affect on tipping cascades in socio-ecological systems. As it stands though, I personally don't think sufficient evidence is presented for tipping to be likely at the integrated system level, but that doesn't mean it shouldn't be discussed.

We have substantially edited the section to acknowledge that what we are essentially presenting is a hypothesis of integrated, systemic tipping, and acknowledge it is based on limited evidence. We also agree that this is not necessary for inclusion in the paper, and acknowledge the (significant) possibility that this may not in fact be a single, integrated tipping element, but a collection of interdependent ecological tipping

elements that may not show systemic tipping. Nonetheless, we do also think that we have included a stronger case for integration here.

1246-1251: The key difference between heterogeneity in HTS vs Amazon/corals is that shallow tropical coral reefs and the Amazon rainforest (NB, we consider the Amazon rainforest to be the tipping element, not the whole Amazon basin, as it shares a common tipping dynamic within same biome) are all the same biome/ecosystem functional group, whereas HTS integrates quite different biomes.

Deleted 'basin' ; while the rainforest is considered a biome, there are many different forest types within this biome in the Amazon and across other areas. However, it is clearly true that the HTS consists of many biome types. Not to argue this point too far, the concept of 'biomes' is poorly defined and descriptive, and every map of the world's biomes defines them differently, so one doesn't want to rest a definition of tipping elements too much on this as a foundation. Ecologists treat the concept of tipping somewhat differently than do climate scientists, although no less seriously and urgently, and similarly as total system changes, so there's that as an issue here as well.

Furthermore, the evidence presented for alternative states emerging are to me all evidence of ecosystem state change under continued pressure, but insufficient for full regime shifts to a new and self-sustaining alternative state. A degraded ecosystem can be quasi-stable under sustained pressure, but to prove it's in an alternative state / system attractor in a dynamical systems sense (the theoretical underpinning to tipping points), one would have to demonstrate that feedbacks exist that would maintain this new state even if the pressure relented, instead of allowing recovery to an original-like state (albeit with some adaptive differences between original and recovered state, as per Holling's ecological rather than engineering resilience). Invasive species are a candidate driver in this sense (e.g. cited in GTPR23 as playing a role in localised tipping in savannah/dryland ecosystems), though whether it could act as a driver at the integrated system level for the HTS rather than some specific habitats within it is harder to see.

We have revised the text to make a stronger case for possible tipping of this entire system. The comparisons between tipping to alternative states in the ecological literature are very interesting points here, but would be too tangential to discuss at length in this paper. A sentence was added mentioning this. It is well established in the ecological literature that many novel ecosystem states are irreversible, and species extinctions, particularly for dominant or keystone species, that certainly is irreversible. Because large numbers of species extinctions are central to the changes discussed here, it is a solid assumption that the system itself is likely to be tipped irreversibly; invasive species, soil erosion, and other changes add additional nails to the coffin.

1259-1261: Good to be thinking about the alternative state would be (bearing in mind need for additional dynamical evidence mentioned above) – would low diversity grasslands be expected across all topography though, given the spatial and climatic heterogeneity?

This is merely speculative on our part, based on some observed changes in this and other systems. But it is uncertain.

1264-1266: For HTS to be considered one integrated tipping element (in line with AM22/GTPR definitions, at least), I think the tipping dynamic for it either needs to be a process spanning and involving all of these sub-systems, or be the same tipping dynamic in each of them but with near-synchronous thresholds. As it stands it feels more like tipping is possible within each HTS ecological, agricultural, and human sub-system without a clear common tipping dynamic or threshold across them all, with correlation in their degradation/tipping more likely to be due to sharing common drivers in warming and habitat loss/degradation rather than direct causation between them.

We are speculating that there is a real possibility that what links them are the two sources of water on which all of the ecological and human systems depend: the river systems originating in the Himalayan glaciers, and the monsoon. Warming and land-use change, including river damming, exacerbate and interact with these central, fundamental drivers, the sources (and timing) of water. While climate and SRM might drive complex changes in water from these two sources (e.g. snow vs. rain in the Himalaya, flooding vs. drought (both are not only likely, but happening already), changes in the seasonality of water availability relative to temperature seasonality, rising sea levels and coastal erosion), these effects on water are felt across a very large scale and across diverse systems. It is unclear and perhaps unlikely that there is a single, unique threshold (that is, a point that is reached for a certain number of mm of water on a certain day) for all of the HTS, but that is really also the case for more familiar systems such as coral reefs and Amazon forests. For example, habitat fragmentation due to development and road construction in the Amazon rainforest, combined with drought and high temperatures and human activity, makes fires more frequent, intense and extensive, and these combine to change forest permanently to degraded grassland/shrubland; but this happens in a mosaic patchwork, not to the entire region in a single swipe. One of the authors in particular (JG) also believes that readers need to be prodded to think more broadly at this difficult interface between climate and ecological science, and to consider other less-familiar systems like the HTS, but of course it should be introduced in a rigorous manner.

This is to some extent true of coral tipping too, as tipping dynamics are localised to each reef rather than spanning the whole element, but the reason we grouped those as a tipping element anyway is that they share a common specific tipping dynamic (i.e.

bleaching recurrence leading to mass mortality) with likely regionally-to-globally similar warming thresholds (i.e. ~1.5C) leading to near-synchronous tipping across the whole biome/functional group. Ideally then there'd be some discussion of whether there's a specific process that could feasibly span across the whole HTS system and precipitate tipping in its subsystems at approximately the same level of climate change, or highlight this as a gap to explore in future before HTS can be considered as a regional tipping element.

We are positing that the driver is a single connected hydrological system of water availability across the entire HTS region, originating from the Himalayan glaciers feeding a network of major river systems that cross the subcontinent and end in the Indian Ocean, plus the regional monsoon. It is at present difficult to identify a clear threshold, but it seems plausible that the entire system would be affected by these changes in a linked manner. There are certainly great gaps in our awareness and understanding of these linked, potentially integrated changes!

1276-1278: As above, are these drivers for tipping in each HTS sub-system, or for integrated HTS system as a whole?

We are arguing that these hydrological drivers are linked for the HTS system as a whole.

1294: Are the Western Ghats part of HTS as defined? The west coast of India seems a bit far removed from Himalaya-to-Sundarbans. (Eastern Ghats would make more sense, but even they don't quite make it to edge of GBM drainage basin).

You are absolutely correct! That was incorrect and has been deleted.

1321: Suggest changing "Earth System" to "climate", as the Earth system view is that climate, biological, and human dimensions are all aspects of the overall Earth system they are all a part of.

Changed to "climate" as recommended.

1351: AM22 not necessarily the best citation on this – we compiled more evidence & papers on boreal tipping dynamics in the GTPR23 biosphere chapter: https://global-tipping-points.org/section1/1-earth-system-tipping-points/1-3-tipping-points-in-the-biosphere/

Citations more directly relevant have been used now.

1359, 1363 & Table 1: Permafrost "melting" should be permafrost thawing for accuracy.

Changed to "thawing" or "thaw" throughout.

1394: Cite Table 1 here as discussing results it summarises (could arguably have table here rather than up top too, but I can see arguments for placement at either place).

We have cited it.

1398-1399: Sentence a bit fragmented here, and probably need to clarify for readers why AMOC can overcompensate in previous sentence but not compensating here.

We have edited this section for clarity, saying " For two tipping elements, the effect of SRM at a minimum did not compensate for the overall effect of climate change on their drivers. For one of these two tipping elements, AMOC, we determine the range of feasible impacts of SRM to extend from not compensating for, to overcompensating the impacts of climate change on its drivers - this is the only tipping element where SRM was seen to feasibly overcompensate the overall effect of climate change on its drivers."

1400: And for some elements, we're not fully sure of the relative importance of different drivers yet either.

We have added in a sentence stressing this "Furthermore, our 'overall judgements' were based on our assessment of the relative importance of different drivers, and for many tipping elements this is not fully known."

1448: I like this uncertainty typology, makes discussion very clear.

Thank you

Table 1 & Supp. Info.: SPG is misspelled as SGP in several places (e.g. SI line 18, 74; Table1 SPG row evidence strength column), so need to check through for that.

We have changed this

Author Roles in reviewer comments: GF responded to all reviewer comments except for those concerning the biosphere section which were responded to by YF and JG